# Multimodal single cell analyses reveal gene networks of planarian stem cell differentiation

Alberto Pérez-Posada [1,2,3] ✉, Helena García-Castro [1,2,3], Elena Emili[1,8], Anna Guixeras-Fontana [4], Virginia Vanni [1,2,3], David Salamanca-Diaz[1,2,3], Cirenia Arias-Baldrich[1], Siebren Frölich [5], Simon J. van Heeringen[5], Francesc Cebrià [4,6], Nathan Kenny [7] & Jordi Solana [1,2,3] ✉

Cell type identity is controlled by gene regulatory networks (GRNs), where transcription factors (TFs) regulate target genes (TGs) via open chromatin regions (OCRs), often specific to one or multiple cell types. Classic GRN discovery using perturbations is laborious and not easily scalable across the tree of life. Single-cell transcriptomics enables cell type-resolved gene expression analysis, but integrating perturbation data remains difficult. Here, we investigate planarian stem cell differentiation by integrating single-cell transcriptomics and chromatin accessibility data. The integrated analysis identifies gene networks matching known TF interactions and highlights TFs that may drive differentiation across multiple cell types. Our data reveals at least two major cell type supergroups linked by their regulatory logic, including *alx3-1+* cells, comprising muscle, neurons and secretory cells, and *hnf4+* cells, comprising gut phagocytes, goblet cells and parenchymal cells. We validated our data demonstrating high overlap between predicted targets and experimentally validated differentially regulated genes. Overall, our study integrates TFs, TGs and OCRs to reveal the regulatory logic of planarian stem cell differentiation, showcasing a comprehensive catalogue of GRN computational inferences that will be key to study this process.

Gene regulation underlies many cellular decisions, including cell fate and identity. Pluripotent stem cells undergo distinct molecular changes as they differentiate into mature cell types, including changes in gene expression and chromatin dynamics[1]. These changes involve different kinds of genetic regulators, such as chromatin remodelers and transcription factors (TFs). Remodelers play a critical role in TF regulation, as chromatin marks and accessibility facilitate their binding and interaction with chromatin. Transcription factors often function in multiple cell types, stages or conditions in context-specific ways depending on the co-expression of other factors[2–4]. Ultimately, these factors orchestrate the transcription of specific targets, thereby determining cell type identity. Thus, cell differentiation comprises the combined expression of TFs and the combined accessibility of open chromatin regions (OCRs) acting as cis-regulatory elements (CREs). This combination creates a 'regulatory logic' forming gene regulatory networks (GRNs)[5–7]. While the general dynamics of this process have

[1]Department of Biological and Medical Sciences, Oxford Brookes University, Oxford, UK. [2]Living Systems Institute, University of Exeter, Exeter, UK. [3]Department of Biosciences, University of Exeter, Exeter, UK. [4]Departament de Genètica, Microbiologia i Estadística, Facultat de Biologia, Universitat de Barcelona, Barcelona, Spain. [5]Department of Molecular Developmental Biology, Radboud University, Nijmegen, The Netherlands. [6]Institut de Biomedicina de la Universitat de Barcelona (IBUB), Barcelona, Spain. [7]Department of Biochemistry, University of Otago, Dunedin, New Zealand. [8]Present address: Light Imaging Facility, Epigenetics and Neurobiology Unit, EMBL Rome, Rome, Italy. ✉e-mail: ap.posada1@gmail.com; j.solana@exeter.ac.uk

been studied in a number of model species[8–11], the mechanisms governing cell differentiation into various lineages remain largely unexplored in most multicellular organisms.

Single-cell methods have transformed the study of differentiation trajectories in a variety of animal species[12–14]. The initial step in characterising the potential differentiation pathways of pluripotent stem cells consists in identifying their distinct differentiation products, a task accomplished through single-cell transcriptomics (scRNA-seq)[15,16]. This technique enables the identification of expressed transcripts within individual cells, allowing for the grouping of cells into specific cell types. Computational algorithms are then employed to reconstruct the transitional states between stem cells and each differentiated cell type[17–19]. However, despite the ability to characterise the expression of transcripts, uncovering the GRNs governing their activity remains challenging.

Recently, novel single-cell methods have emerged to characterise the chromatin state to reveal OCRs and CREs. These methods leverage the assay for transposase-accessible chromatin with sequencing (ATAC-seq)[20,21], which identifies OCRs, including the enhancers and promoters that play a pivotal role in transcriptional regulation. Single-cell ATAC-seq (scATAC-seq) has been successfully employed in various models and paradigms[22–28]. One major challenge lies in integrating scATAC-seq data with scRNA-seq data and extracting regulatory information from the combination of chromatin accessibility and expression data[29,30]. Recent single-cell technologies predict TF/target gene interactions across various contexts[31,32] but often lack experimental validation, and it is unclear if these methods can scale beyond individual tissues to whole complex organisms.

Planarians are an ideal model organism to address this challenge as they have adult pluripotent stem cells that constantly differentiate to replace aged cells of all cell types[33,34]. A single planarian stem cell can differentiate into all cell types of the adult worm[35]. These cells also enable planarians' amazing regenerative capacities[36–38]. Transcription factors and epigenetic regulation have been already studied in planarians[39–43]. Using scRNA-seq, the major differentiated cell types that mature from planarian stem cells have been described[44,45]. Planarians are also very amenable to gene knockdown by RNAi[46]. Single-cell analysis techniques hold significant potential for investigating RNAi knockdown experiments, but there are still several challenges that need to be addressed[47–50]. Cell dissociation techniques can trigger stress responses and introduce biases, resulting in cell death and variations in cell survival rates[51–53]. Additionally, including different samples with current methods can introduce batch effects[54]. However, fixative cell dissociation approaches like ACME can mitigate the first concern by minimising stress-induced effects[55]. Moreover, combinatorial single-cell transcriptomic approaches like SPLiT-seq enable sample multiplexing and facilitate convenient multi-sample experiments[56]. By combining ACME and SPLiT-seq, it becomes possible to analyse multi-sample experiments, such as RNAi knockdown studies, with greater efficiency and accuracy[57,58].

Here, we report the first integration of scRNA-seq and scATAC-seq in *Schmidtea mediterranea*, in a whole adult organism. We combined 98,363 single-cell transcriptomes with 3659 single-cell ATAC profiles. Using the graph-based correlational tool WGCNA, we predicted gene sets and OCRs active in one or more broad types. We predicted key transcription factors involved in the differentiation of all major cell lineages derived from planarian stem cells. We predicted TFs influential in each broad cell type, and their targets, using ANANSE, a graph analysis computational approach[59]. Our results reveal two major cell type supergroups according to their regulatory logic, including transcriptomic, accessibility and transcription factor data: the *alx3-1*+ cells including neurons, muscle and secretory cells, and the *hnf4*+cells including gut phagocytes, goblet cells and the recently described

parenchymal cells. To validate our findings, we reanalysed previously published TFs knockdown data, revealing agreement with our predictions. Finally, we performed RNAi of *hnf4* coupled with single-cell analysis, confirming that it regulates parenchymal cells in addition to gut phagocytes. Altogether, our experiments reveal the regulatory logic of planarian stem cell differentiation and how this translates into major supergroups of cell type affinity. Our results underscore that the characterisation of all differentiation trajectories, and the GRNs that underlie them, is possible by combining single-cell methods and perturbation experiments with single-cell resolution.

## Results

### An integrated atlas of planarian stem and differentiated cells

To understand the regulation of differentiation from pluripotent stem cells to all adult cell types in planarians we generated an integrated multimodal single-cell atlas with scRNA-seq and scATAC-seq data. We compiled previously generated datasets[55,57] as well as newly generated experiments using ACME and SPLiT-seq (Fig. 1A, Supplementary Data 1). On the other hand, we used Trypsin dissociation and the 10X Genomics commercial approach to obtain a scATAC-seq dataset (Fig. 1A, Supplementary Data 1). We mapped these datasets to the recently released version of the *S. mediterranea* genome[60] (Supplementary Data 2). To analyse scATAC-seq data and obtain cell clusters we used CellRanger[61] and Seurat[62]. This allowed us to obtain a dataset with 3659 cells distributed in 11 clusters (Fig. 1B). We processed SPLiT-seq data with our analysis pipeline[55,63] and Seurat, to obtain a total of 98,363 cells in 59 cell clusters. We elucidated their identity with a previously published dataset (Fig. 1B, Supplementary Fig. 1, Supplementary Data 3, Supplementary Figs. 1, and 2A)[57]. The average numbers of UMIs and genes quantified per cell remain low, as characteristic of SPLiT-seq (Supplementary Fig. 3A,B). However, the integrated approach increases the total number of reads in each cluster, and therefore, the total number of genes quantified (Supplementary Fig. 2C). We then integrated the scRNA-seq and scATAC-seq datasets to transfer the known scRNA-seq identities to the scATAC-clusters (Fig. 1B, C, Supplementary Fig. 2B, Supplementary Data 3) using canonical-correlation analysis (CCA)[64,65]. This approach leverages correlation between expression data and scATAC signal detected within gene bodies. While the scATAC-seq data was shallower and less resolved than the scRNA-seq, the assay managed to detect open chromatin profiles for all major planarian broad cell types[44,45], including neoblasts, three stages of epidermal differentiation, phagocytes, basal/goblet cells, muscle, neurons, parenchymal cells (referred to as *cathepsin*+ cells in other single-cell studies[44]), protonephridia and secretory cells (referred to as parapharyngeal or parenchymal in other single-cell studies[44,66]) (Supplementary Fig. 4A–C). We grouped scRNA-seq clusters in 11 corresponding broad groups using this integration data (Supplementary Data 3, Supplementary Fig. 2).

To identify genes with both open chromatin and gene expression specific to each cluster we cross-referenced the markers of both datasets (Fig. 1D, Supplementary Fig. 4D–U). We inspected the genomic regions identified, with their associated gene annotations and open chromatin peaks (Supplementary Fig. 5A). We then obtained genomic coverage tracks of the scATAC-seq (Supplementary Fig. 5B) and scRNA-seq (Supplementary Fig. 5C) signal of these regions, for each broad group. This analysis included a bulk ATAC-seq sample that showed good agreement with the scATAC-seq (Supplementary Fig. 6). Our scATAC-seq dataset contained multiple regions specifically open in each of the major differentiated types (Fig. 1E). Altogether, these analyses revealed genes with both open chromatin features and RNA expression, validating the quality of the scATAC-seq data. This shows our integrated multimodal dataset captures the transcriptomic and epigenomic landscape of each planarian differentiated broad type.

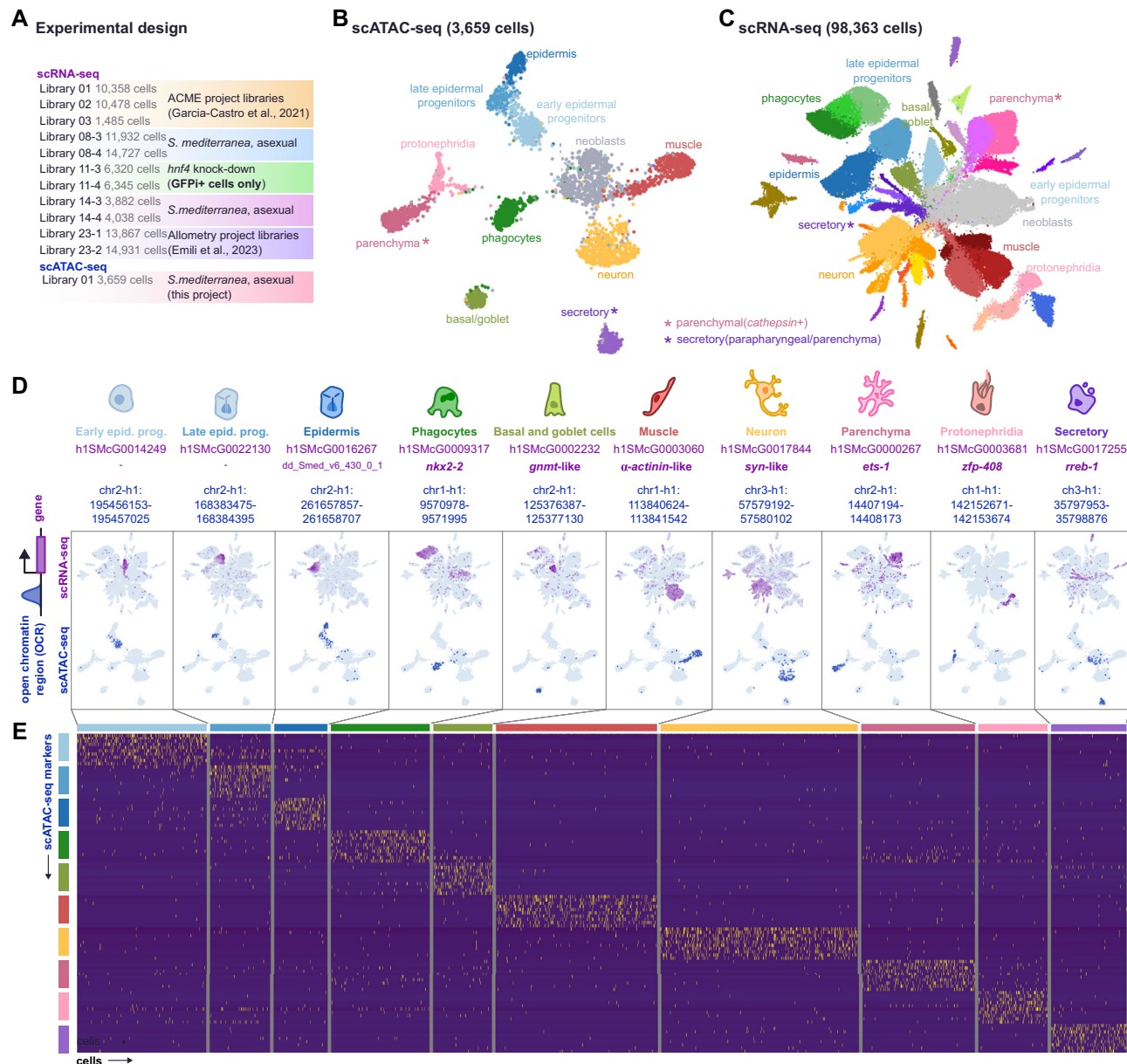

**Fig. 1 | An integrated atlas of planarian differentiated cells. A** Summary of the experimental design of the different libraries. **B** UMAP projection of 3659 individual cellular profiles of chromatin accessibility in *S. mediterranea*. **C** UMAP projection of 98,363 individual cellular transcriptomes in *S. mediterranea*. **D** Feature plots of ten different genes (up) and open chromatin regions (down) specific to broad differentiated cell types in *S. mediterranea*. **E** Heatmap of markers of scATAC-seq cells, depicting chromatin regions specifically open in each of the broad differentiated cell type categories.

## The transcriptomic landscape of planarian stem and differentiated cells

Gene expression is dynamic: some genes are expressed broadly in all cell types and tissues, while others are very highly specific to one cell type. Genes are often expressed in multiple cell types[67,68], and likely the combination of genes expressed in each cell type defines their identity. In single-cell analysis, marker finding algorithms usually perform one-against-all comparisons. This approach often excels at revealing genes very specific to any one cell type, at the expense of genes expressed in multiple cell types. One approach that overcomes this limitation is Weighted Gene Coexpression Network Analysis (WGCNA)[69,70] as it detects modules of co-expressed (correlated) genes regardless of their correlation being in one or more cell types (Supplementary Note 1).

We investigated gene co-expression across the different cell types of our dataset using WGCNA at a pseudobulk level, which led to high sensitivity in contrast with the sparsity of individual single-cell data

points. We obtained a total of 77 modules of co-expression; 24 of these modules had average expression peaking in one single cell type ('sE' modules), and 53 modules had expression peaking in multiple cell types ('mE' modules, Fig. 2A, Supplementary Figs. 7, and 8A, Supplementary Data 4, See Methods). Our classification largely agrees with other metrics for specificity like τ from Yanai et al.[67] (Fig. 2A, Supplementary Note 1). These included modules composed of genes with expression in similar cell types (i.e. in two or more types of the same broad group), such as epidermis (mE05, mE07), phagocytes (mE08, mE23), muscle (mE24), neurons (mE52), parenchyma (mE32, mE33, mE41, mE42), protonephridia (mE18), or secretory cells (mE51). Interestingly, we also found 'mE' modules expressed in distinct broad cell types (Supplementary Data 4, 5, 6, 7), such as modules with expression in neuronal and muscle types (mE26, mE27), modules with expression in the pharynx cell type and *psd*+ cells (mE19), which are both pharyngeal, and modules containing cilia genes (mE50 and mE53) with

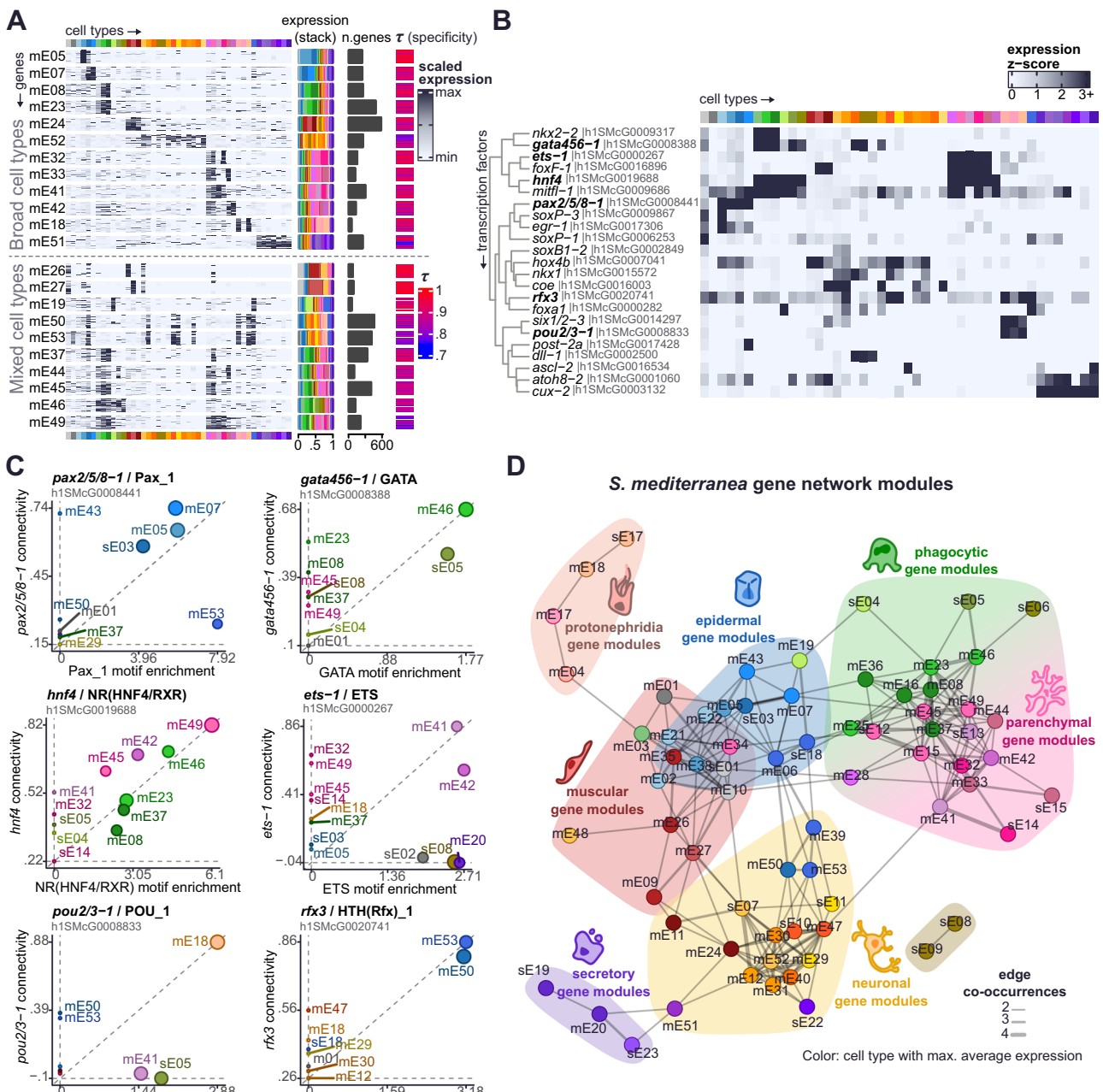

**Fig. 2 | The transcriptomic landscape of planarian stem and differentiated cells.** **A** Heatmap of gene expression of genes from several modules of co-expression. Genes in rows, grouped in modules, and cell types in columns. A subsample of twenty genes per module is shown. Right annotation of the heatmap: from left to right, (i) stacked bar plot of average gene expression, (ii) number of genes in modules; (iii) Tau metric of expression specificity (as calculated in Yanai et al., 2005). **B** Heatmap of gene expression of well-known planarian Transcription Factors (TFs). TFs in rows, cell types in columns. TFs have been grouped based on similarity of gene expression, and TFs showed in panel **C** are highlighted. **C** Scatter plot showing the connectivity of the planarian TFs to modules of co-expression, in relation to the motif enrichment of the same TF class in the promoters of genes from modules of co-expression. Dot size indicates significance of hypergeometric test (*q*-value < 0.1). **D** Module-wise network of modules in *S. mediterranea*. Nodes represent co-expression modules; edges represent connections between them. Edge width indicates the number of occurrences of a given module-module connection in different analyses. Shaded areas represent module communities.

expression in epidermal, neuronal and protonephridia types. We observed several modules of co-expression in both gut and parenchymal cell types, including mE37, mE44, mE45, mE46 and mE49. These observations indicated that gut and parenchymal broad types are strongly associated in their gene expression patterns. Taken together, our WGCNA analysis revealed modules of genes expressed in multiple distinct cell types that likely underlie cell similarities.

To understand the regulatory logic of these modules of gene co-expression, we annotated and analysed the expression of transcription factors. Using a combined approach of sequence homology, identification of DNA binding protein domains and literature curation (expanding on Neiro et al.[39] and similar to King et al.[71]), we annotated 665 TFs (Supplementary Data 2) and identified a set of 517 TFs with cell type-specific expression in our scRNA-seq dataset. These TFs showed specific expression in one or more cell types and were highly connected (correlated) to one or more modules (Fig. 2B, Supplementary Figs. 7, and 8B, Supplementary Note 1). Some of them were more highly connected to modules of multiple cell types than to modules of

single cell types. For example, we detected high connectivity of well-known TFs, such as *foxa*[72], *gata456-1*[73], and *foxF-1*[74], in modules mE19, mE46 and mE41, respectively (Supplementary Fig. 8B; labels on the right side). Interestingly, the highest connectivity of transcription factor *hnf4* was to module mE49, a mixed module of genes expressed in gut and parenchymal cells (Fig. 2B, Supplementary Fig. 8B), agreeing with its expression in both gut phagocytes and parenchymal cell types (Fig. 2B). These patterns highlighted TFs that may regulate gene expression in several cell types.

In agreement with the connectivity, we observed enrichment of motifs for the same transcription factors in the promoters of the genes from the same modules (Fig. 2C, Supplementary Data 8, Supplementary Fig. 8C). For example, TF *pax2/5/8–1*[75] was highly connected to epidermal modules mE05, mE07, and sE03, which had high enrichment of a Pax motif. The TF *gata456-1*[73,76] was highly connected to gut modules mE46 and sE05, whose gene promoters show high enrichment of the GATA motif. The nuclear receptor motif, associated to *hnf4*, is highly enriched in the gut and parenchyma modules mentioned above. Interestingly, we observed an orthologue of the *Rfx* TF family highly connected to cilia modules, whose gene promoters were enriched in this motif. This agrees with previously described functions of *Rfx* in cilia formation[77–79]. Other examples include *pou2/3-1*[80,81] in protonephridia module mE18, *ets-1* in parenchymal modules, and *egr*[82] in epidermal modules, among others (Supplementary Fig. 8D).

To further explore the dynamics of these modules we analysed their cross-connections (i.e. the number of connections between genes of different modules), similarity in motif enrichment, similarity in functional category enrichment between modules, and the overall profile of TF connectivity of each module (Supplementary Fig. 9, Supplementary Data 8, Supplementary Note 1). We retrieved similar connections between modules across all these analyses (Fig. 2D). For example, and most prominently, parenchymal and gut modules were highly connected within themselves, but they also shared many cross-connections. We also observed connections between muscle, secretory and neuronal modules, and also between neuronal, pharynx, and cilia modules.

Overall, our analyses suggest the existence of several major programmes of gene expression controlled by similar groups of TFs, which likely involve gene regulation of specific major cell types, but also across multiple major cell types.

## The chromatin accessibility landscape of planarian cells
Based on the idea that genes are expressed in multiple cell types and likely controlled by multiple TFs, we wondered if similar patterns were also observable at the chromatin level. To investigate this, we examined chromatin accessibility dynamics across cell types using the same weighted correlation network approach of WGCNA. Using 14,397 OCRs, we detected 67 modules of co-accessibility, or OCR modules, across multiple cell types (Fig. 3A, Supplementary Fig. 10A–D, Supplementary Data 9). This set of OCRs robustly groups the 'differentiated' (i.e. non-neoblast) cell types into several higher order groups: (1) the epidermal lineage, (2) a group of phagocytes, parenchymal cells, basal/goblet cells, and protonephridia, and (3) a group of muscle cells, neurons, and secretory cells (Fig. 3A, Supplementary Fig. 10A). These groups tend to share regions of open chromatin, as illustrated by the co-accessibility modules with peak openness in multiple cell types; for example, epidermal OCR modules mO11, mO12, mO48; gut/parenchymal OCR modules mO24, mO50, mO25, mO39, mO57, mO43; or neuron/muscle/secretory modules mO55, mO33, mO37, mO52. Many of these modules revealed OCRs co-accessible in neoblasts and one or more cell types, suggesting OCRs important for differentiation trajectories, such as mO01, mO03, or mO08. Still, OCRs of many of these modules appeared relatively accessible in neoblasts, which agrees with our previous observation that neoblasts showcase a heterogeneous profile of chromatin accessibility (Supplementary Fig. 4), and aligns

with previous observations that neoblasts lack a specific chromatin signature[40]. Interestingly, we observed many of these OCRs lay next to genes that are expressed in the same cell types (Fig. 3A, right panel), and that OCR accessibility and expression of nearby genes tend to correlate positively (Supplementary Fig. 10E). This is further supported by the enrichment of gene/OCR pairs between gene and chromatin modules associated to the same cell types (Supplementary Fig. 10F, G, Supplementary Data 10), which also revealed associations between parenchymal and gut cell types, and neurons, muscle, and secretory cells.

Motif enrichment analysis of OCRs from each chromatin module suggests an underlying regulatory logic (Fig. 3B, Supplementary Fig. 11). For example, motifs of planarian epidermal TFs such as Pax, Sox and p53[75,82] appear enriched in epidermal chromatin modules. We also detected enrichment of motifs from TFs linked to intestinal fate (GATA, HNF4, Nkx/Bapx) in phagocytes and basal/goblet OCR modules. The HNF4 motif was also detected in OCRs modules also co-accessible in parenchymal cells, which also showcased enrichment of the ETS and Forkhead motifs. The latter was also found in a module of OCRs co-accessible in muscle, in agreement with previous observations[74]. Modules of OCRs co-accessible in neurons were enriched in the Sox family of motifs, in agreement with the described expression and role of *soxB-1*[83], and in the NFY motif, which has been suggested as a global regulator of neuronal cell types in planarians and other animals[84]. We found a POU motif enriched in modules of protonephridia OCRs, which was different from a second POU motif only enriched in OCRs co-accessible in muscle, neurons, and secretory cells. Interestingly, many of these motifs agreed with the expression of TFs in these broad cell types (which we measured as the connectivity between a TF and the accessibility profile of modules), as shown in Fig. 3C and Supplementary Fig. 10H–P. We observed similar trends when performing WGCNA on a more conservative dataset of OCRs found to be differentially accessible in differentiated cell types compared to neoblasts (Supplementary Note 2, Supplementary Data 11, 12, 13, 14). When comparing chromatin accessibility of neoblasts to that of other cell types, we did not find evidence of OCRs specifically accessible only in neoblasts, in consistency with our and others' observations[40] (Supplementary Data 11).

Overall, our single-cell chromatin accessibility data show group similarities in the chromatin landscape of several planarian cell types, aligning with our observations at the gene expression level. These similarities suggest common regulatory principles between multiple planarian cell types.

## Networks of influential TFs for planarian cell fates
Recent efforts in the community have sought to integrate multimodal data such as gene expression and chromatin dynamics to establish associations between TFs and target genes (TGs) via binding of TFs to regions of open chromatin, and to study these regulatory programmes at the network level[7,85,86]. To combine gene expression and chromatin accessibility data, we used ANANSE[59], a tool that leverages gene expression, chromatin accessibility, distance from the Transcription Start Site (TSS), and motif enrichment analysis, in an additive model to create networks of TFs and TGs. These networks are composed of nodes (genes, including TFs and target genes) and weighted edges (the interactions between TFs and TGs), where each interaction between a TF and a TG is assigned a score (Fig. 4A). ANANSE uses motif databases and orthology assignment to associate motifs to TFs (Supplementary Data 15, Supplementary Fig. 12A, for a detailed description, see Supplementary Note 3). We aggregated our scRNA-seq and scATAC-seq in pseudobulk data to isolate the gene expression and chromatin signal of every independent broad cell type, to generate a network for each, for a total of eleven networks of TF-target genes (Supplementary Fig. 12B). We pruned these networks for lowly-scored interactions and constructed TF-TG graphs which we used to calculate a centrality score

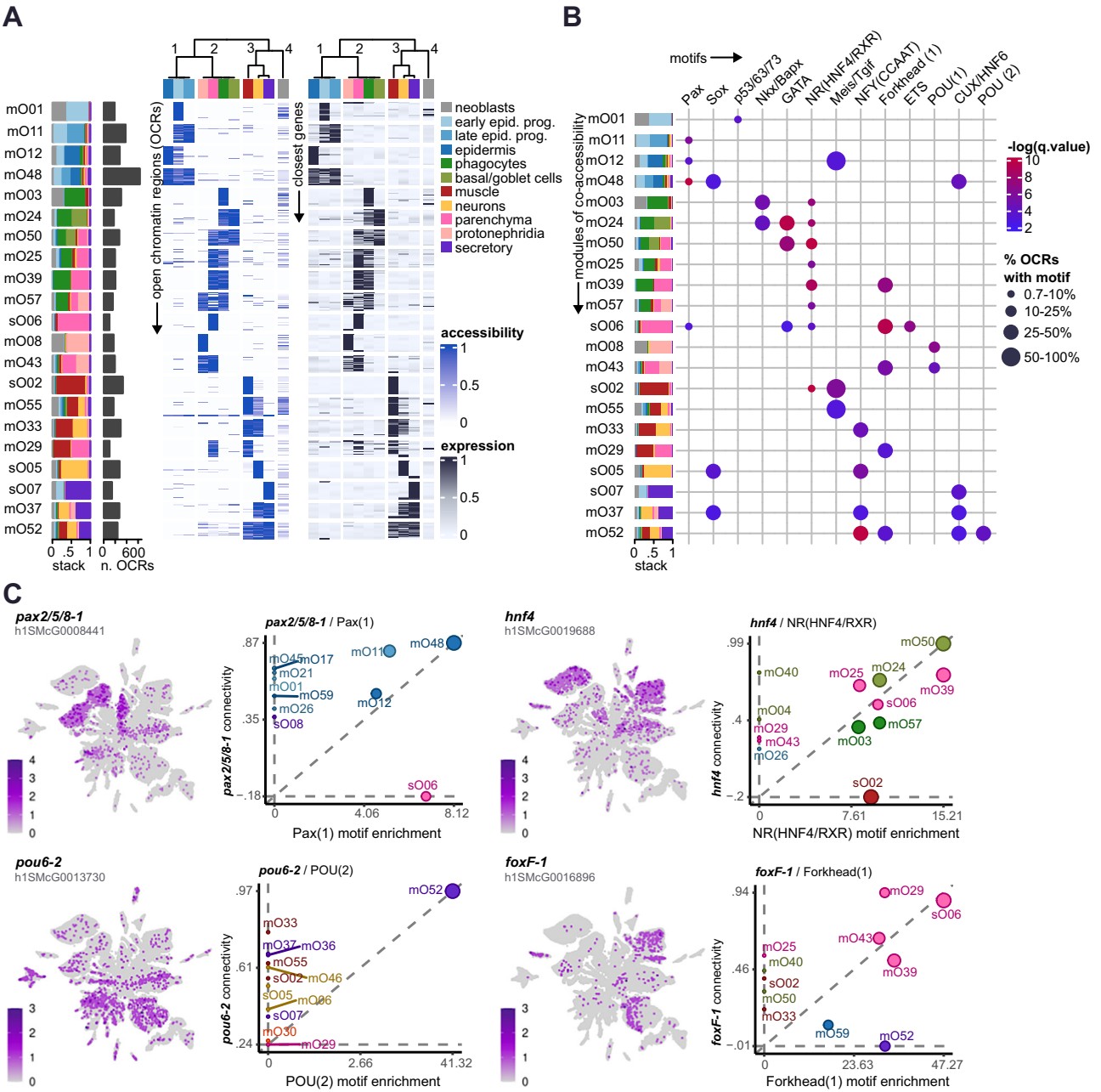

**Fig. 3 | Chromatin dynamics of pluripotent and differentiated planarian cells.** **A** (Left) Heatmap of scaled accessibility of open chromatin regions (OCRs) from several modules of co-accessibility. OCRs in rows, broad cell types in columns. A subsample of 20 OCRs per module is shown. Leftmost annotation: stacked bar plot of average co-accessibility across broad cell types, and bar plot of number of OCRs per module. (Right): Heatmap of gene expression of genes in the vicinity of the OCRs shown on the left heatmap. Genes in rows, broad cell types in columns. In both heatmaps, broad cell types (columns) have been arranged based on chromatin accessibility similarity. **B** Dot plot showing motif enrichment analysis on the OCRs from the modules of co-accessibility shown in (**A**). **C** UMAPs of gene expression and scatter plots showing the connectivity of the planarian TFs to modules of co-accessibility, in relation to the motif enrichment of the same TF class in OCRs from modules of co-accessibility. Dot size indicates significance of hypergeometric test (q-value < 0.1).

(Supplementary Note 3) for each TF in each network. Correlating these profiles of TF centrality for each network revealed that epidermal cell types clustered together, as expected, and that the most similar cell type to gut phagocytes were the parenchymal and basal/goblet cells (Supplementary Fig. 12C–E). This suggests that not only are their gene expression and chromatin accessibility patterns similar, but also their transcription factor-based regulation.

To investigate if these similarities go beyond the molecular signature of differentiated cell types, we used *ANANSE influence* to compare the cell fate networks from neoblasts to every differentiated cell type (Fig. 4A, B). *ANANSE influence* uses differential gene expression

(DGE) data to compare the two networks and identify the so-called influential factors: TFs whose expression changes the most and show highest binding to differentially expressed genes (DEGs) between two networks. Comparing a differentiated cell type network against the neoblasts network can predict key TFs driving the differentiation of pluripotent stem cells to that differentiated cell type. Thus, we performed DGE analysis on our scRNA-seq dataset comparing every broad cell type against neoblasts independently (Supplementary Fig. 13A, Supplementary Data 16). We used this data to generate the cell fate networks from neoblast to every major cell type and retrieved dozens of influential TFs for each cell fate alongside their target genes (Fig. 4B,

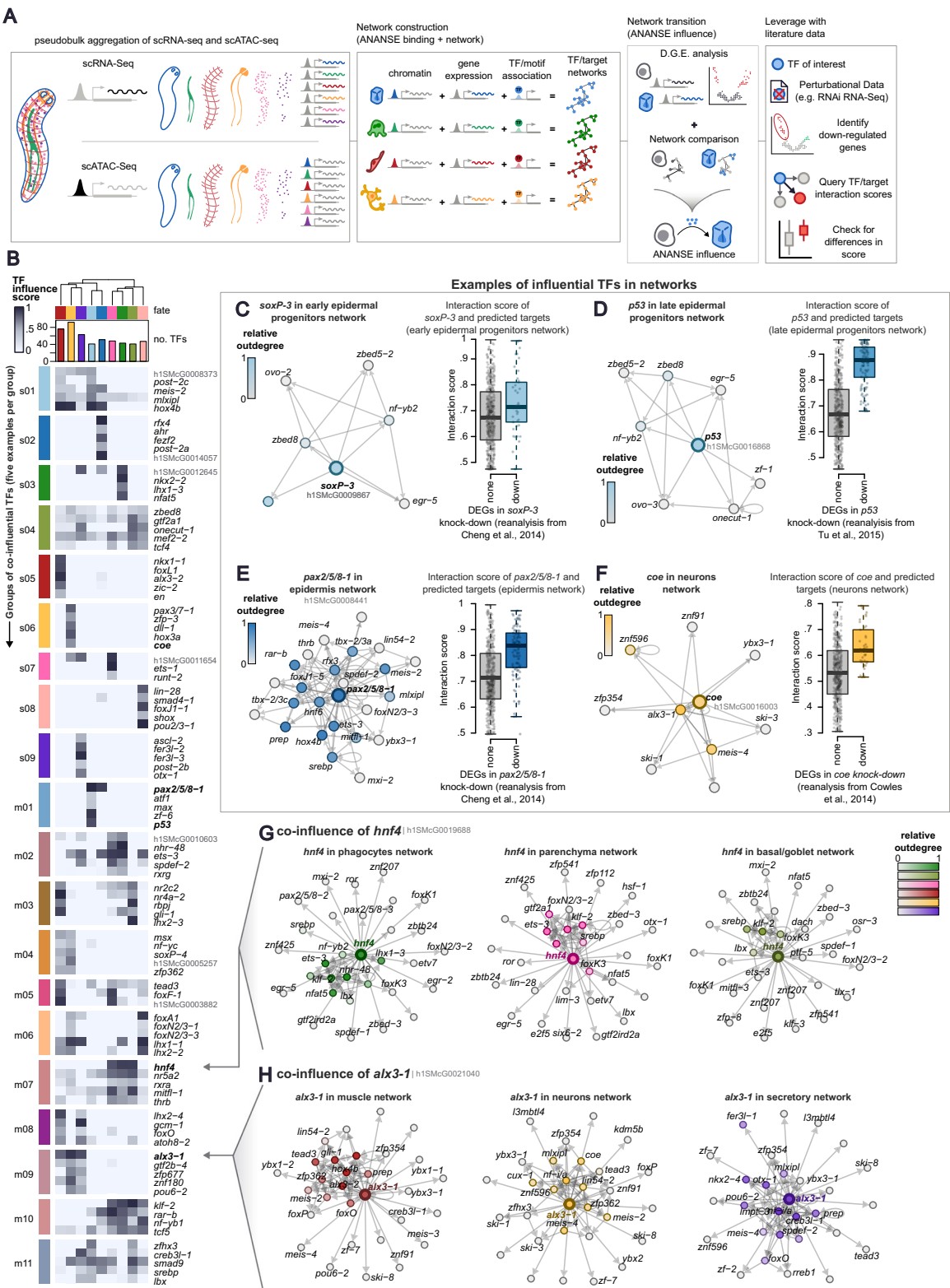

**Fig. 4 | TF/target gene regulatory networks of planarian cell differentiation.**
**A** Graphic summary of ANANSE. **B** Heatmap of top co-influential TFs for different modules of co-influential TFs. TFs in rows, fates in columns. Top tree: clustering of fates based on the co-influence profile of these TFs. **C**–**F** (Left) subgraphs and (right) boxplots showing the interaction score of predicted target genes for different influential TFs with knockdown functional data from the literature: **C** soxP-3 (N: 48,666 non-downregulated genes, 42 downregulated genes), **D** p53 (N: 48594 non-downregulated genes, 114 downregulated genes), **E** pax2/5/8-1 (N: 48,606 non-

downregulated genes, 102 downregulated genes), and **F** coe (N: 48,664 non-downregulated genes, 44 downregulated genes). Colour intensity of graph nodes indicate relative outdegree (number of emitting connections) in differential network. Centre line, median; box limits, upper and lower quartiles; whiskers, 1.5x interquartile range; points, data points. **G** subgraphs of hnf4 and neighbouring TFs in the (left) phagocytes, (middle) parenchyma, and (right) basal/goblet networks. **H** subgraphs of alx3-1 and neighbouring TFs in the (left) neurons, (middle) muscle, and (right) secretory networks.

for summary networks see Supplementary Figs. 14, and 13B–J, Supplementary Data 17, 18, 19). Our predicted influence networks recapitulate well-known TFs from the scientific literature, such as *nkx2–2* in phagocytes[87–89], or *pou2/3*-1 in protonephridia[81]. Some of the predicted targets also align with expectations extracted from the literature, such as *gata456*, predicted target of *nkx2-2* in phagocytes[73,76,87,89], or *ca-VII*, predicted target of *pou2/3-1* in protonephridia[80] (Supplementary Fig. 14). These observations lend support to our ANANSE analysis and suggest it can predict interactions between TFs and target genes.

Leveraging ANANSE with functional data from the literature (from RNAi knockdown of TFs, Fig. 4A) further highlighted the ability of ANANSE to successfully predict network interactions between TFs and target genes. For example, we identified *soxP-3*, *p53*, and *pax2/5/8-1* as influential in the epidermal lineage[75,82] (Fig. 4C–E), or *coe* in neuronal differentiation[90,91] (Fig. 4F), and their top predicted targets included genes downregulated in previously published knockdown experiments (Fig. 4C–F, right panels; Supplementary Figs. 15, and 16, Supplementary Data 20, 21, 22, 23, Supplementary Note 4). In addition, we orthogonally validated our ANANSE predictions by leveraging knockdown data from the anteriorly expressed *prep* transcription factor. Top predicted *prep* targets from our networks showed anterior expression[92,93] and include genes downregulated in functional knockdown experiments[94] (Supplementary Fig. 17, Supplementary Data 22–24, Supplementary Note 4). Interestingly, *prep* is influential in several cell types (epidermal, parenchymal and protonephridia, Supplementary Fig. 14), suggesting that its expression in several cell types specifies their anterior identities. Overall, these orthogonal validations lend support to ANANSE's power of detection.

Several transcription factors appear as influential for more than one fate (Fig. 4B), in agreement with previous knowledge. For instance, *foxF-1* was most influential in parenchyma and muscle, as recently described[74]. To further investigate these patterns, we clustered the influential TFs in groups of co-influence, detecting sets of TFs that share a similar profile of influence over one or multiple fates (Fig. 4B, Supplementary Fig. 13K, Supplementary Data 25, Supplementary Note 2; See "Methods"). For instance, module m05 contained TFs co-influential in muscle and parenchyma (Fig. 4B, Supplementary Fig. 13L), including *foxF-1*[74]. Our groups of co-influence included TFs influential in protonephridia/neurons (m06), muscle/secretory cells (m08), and neurons/muscle (m04). Group m07 was influential in parenchymal and gut cells, which included *hnf4* among the top five (Fig. 4G). Group m09 was influential in neuron, muscle, and secretory cells, and included *alx3-1*[95,96] among the top as well (Fig. 4H). Together with our previous analyses, this suggests that *hnf4* is an important TF to regulate the differentiation of neoblasts into cell types other than gut phagocytes, and that *alx3-1* might be an important regulator for neuronal, muscular, and secretory cell fates.

Overall, our results show that the graph-based multimodal integration model from ANANSE can elucidate the regulatory logic of planarian stem cell differentiation. Our results suggest that this similarity of networks underlies the differentiation process, achieved by a combinatory logic of broad, co-influential, and cell-type-specific TFs.

## Major groups of differentiated planarian cells

Throughout our analyses we observed that planarian differentiated cell types tended to group together in a consistent manner, based on gene expression, chromatin accessibility, and TF network dynamics (Figs. 2–4). Specifically, we observed a supergroup formed by neurons, muscle, and secretory cells, and another supergroup formed by parenchymal and gut (phagocytes, basal, and goblet) cells (Fig. 5A–C). Based on the co-influential profile of *hnf4* and *alx3-1*, we decided to call these supergroups *hnf4*+ and *alx3*+ groups. *hnf4* is a gut transcription factor with documented expression in parenchymal cells[44], albeit no functional roles have been reported. *alx3-1* is an aristaless-homeobox transcription factor with documented expression in neurons and

muscle cells[95], which we detect as expressed in neurons, muscle, and also secretory cells (Supplementary Fig. 18A). This suggests that *hnf4* and *alx3-1* are key regulators of these cell types.

Roles of *alx3-1* in neurons and muscle have been reported[95], however, our data predict that it is also a key regulator of secretory cells. To assess this role, we first investigated whether *alx3-1* target genes can be detected in secretory cells. We re-analysed *alx3-1* RNAi knockdown data from Akheralie et al.[95] and detected mostly gene downregulation (Fig. 5D, E, Supplementary Fig. 18B Supplementary Data 26). We detected expression of these genes in neurons and secretory cells, as shown by our gene scoring of cell types (Supplementary Fig. 18C). Interestingly, we found high interaction and weighted binding predicted scores between these genes and *alx3-1* in the muscle, neuron, and secretory networks (Fig. 5F, Supplementary Fig. 18D). Based on the agreement between downregulation, high interaction score, and co-expression in the same broad cell type, we identified gene h1SMcG0000140 as a candidate target gene of *alx3-1* in secretory cells (Fig. 5F) with reported expression in secretory cells[44]. To validate this prediction, we knocked down *alx3-1*, and evaluated h1SMcG0000140 expression by in situ hybridisation, revealing a significant decrease of h1SMcG0000140+ secretory cells in newly generated tissue in *alx3-1(RNAi)* animals (T-test, $p = 0.000068$, Fig. 5G, Supplementary Fig. 18E, Supplementary Data 27). This could be because secretory cells fail to express h1SMcG0000140 in the absence of *alx3-1*, indicating a role of *alx3-1* in their gene expression, or because *alx3-1* is needed for their maintenance. Together with previously reported roles in muscle and neurons[95], this previously unreported role of *alx3-1* in secretory cells shows that *alx3-1* has a role in these three planarian cell fates, underscoring the validity of ANANSE's prediction of influence in all three. Altogether, our results show that muscle, neuronal and secretory cell types likely share a common regulatory logic, with a role of *alx3-1* in all three of them.

## Single cell analysis of *hnf4* RNAi unveils gut and parenchymal defects

Our analyses suggest that phagocytes, parenchymal, and basal/goblet cells form a common supergroup that we called *hnf4*+ cells. *hnf4* was initially reported to be expressed in the gut[35,97] and their progenitors, the gamma neoblasts[43,88], and for its role in phagocyte differentiation[98,99]. Recent single-cell transcriptomic studies have also revealed a prominent expression in parenchymal cells[44,45]. Our results indicated that *hnf4*-related motifs are highly enriched in genes expressed in both phagocytes and parenchymal cell clusters, as well as in the OCRs of those cell types. Our ANANSE analysis predicted *hnf4* as a top influential factor for both gut cell types and parenchyma. These observations raise the question of whether *hnf4* is a regulator of parenchymal cells in addition to gut cells.

To test the validity of this computational inference, we performed *hnf4* RNAi experiments in two biological replicates. We examined their phenotype and then analysed the underlying molecular differences using single-cell transcriptomics (Fig. 6A). We reasoned that, contrary to ISH or qPCR, scRNA-seq could systematically measure all genes across computationally dissected broad types and could be directly compared with our cell-type-wise GRNs. Compared to control animals, *hnf4(RNAi)* worms from both replicates showed frequent depigmentation and necrotic lesions in the pre-pharyngeal area, but also in other body parts, by 9 days post injection (Supplementary Fig. 19). This area gradually accumulated damage, often resulting in cleaving or disintegration of the head by days 12–15 post injection. These animals did not regenerate and eventually died.

We used worms on day 9 post injection to generate a single-cell dataset of 41,016 cells using ACME and SPLiT-Seq. Automated cell cluster annotation using our reference atlas (Fig. 1) identified all major planarian cell types, and similar metrics as other SPLiT-Seq

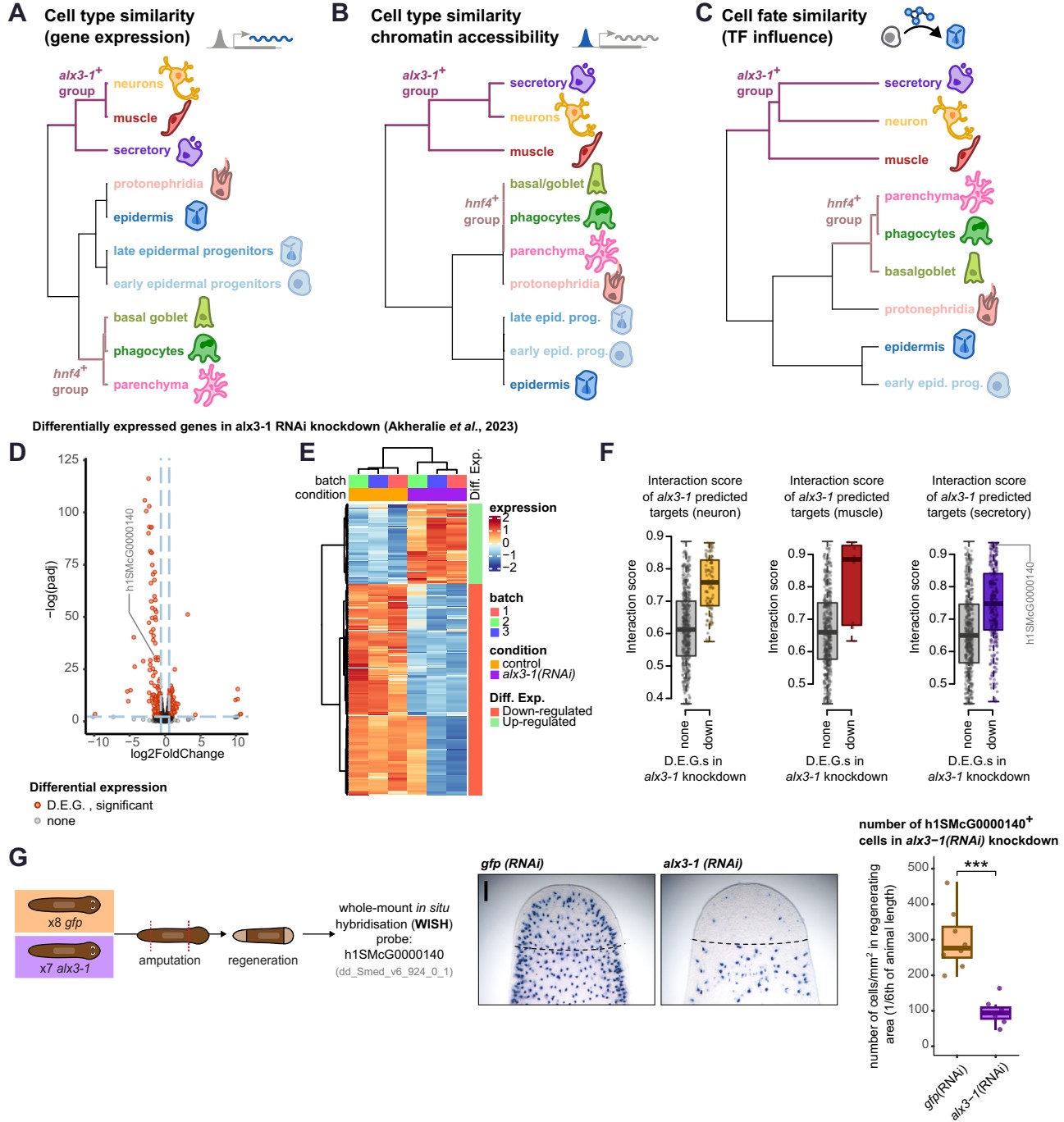

**Fig. 5 | A common regulatory logic that groups planarian cell types. A** Co-occurrence tree of broad cell types based on gene expression similarity. **B** Co-occurrence tree of broad cell types based on chromatin accessibility similarity. **C** Co-occurrence tree of broad cell types based on TF co-influence and graph centrality similarity. **D** Volcano plot showing the expression of differentially expressed genes in the *alx3-1(RNAi)* knockdown over gfp*(RNAi)* (intact animals); re-analysis of data from (Akheralie et al., 2023) (Wald Test, two-sided, multiple comparisons correction). Two outlier genes (log(padj) = 451; log(padj) = 229) not shown. Gene h1SMcG0000140 has been highlighted. **E** Heatmap showing expression of differentially expressed genes in the *alx3-1(RNAi)* knock-down and control animals, re-analysis of data from (Akheralie et al.[95]). **F** Box plots showing the predicted ANANSE interaction score between *alx3-1* and target genes on the networks of neurons (left) (N: 48,617 non-downregulated genes, 91 downregulated genes), muscle (centre)(N: 48,700 non-downregulated genes, 8 downregulated genes) and secretory (right)(N:

48,427 non-downregulated genes, 281 downregulated) cells, cell types with high gene scores for *alx3-1(RNAi)* down-regulated genes, and where *alx3-1* is expressed. Score of h1SMcG0000140 in the secretory network has been high-lighted. Centre line, median; box limits, upper and lower quartiles; whiskers, 1.5x interquartile range; points, data points. **G** whole-mount in situ hybridisation (WISH) of gene h1SMcG0000140 in control and *alx3-1* knock-down animals. Left: overview of the experimental design. Centre: whole-mount in situ hybridisation (WISH) showing h1SMcG0000140+ cells on planarian regenerating heads, for control (left) and *alx3-1* knockdown (right). Scale bar = 0.2 mm. Right: box plot of number of h1SMcG0000140+ cells per mm² of regenerating tissue in control and *alx3-1* knockdown animals. *N* = 16 animals (one animal lost during the protocol). *T*-test, one-sided (null hypothesis: greater), *p* = 0.000068, no adjustments for multiple comparisons. Centre line, median; box limits, upper and lower quartiles; whiskers, 1.5x interquartile range; points, data points.

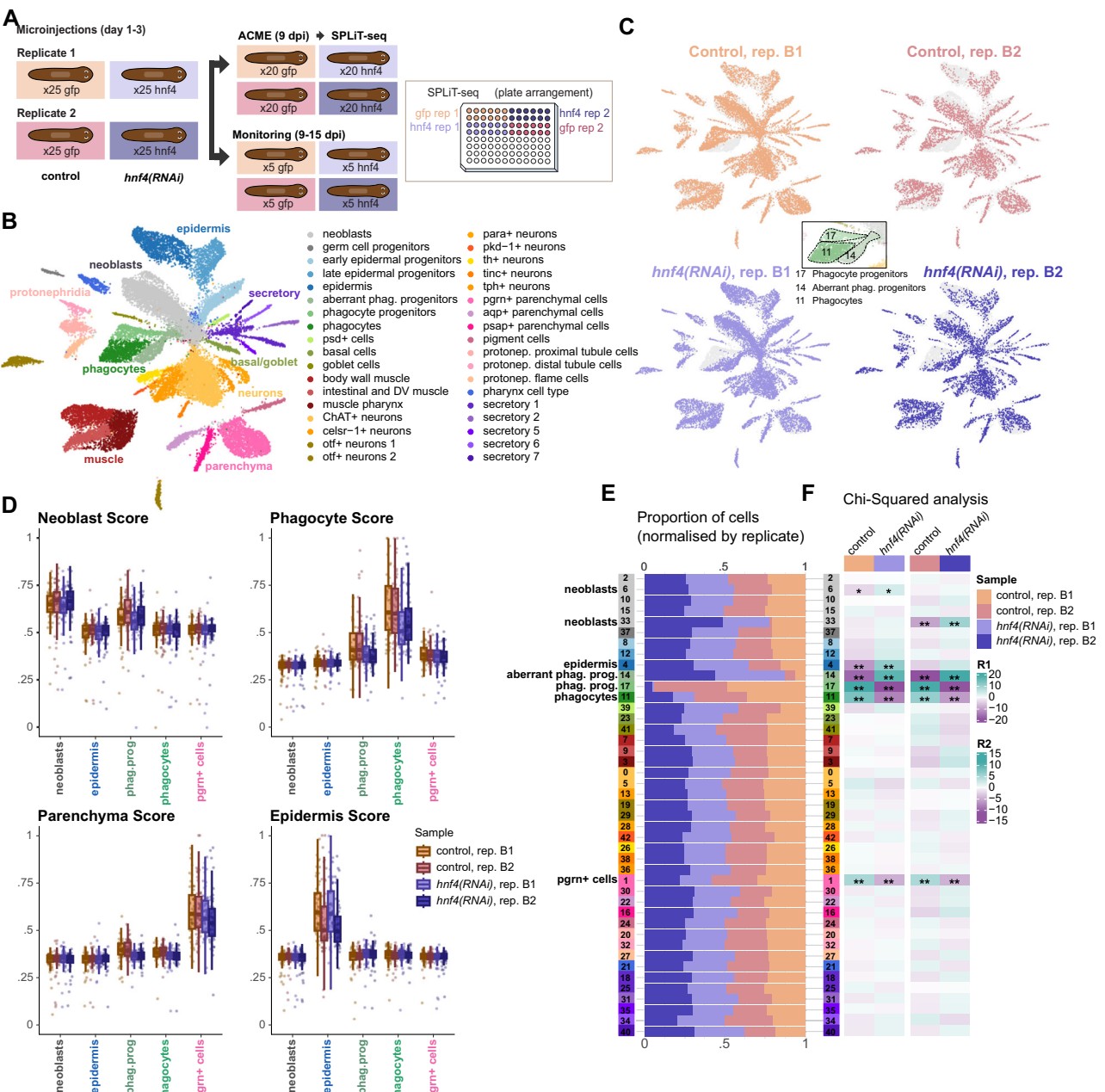

**Fig. 6 | Multiplex single-cell analysis of *hnf4* knock-down cells. A** Overview of the experimental design of the knock-down. **B** UMAP projection of 26,596 individual cell transcriptomes in the *hnf4*i dataset. **C** Distribution of cells from the different conditions and replicates across the UMAP projection. **D** Gene score of neoblasts (left) and phagocytes (right) markers in neoblasts, epidermis, phagocyte progenitors, phagocytes, and *pgrn+* parenchymal cells of each condition and replicate. Data points are individual cells. N: neoblast control 1 = 1565, neoblast control 2 = 541, neoblast *hnf4(RNAi)* 1 = 1842, neoblast *hnf4(RNAi)* 2 = 650, epidermis control 1 = 316, epidermis control 2 = 98, epidermis *hnf4(RNAi)* 2 = 152, phagocyte progenitors control 1 = 428, phagocyte progenitors control 2 = 183, phagocyte progenitors *hnf4(RNAi)* 1 = 501, phagocyte progenitors

*hnf4(RNAi)* 2 = 244, phagocytes control 1 = 484, phagocytes control 2 = 195, phagocytes *hnf4(RNAi)* 1 = 180, phagocytes *hnf4(RNAi)* 2 = 105, *pgrn+* cells control 1 = 735, *pgrn+* cells control 2 = 385, *pgrn+* cells *hnf4(RNAi)* 1 = 548, *pgrn+* cells *hnf4(RNAi)* 2 = 321. Centre line, median; box limits, upper and lower quartiles; whiskers, 1.5x interquartile range; points, outliers. **E** Fraction of cells from each condition and replicate on each Seurat cluster. Row colours indicate transferred cell identities. Name labels for cell clusters with significant differences in abundance (see **F**). **F** Heatmap of residuals from Chi-squared post-hoc analysis (two-sided, no adjustments for multiple comparisons) of cell abundances in each Seurat cluster. Asterisks represent level of significance; $p < 0.05$, $p < 0.01$.

experiments (Fig. 6B, Supplementary Fig. 20A–D, Supplementary Data 28). We observed that *hnf4* expression in the *hnf4(RNAi)* cells was substantial, indicating that despite strong effects, the knockdown did not translate into reduced mRNA levels (Supplementary Fig. 20E). In fact, in our reanalysis of TF RNAi data (*soxP-3, pax2/5/8-1,* and *prep*) we similarly failed to detect downregulation of the targeted transcription factor (Supplementary Note 4). This suggests that other factors,

including possible compensatory effects, detection of the dsRNA, or of cleaved mRNAs can hinder the measurement of the genes knocked down by RNAi. Single-cell data in general, and ACME dissociated cells in particular, are enriched for nuclear RNA[55], which could be limiting our ability to detect lower levels of cytoplasmic mRNA.

When analysing the knockdown scRNA-seq dataset, we observed that cluster 17 (phagocyte progenitors) consisted almost entirely of

*control(RNAi)* cells, while cluster 14 was mostly formed by *hnf4(RNAi)* cells (Fig. 6C). In our automated label transferring for cell type annotation, cluster 14 was the only cluster that received labels from two distinct broad types, namely neoblasts and phagocyte progenitors (Supplementary Fig. 20B). Thus, we termed cluster 14 as "aberrant phagocyte progenitors", and we proceeded to investigate the nature of these cells. One possibility is that they might fail to activate genes related to phagocyte biology. Alternatively, they might fail to deactivate genes related to stem cell biology. To differentiate between these scenarios, we scored the aggregated expression of neoblast and phagocyte markers in neoblasts, phagocyte progenitors, and phagocytes from each experimental group, including epidermis and *progranulin+* parenchymal cells and scores as controls (Supplementary Data 29). We found no differences between control and *hnf4(RNAi)* samples for the neoblast marker score, but the phagocyte score was significantly reduced in phagocytes and phagocyte progenitors in RNAi samples (Fig. 6D, upper-tail Wilcoxon test, Supplementary Fig. 20F). Interestingly, the parenchymal score was also reduced in the phagocyte progenitors. This indicated that *hnf4(RNAi)* phagocyte progenitors fail to activate phagocyte genes, and that genes expressed by parenchymal cells might also be affected in these aberrant progenitors.

When testing for differences in cell type abundance across treatments, we observed that both phagocyte progenitors and differentiated phagocytes are significantly reduced in *hnf4(RNAi)* samples of both replicates (Fig. 6E, F, Chi-Squared test, Supplementary Data 28). Importantly, the only other cluster significantly reduced in *hnf4(RNAi)* samples of both replicates was cluster 1, containing *progranulin* (*pgrn*)+ parenchymal cells. These are the major cell types where *hnf4* is expressed, lending support to the specificity and the effectiveness of our knockdown. All other clusters were not significantly affected except for clusters 4, 6, and 33. Cluster 4, the major epidermal cell cluster, was significantly increased in *hnf4* RNAi in one biological replicate, likely due to a composition bias induced by the sharp decrease in gut phagocyte and parenchymal cells. Clusters 6 and 33, containing neoblasts, were significantly increased in the knockdown condition of one biological replicate, and likely contain neoblasts early in the differentiation process towards aberrant phagocyte progenitors. Taken together, these analyses showed that *hnf4* RNAi led to decreased proportions of both gut phagocytes and parenchymal cells.

## Distinct *hnf4*-mediated gene regulation in phagocytes and parenchymal cells

Our multiplexed approach includes biological replicates, which are essential in bulk RNA-seq for identifying significantly regulated genes. Several researchers have highlighted that replicates are equally important for single-cell pseudobulk methods[47]. However, the high cost of such methods presents a challenge. Single-cell combinatorial barcoding techniques, like SPLiT-seq, offer a solution by allowing the multiplexing of multiple samples within a single experiment, thereby reducing costs and batch effects.

To analyse gene expression changes in each cell type, we aggregated gene expression counts based on cluster identity and sample origin. This allowed us to create pseudo-bulk count tables for each cell type and broad group separately, enabling computational dissection of each cell type within each sample. We then employed DESeq2[100] for DGE analysis, incorporating biological replicates for each cell type (Fig. 7A, Supplementary Data 30). To evaluate the response rate to the knockdown in each cell type, we analysed the relationship between cluster size and the number of DEGs detected in each broad cell group (Fig. 7B) and individual cell cluster (Supplementary Fig. 21A, B). This analysis showed that most DEGs are in phagocytes and parenchymal cells, consistent with the expression of *hnf4* in both (Fig. 7C).

We visualised DEGs in each broad cell type as volcano plots. These confirmed that most DEGs occurred in phagocytes and parenchymal cells (Fig. 7D). This lends further support to the effectiveness of our knockdown. Despite a high *hnf4* expression, basal-goblet cells had a relatively low number of DEGs, which could be explained by their lower numbers, or a lower turnover rate, or both. In fact, a relatively higher turnover rate of gut phagocytes would also explain the larger effects compared to parenchymal cells. The most highly significant DEGs in both corresponded to downregulated genes, consistent with a role of *hnf4* as a transcriptional activator. Many genes were significantly up- or down-regulated in both phagocytes and parenchymal cells (46, 10%), but many others were differentially regulated only in phagocytes (348, 75%) or parenchymal cells (67, 15%) (Fig. 7E). These gene sets are enriched in broadly different gene ontology terms (Supplementary Fig. 21C–F), suggesting they are functionally independent.

We then aimed to determine if DEGs detected in our in vivo knockdown data overlapped with in silico predicted *hnf4* target genes from our ANANSE analysis (Fig. 4). Importantly, these two analyses are entirely independent. To analyse this overlap, we examined the interaction score of DEGs in phagocytes, parenchymal cells, and those regulated in both, comparing these to the rest of the genes (Fig. 7F). On average, in vivo DEGs of all three groups had increased in silico predicted ANANSE interaction scores in both the phagocyte and parenchymal cell ANANSE networks. Altogether, these analyses showed that *hnf4* regulates independent but overlapping gene expression programs in both gut phagocytes and parenchymal cells. Moreover, modelling of the detection power of ANANSE by logistic regression returned positive coefficients of correlations both in the phagocytes and parenchymal network ($p$-val <0.05, Supplementary Fig. 21G). These results validate our in silico ANANSE analysis, and together with similar observations in data from the public literature (Fig. 4), lend support to our ANANSE predictions for all other TFs and interactions.

Finally, we questioned what other factors could be responsible for the differences between phagocytes and parenchymal cells. We analysed motifs enriched in all three DEG groups (Fig. 7G). Consistently, we found nuclear receptor HNF4 motifs in all three groups, indicating that the effects are due to an effective *hnf4* knockdown rather than off-target effects[101]. We also found motifs specific to phagocyte DEGs and parenchymal cell DEGs. Interestingly, a homeobox factor motif was highly enriched in phagocytes, and a Fox factor was highly enriched in parenchymal cells (Fig. 7G). We hypothesised that these could correspond to *nkx2-2* and *foxF-1* (Supplementary Fig. 21H), which have been shown to be important for phagocyte and parenchymal cell differentiation, respectively[74,87,88]. We examined the connectivity of these factors with WGCNA modules (Supplementary Fig. 21I). We observed *nkx2-2* had higher connectivity to phagocyte-only modules, and *foxF-1* to parenchymal cell modules. Altogether, these analyses corroborate that DEGs detected in vivo have *hnf4*-related motifs as well as motifs of other factors expressed in each of the two cell types and suggest that these factors may synergise with *hnf4* to regulate phagocyte- and parenchymal cell-specific expression.

## Combinatorial regulation of phagocytes and parenchymal cells
To further investigate the effects of *hnf4* in relation with *foxF-1* and *nkx2-2*, we performed single and double RNAi experiments in triplicates and assessed their phenotypes (Fig. 8A, Supplementary Note 5). We also performed a double *hnf4(RNAi)+gfp(RNAi)* double knockdown to control for the potential effect of co-injection. Animals from double TF knockdowns showed stronger phenotypes and slower survival rates across 20 days compared to *hnf4* RNAi, except for the double *hnf4(RNAi)+gfp(RNAi)*, which had attenuated effects (Fig. 8B, C, Supplementary Fig. 22A-B, Supplementary Data 31). This was also reflected in their transcriptome dynamics (Fig. 8D). DGE analysis revealed sets of overlapping DEGs across conditions. Comparing DEGs from *hnf4(RNAi)* animals with those from *hnf4(RNAi)+gfp(RNAi)* animals revealed genes downregulated in the single and double knockdown, which we termed low-dose-response, suggesting these are strong *hnf4*

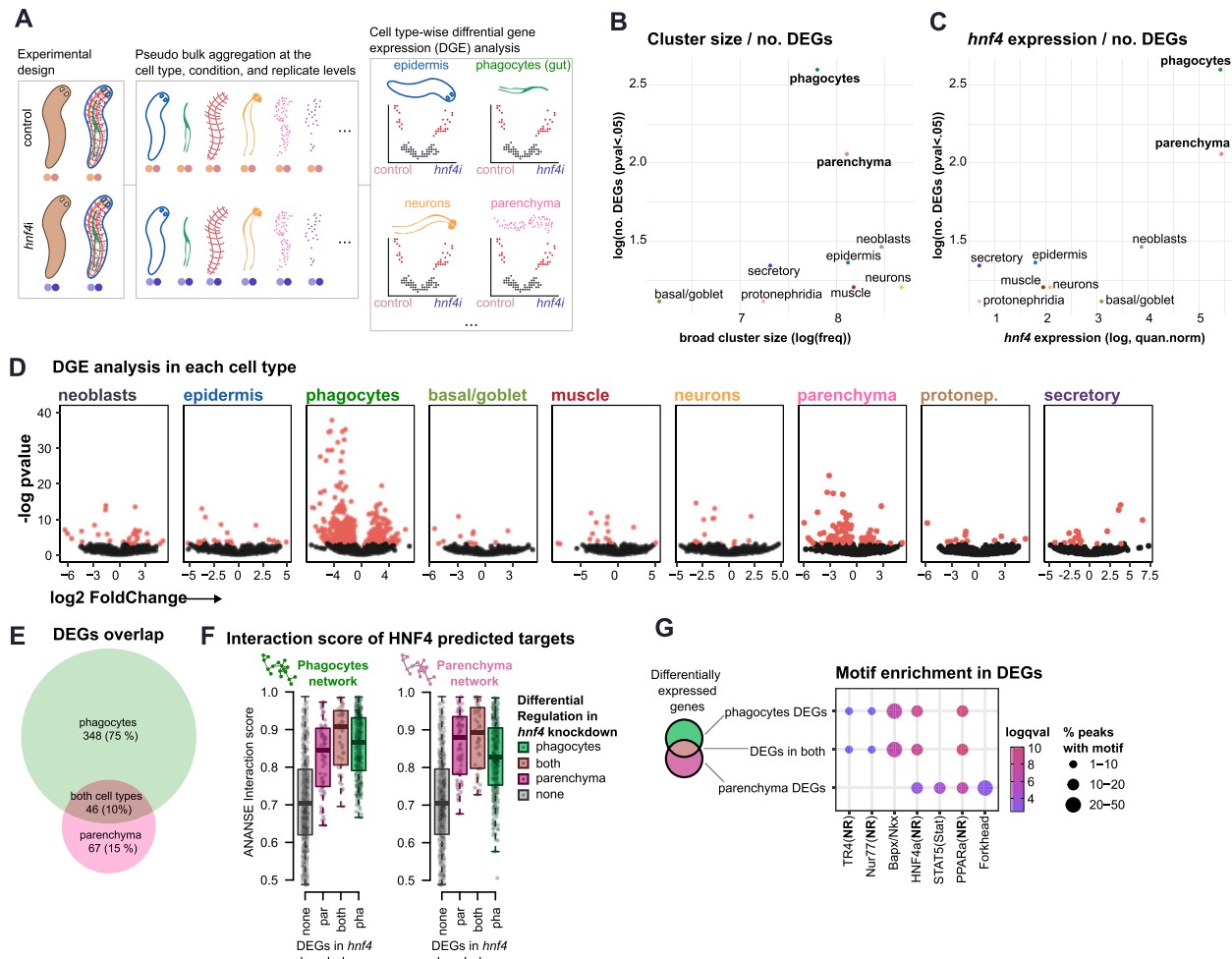

**Fig. 7 | Differential gene expression of *hnf4*i knock-down in different cell types.**
**A** Schematic of computational dissection of the scRNA-seq dataset to perform differential gene expression (DGE) analysis between control and *hnf4*i cells.
**B** Scatter plot showing the number of differentially expressed genes on each broad cell type (DEGs) in relation to the broad cell type cluster size (number of cells).
**C** Scatter plot showing the number of DEGs on each broad cell type in relation to the level of expression of *hnf4* on each broad cell type. **D** Volcano plots of DEGs on each cell type. *X* axis: log fold change of expression in *hnf4*i cells relative to control cells. *Y* axis: negative log *p*-value. Test for **B**–**D** Wald test, two-sided, no adjustments for multiple comparisons. **E** Venn diagram showing the overlap of DEGs in phagocyte and parenchymal cells. **F** Box plots showing the predicted ANANSE interaction score between *hnf4* and target genes. Centre line, median; box limits, upper and lower quartiles; whiskers, 1.5x interquartile range; points, data points. N: non-downregulated genes = 48,247, genes downregulated in parenchyma = 67, genes downregulated in phagocytes = 348, genes downregulated in both = 46. **G** Motif enrichment in DEGs based on the overlap between phagocytes and parenchymal cells.

targets as they respond to attenuated hnf4 inhibition and are enriched only in phagocytes and parenchymal cells (Supplementary Fig. 23A–C). Conversely, high-dose response genes were only downregulated in the single knockdown and seemed to be also enriched in other cell types where *hnf4* is not expressed, suggesting indirect effects (Supplementary Fig. 23C). Overall, despite the attenuated phenotypic effects, the double RNAi control treatment with *gfp* and *hnf4* led to specific effects.

We then compared DEGs between the single and the double TF knockdowns (Fig. 8E–G), together with the DEGs detected in computationally dissected phagocytes and parenchyma of our *hnf4(RNAi)* single cell analysis (Fig. 7C–E, Supplementary Data 32). Detailed exploration and scoring of these overlapping sets of DEGs revealed distinct patterns of expression across different cell types, as well as differences in specificity and sensitivity between our bulk and single-cell analyses (Supplementary Fig. 22C–U). The larger set of DEGs (356) was for *hnf4(RNAi)+nkx2-2(RNAi)*, consistent with the stronger phenotypic effects. DEGs shared between *hnf4(RNAi)* bulk and *hnf4(RNAi)+nkx2-2(RNAi)* and *hnf4(RNAi)* in computationally dissected phagocytes (74 and 60 genes) were mostly expressed in phagocytes and

contained the previously mentioned low-dose-response genes (Supplementary Figs. 23B-C, and 22V), revealing a high overlap between these treatments. Conversely, genes detected only in *hnf4(RNAi)* phagocytes (213 genes) had broader expression in other cell types, suggesting that they are specifically downregulated in phagocytes and can thus not be observed in bulk experiments, where differences across tissues average out (Supplementary Fig. 22I). A group of 89 DEGs only in *hnf4(RNAi)* whole animals displayed no enrichment in phagocytes and contained the largest number of high-dose-response genes, suggesting their downregulation results from indirect responses (Supplementary Figs. 22W, and 23B-C). These results indicate that the bulk double knockdown and the single-cell knockdown analyses achieve similar results, but the latter has higher specificity and detects less indirect effects. This results also show that *nkx2-2* and *hnf4* share many DEGs, suggesting that they synergise in their gene regulation activity.

We then focused on DEGs from the *hnf4(RNAi)+foxF-1(RNAi)* double knockdown. Collectively, they scored phagocytes and parenchymal cells higher. A group of 101 DEGs unique to this treatment only scored parenchymal cells (Fig. 8I, Supplementary Fig. 22J), and the

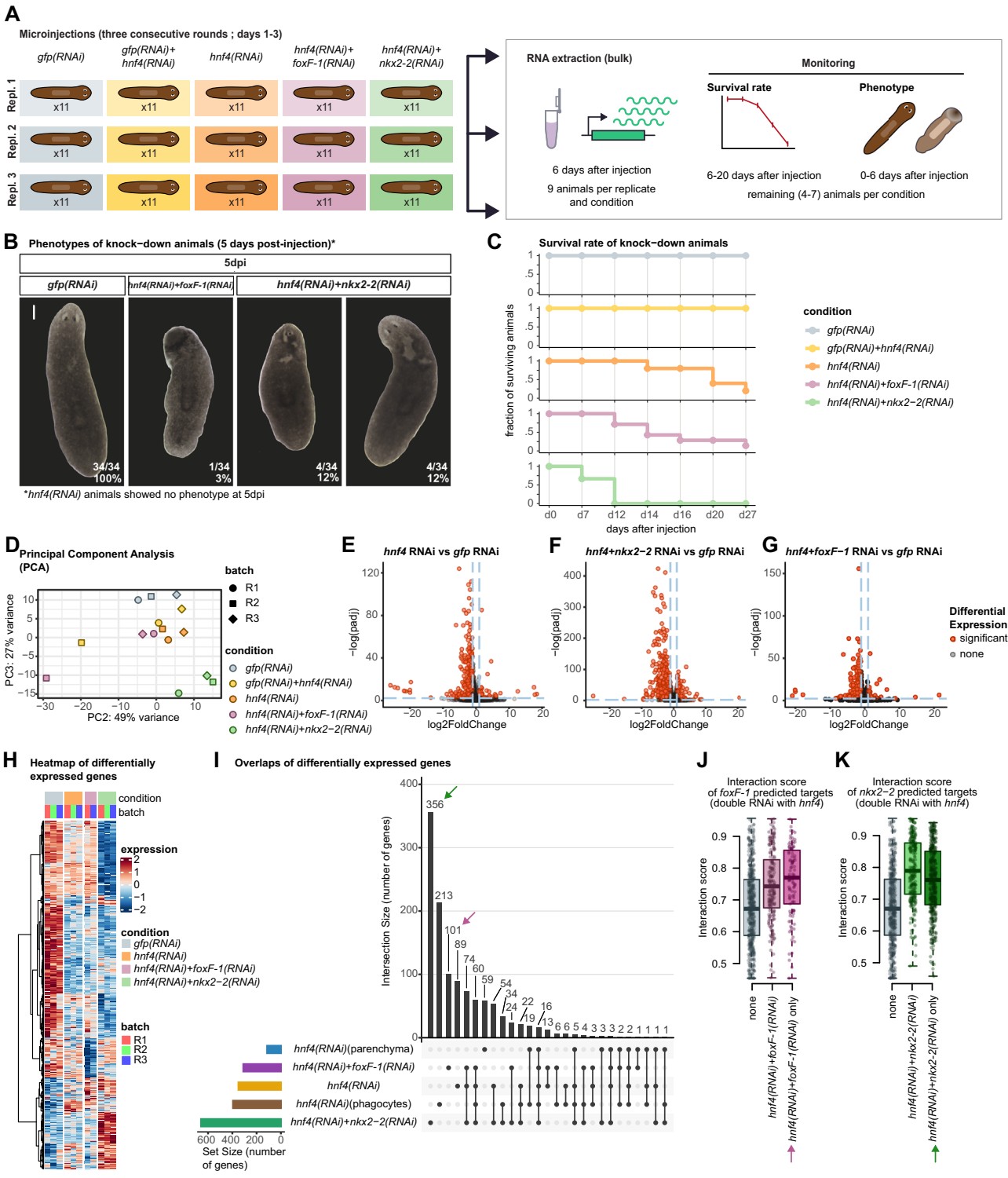

**Fig. 8 | Gene knockdown elucidates the regulatory logic of phagocytes and parenchymal cells. A** Overview of the experimental design of the knock-down. **B** phenotype of control and knock-down animals (one experiment, four to seven individuals per condition). Scale bar = 0.2 mm. **C** Survival rate curves of control and knock-down animals. **D** Principal Component Analysis (PCA) of the RNA-Seq samples. **E** Volcano plot of differentially expressed genes (DEGs) of *hnf4(RNAi)* knockdown animals. **F** Volcano plot of DEGs of *hnf4+nkx2-2(RNAi)* knockdown animals. **G** Volcano plot of DEGs of *hnf4+foxF-1(RNAi)* knockdown animals. Tests for **E–G** Wald Test, two-sided, multiple comparisons correction. **H** Heatmap of DEGs in all three knockdown experiments. Genes in rows. Samples (conditions and batches) in columns. **I** Up-set plot of intersections of DEG sets between the combined knockdown experiments and our single cell RNA-Seq knockdown of *hnf4*. Black bars

indicate number of genes on each set. *Y* axis indicates intersection size (number of genes). *X* axis indicate the different gene sets, corresponding to intersections (shown as presence/absence dot plots). Coloured bar plots indicate number of DEGs per experiment. Arrow indicates gene sets shown in (**J, K**). Box plots showing the predicted ANANSE interaction score and between **J** *foxF-1* and target genes in the parenchyma network (N: non-downregulated genes = 48,409, any down-regulated gene in *hnf4+foxF-1(RNAi)* = 197, genes downregulated only in *hnf4+foxF-1(RNAi)* = 102), and **K** *nkx2-2* and target genes in the phagocytes network (N: non-downregulated genes = 48,083, any downregulated gene in *hnf4+nkx2-2(RNAi)* = 259, genes downregulated only in *hnf4+nkx2-2(RNAi)* = 366). Centre line, median; box limits, upper and lower quartiles; whiskers, 1.5x interquartile range; points, data points.

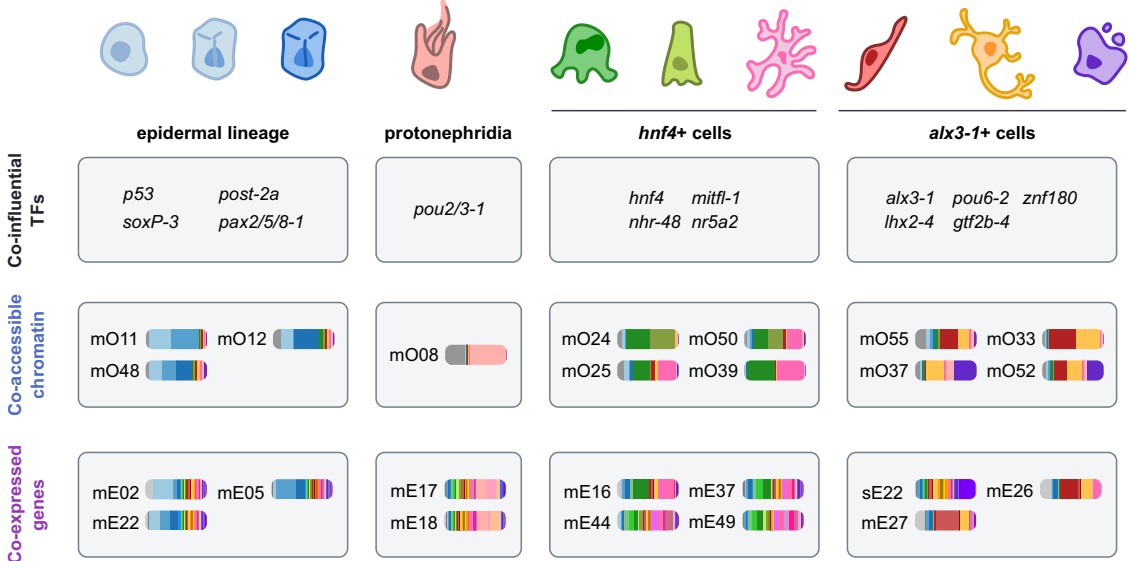

**Fig. 9 | Two supergroups of planarian differentiated cell types with a common regulatory logic.** Our analyses reveal that planarian differentiated cell types can be grouped in at least two supergroups: *hnf4*+ cells, and *alx3-1*+ cells. Our analyses of gene expression and open chromatin found commonalities between these cell types such as groups of co-influential Transcription Factors, modules of co-accessible regions of open chromatin, and modules of co-expressed genes.

remaining 208 DEGs were shared with other *hnf4* RNAi treatments. Only a few genes were shared with DEGs from *hnf4(RNAi)* parenchymal cells (24, including the groups of 16, 3, 2, 2, and 1, Fig. 8I), suggesting that contrary to *nkx2-2* targets, *foxF-1* DEGs are largely independent from *hnf4*. Interestingly, querying the interaction score of these gene sets in our networks revealed that genes downregulated only in the *hnf4(RNAi)+foxF-1(RNAi)* double knock-down had a higher interaction score with *foxF-1* than other DEGs (Fig. 8J), which was not true for genes downregulated only in *hnf4(RNAi)+nkx2-2(RNAi)* (Fig. 8K). This suggests that the independent response to the knockdown of a second TF is higher for *foxF-1* than for *nkx2-2*.

Taken together, our analyses suggest that *nkx2-2* likely partners with *hnf4* to drive differentiation of phagocytes, whereas *foxF-1* might share some downstream target genes with *hnf4* but drives parenchymal differentiation independently of *hnf4*. Overall, our computational inference of GRNs followed by single and double knockdown provides new insights and formulates new hypotheses about how genome regulation drives planarian stem cell differentiation.

## Discussion

GRNs arise from the interplay between TFs and OCRs, forming a regulatory logic that regulates gene expression. Classical perturbation experiments provide direct evidence of TF gene regulation, and single-cell technologies allow to model GRNs computationally. Combining both approaches can provide scalable tools to reconstruct GRNs widely across many species. In this study we used single cell technologies to infer GRNs underlying planarian stem cell differentiation. We generated gene expression and chromatin accessibility atlases and used these to infer GRNs and potential regulators of cell fate across multiple cell types, which show a common combinatorial logic in gene expression, chromatin accessibility, and TF regulation.

Our analyses reveal at least two supergroups of cell types with common regulatory logic: the *alx3-1*+ group (comprising muscle, neurons, and secretory cells), and the *hnf4*+ group (comprising phagocytes, basal/goblet, and parenchymal cells, Fig. 9). These groups are consistent with those reported by Chai and collaborators based on regulatory motif usage[102], showing their existence in two other platy-helminth species. Our data retrieves these supergroups from transcriptomic and OCR data. Our functional validation shows that *alx3-1* and *hnf4* play a role in these lineages. Besides those supergroups,

epidermal cells and their progenitors grouped consistently together, and protonephridia resembled epidermal cells in gene expression but was more similar to *hnf4*+ cells in our chromatin and TF influence analyses. This can be due to a lack of resolution, but also due to the heterogeneity of protonephridia, comprising tubule and flame cells, which could arise from different lineages.

The *alx3-1*+ and *hnf4*+ supergroups might arise from functional similarities, lineage relationships, hierarchical regulatory relationships, or other technical or biological factors. For instance, neuronal and secretory cells are functionally linked by their exocytic activity, and gut phagocytes and parenchymal cells are similarly linked by endocytic activity[74]. However, goblet cells are thought to have exocytic activity[87,97] but belong to the *hnf4*+ supergroup, therefore challenging the notion of functional similarities underlying supergroups of cells.

These supergroups largely align with the neoblasts subclasses previously proposed in the literature[66,88,103]. Of note, our grouping arises from differentiated cell data, as opposed to neoblast classes. This suggests that gamma neoblasts give rise to all *hnf4*+ cells, which share a regulatory code. Likewise, sigma neoblasts may be the progenitor to all *alx3-1*+ cells. Whether gamma and sigma neoblasts can be further subclustered in individual classes for broad types, or specific cell types remains a question. Recent works propose that neoblasts undergo specialisation in the G2 phase and frequently divide asymmetrically giving rise to an unspecialised neoblast that retains pluripotency and a specialised neoblast[104]. Recently, it has been shown that many of these specific cell type fates can already be identified in X1 neoblasts[71], suggesting that these lineage decisions may happen within the neoblast compartment.

Our data shows for the first time a previously undescribed role of *alx3-1* in regulating secretory cell fate, in addition to roles in muscle and neurons[95]. This poses the question of whether *alx3-1* might be a global initiation factor of the *alx3-1*+ cell identity. Another potential regulator is *pou6-2*, which we found expressed in neurons, muscle and secretory cells, and also had an enriched POU motif in the OCRs co-accessible in these very same cell types.

Similarly, we identified *hnf4* as a regulator of parenchymal cells, in addition to its described role in gut cells[98,99]. We detected previously undescribed changes in cell abundance and gene expression in both cell types. While we could not confirm a decrease in *hnf4* cytoplasmic

mRNA levels after RNAi, we observed knockdown effects in cell numbers and gene expression in the tissues that express *hnf4*, with enrichment of *hnf4* motifs, and in a pattern consistent with the downregulation of a transcription factor. All of these independent lines of evidence support the effectiveness of our knockdown and argue that the observed effects are direct effects caused by the knockdown rather that off target effects[101].

Gene expression and motif enrichment analyses suggest TFs *nkx2* and *foxF-1* are differentially specific to each of these types, consistent with previous studies[74,87,88]. To further investigate potential interactions between these factors, we performed double knockdown analyses, revealing substantial overlap between *hnf4* and *nkx2-2* responsive genes, and less overlap between *hnf4* and *foxF-1* responsive genes. One possible reason is that only the expression of *nkx2-2* is required to discriminate between phagocyte and parenchymal cell identity, and that *foxF-1* regulates parenchymal gene expression in a manner largely independent from *hnf4*. Future works might delve into whether *foxF-1* is required to drive neoblasts into parenchymal fate by controlling cell fate regulators or by regulating parenchymal effector genes. We envision that future studies will decode similar relationships in GRNs with functional links akin to logic gates[105].

Regarding neoblasts, we found the accessible regions of neoblast specific genes are open in most other cell types, resembling constitutive promoters. Similarly, accessibility enrichment around the TSS of neoblast marker genes was different from that of markers of other cell types. This aligns with previous results obtained by tissue fractionation[40], revealing that planarian neoblasts follow chromatin regulation rules distinct from those of their differentiated counterparts.

Our results were limited by the resolution of the scATAC-seq, which resolved the major broad cell types but did not detect subclusters (like, for example, neuronal subtypes). Combined with the high abundance of neuronal, epidermal, and muscle cells, the low resolution likely biased our sampling towards these broad cell types, resulting in more OCRs detected compared to less abundant cell types such as secretory or protonephridia cells. The resolution of the scATAC also limited our GRN reconstructions, as some TF/target scores were primarily driven by gene expression similarity and not by chromatin-related metrics such as weighted binding. Other sources of epigenetic information that contribute to regulatory logic, and that are not currently considered, include DNA methylation and epigenetic marks. Finally, our TF binding information derives from data generated in other species. Future studies will add de novo motif finding information and combine it with biochemical binding studies.

Overall, our data demonstrates that GRN computational inference can co-exist with classic functional approaches, as the former is able to formulate data-driven hypotheses of TF influence in cell fates. Candidate influential TFs are predictions and necessitate functional validation by perturbation assays coupled with single cell analysis. Future studies will incorporate a larger number of knockdowns, exploiting the scalability of combinatorial barcoding single cell transcriptomic methods. Our study pioneers this avenue, which will lead to decoding the GRNs underlying regeneration and development broadly across the animal tree of life.

## Methods
### Experimental batches
This study comprises data from 6 independent experiments: 1 scATAC-seq and 5 scRNA-seq (batches 1, 8, 11, 14 and 23). In addition, each scRNA-seq batch comprises multiple libraries (batch 1: libraries 1-3, batch 8: libraries 8.3 and 8.4, batch 11: libraries 11.3 and 11.4, batch 14: libraries 14.3 and 14.4, and batch 23: libraries 23.1 and 23.2) (Fig. 1A).

### Animal culture and collection
All libraries used in this study were generated from asexual *S. mediterranea* worms derived from the clonal line *Berlin-1*[106]. The animals

were kept at 18–20 °C in 1x Montjuic water (1.6 mM NaCl, 1.0 mM $CaCl_2$, 1.0 mM $MgSO_4$, 0.1 mM $MgCl$, 0.1 mM KCl, and 1.2 mM $NaHCO_3$, dissolved in deionised water) at pH 7.0. For experiments performed in Oxford (*hnf4* RNAi characterisation and multiplex single cell analysis), planarians were fed cow liver once or twice per week and starved 7 days minimum before any experimental procedure. Animal collection consisted of a random selection of mixed-size healthy individuals (1–10 mm), except for batches 11 and 23. For batch 11, we selected 6–8 mm animals, as is the standard size for dsRNA injection. Animal collection for batch 23 was performed as described in Emili et al.[57]. For experiments performed in Barcelona (double knockdowns bulk RNA-seq samples and WMISH samples), an asexual clonal line of *S. mediterranea* was maintained at 18–20 °C in 1x Montjuic water at pH 7.0. Planarians were fed once a week with organic veal liver and starved 7 days minimum before any experimental procedure. We selected animals around 6 mm in length.

### Knock-down by RNAi
Batch 11 was generated using knockdown samples treated with *gfp* (control) or *hnf4* dsRNA. Double knockdown RNA-seq samples were generated using knockdown samples treated with *gfp* (control), *hnf4, hnf4+gfp, hnf4+nkx2-2* or *hnf4+foxF-1* dsRNA. Whole-mount in situ hybridisation (WISH) experiments used knockdown animals treated with *gfp* (control), *alx3-1* dsRNA and a probe from h1SMcG0000140 gene. These samples were obtained according to the following protocol:

### Primary PCR (*hnf4* RNAi characterisation and multiplex single-cell analysis).
To amplify *hnf4*, we used cDNA from wild type *S. mediterranea* worms. To amplify *gfp*, we used a DNA miniprep (13 ng/uL) of enhanced GFP in a pAGW vector provided by the Drosophila Genomics Resource Center. Primary PCR was performed using 2 μL of cDNA/DNA miniprep, 2 μL of 10x Standard Taq Reaction Buffer (NEB), 0.4 μL of dNTPs (2.5 μM), 0.2 μL of Hot Start Taq DNA Polymerase (NEB), 4 μL of Primer Forward (2.5 μM), 4 μL of Primer Reverse (2.5 μM) and 7.4 μL of water. The primer sequences were as follows: ggccgcggCGCTGAAATAGCCAGTCACA (hnf4-F), gccccggccGCCGCTTCAGGTGATATGTT (hnf4-R), ggccgcggGTCTATATCATGGCCGACAAG (gfp-F) and gcccggccACTGGGTGCTCAGGTAGTGGT (gfp-R). *Hnf4* primers were designed from the GenBank sequence JF802199.1. Both primer pairs included linkers for Universal T7 primers: ggccgcgg (linker-F) and gccccggcc (linker-R). The thermocycler programme used was: 94 °C (30 s); 35 cycles at 94 °C (20 s), 55 °C (20 s) and 68 °C (30 s); and 68 °C (5 min). We assessed PCR products in a 1% agarose gel, cut the bands under UV light, and froze them in 50 μL of nuclease-free water at −20 °C.

### Primary PCR (double knockdowns bulk RNA-seq samples and WMISH samples).
To amplify *hnf4, nkx2-2, foxF-1, alx3-1* and *h1SMcG0000140* we used cDNA from wild-type *S. mediterranea* worms. Primary PCR was performed using 0.5 μL of cDNA, 3.25 μL of 10x Dream Taq Reaction Buffer, 0.5 μL of dNTPs (10 mM), 0.25 μL of Dream Taq DNA Polymerase, 1.25 μL of Primer Forward (2.5 μM), 1.25 μL of Primer Reverse (2.5 μM) and 18 μL of water. The primer sequences were as follows: ggccgcggCGCTGAAATAGCCAGTCACA (*hnf4*-F), gccccggccGCCGCTTCAGGTGATATGTT (*hnf4*-R), ggccgcggCCACTTACGTTTTGGTGCCA (*nkx2-2*-F), gccccggcTCGTCTTCTCTGTCAGCGTT (*nkx2-2*-R), ggccgcggCTGCAGTAAATGGCCAGGAA (*foxF*-1-F), gcccggcCGCATTTCCTTCTCTATGGTGT (*foxF*-1-R), ggccgcggTCAACTACAGGAGGCTTGCA (*alx3-1*-F), gccccggcGGAGGCTGTGTGACGAATTC (*alx3-1*-R), ggccgcggCAGATCTACGCGGATAAATGCA (*h1SMcG0000140*-F), gccccggcGTTTCTCACCGACATAATTGCC (*h1SMcG0000140*-R). *hnf4* primers were designed from the GenBank sequence JF802199.1, *nkx2-2* primers were designed from the PlanMine sequence dd_Smed_v6_2716_0_1, *foxF-1* primers were designed from the PlanMine sequence dd_Smed_v6_6910_0_1, *alx3-1* primers were designed

from the PlanMine sequence dd_Smed_v6_11150_0_1, *h1SMcG0000140* primers were designed from the PlanMine sequence dd_Smed_v6_924_0_1. For all genes, primer pairs included linkers for Universal T7/SP6 primers: ggccgcgg (linker-F) and gccccggc (linker-R). The thermocycler programme used was: 95 °C (30 s); 35 cycles at 95 °C (30 s), 57 °C (30 s) and 72 °C (60 s); and 72 °C (5 min). We assessed 3 µL/sample of PCR products in a 1% agarose gel and kept the rest for secondary PCR.

**Secondary PCR (*hnf4* RNAi characterisation and multiplex single-cell analysis).** Samples were thawed and centrifuged at maximum speed for 1 min to extract cDNA from the gel bands. The supernatants were collected and used as cDNA input for the secondary PCR. We prepared 100 µL reactions using 3 µL of cDNA, 2 µL of dNTPs (2.5 µM), 10 µL of 10x Standard Taq Reaction Buffer, 1 µL of Hot Start Taq DNA Polymerase, 82 µL of water, 1 µL of Universal T7-F5′ primer (25 µM, gagaattctaatacgactcactataggccgcgg), and 1 µL of Universal T7-R3′ primer (25 µM, agggatcctaatacgactcactataggccccggc). Samples ran in a thermocycler as follows: 94 °C (30 s); 5 cycles at 94 °C (20 s), 50 °C (20 s) and 68 °C (30 s); then 35 cycles at 94 °C (20 s), 65 °C (20 s) and 68 °C (30 s); and 68 °C (5 min). The size of the bands was assessed by running 10 µL/sample in a 1% agarose gel. The remaining volume was purified by 0.75x (*hnf4*) or 1.6x (*gfp*) SPRI size selection (KAPA Pure Beads, Roche) according to the manufacturer's protocol. Purified samples were eluted in 20 µL of nuclease-free water.

**Secondary PCR (double knockdowns bulk RNA-seq samples and WMISH samples).** We prepared 100 µL reactions using 2 µL of primary PCR, 13 µL of 10x Dream Taq Reaction Buffer, 2 µL of dNTPs (10 mM), 1 µL of Dream Taq DNA Polymerase, 5 µL of Universal T7-F5′ primer (2.5 µM, GAGAATTCTAATACGACTCACTATAGGGCCGCGG) and 5 µL of Universal T7-R3′ primer (2.5 µM, AGGGATCCTAATACGACTCACTATAGGCCCCGGC) or 5 µL of Universal SP6-R3′ primer (2.5 µM, AGGGATCGATTTAGGTGACACTATAGGGCCCCGGC) and 72 µL of water. Samples ran in a thermocycler as follows: 95 °C (30 s); 35 cycles at 95 °C (30 s), 57 °C (30 s) and 72 °C (60 s); and 72 °C (5 min). The size of the bands was assessed by running 3 µL/sample in a 1% agarose gel. The remaining volume was purified by QIAquick commercial kit. Purified samples were eluted in 30 µL of nuclease-free water.

**dsRNA synthesis (*hnf4* RNAi characterisation and multiplex single-cell analysis).** For each sample, we mixed 1 µg of purified cDNA, 12.5 µL of 2x Express Buffer (T7 RiboMAX, Promega), 2.5 µL of Express Mix (T7 RiboMAX, Promega), and up to 25 µL of nuclease-free water, and incubated for 4 h at 37 °C. Then, we added 2.5 µL of DNase (1 U/µL, T7 RiboMAX, Promega) and incubated for another 30 min at 37 °C. After incubation, reactions were stopped with 375 µL of Stop Solution (1 M NH4OAc, 10 mM EDTA, 0.2% SDS). The resulting dsRNA was purified using phenol:chloroform. We added 1 µL of GlycoBlue and 400 µL of acid phenol:chloroform (pH 4.5, Thermo Fisher) per reaction and vortexed thoroughly. We centrifuged for 5 min and collected the aqueous top phase into a new tube. We added 400 µL of chloroform, centrifuged for 5 min, and collected the top phase again. To precipitate pellets, we added 1 mL of cold ethanol, vortexed and centrifuged for 15 min. Pellets were washed in 1 mL of 70% ethanol and centrifuged for 10 min. We discarded supernatants and let the pellets dry for 5 min at 37 °C. Pellets were resuspended in 10–20 µL of nuclease-free water. All centrifugations were performed at 4 °C and maximum speed. As a quality check, we ran 0.5 µL of purified dsRNA in a 1% agarose gel. Finally, we measured the concentration in a Nanodrop. water.

**dsRNA synthesis (double knockdowns bulk RNA-seq samples and WMISH samples).** For each sample, we mixed 1 µg of purified cDNA, 8 µL of 5x Transcription Buffer (Thermo Scientific), 4 µL of dNTPs (25 mM), 2 µL of RNase Inhibitor (Applied Biosystems by Thermo

Fisher Scientific), 4 µL of T7 RNA Polymerase (Thermo Scientific) and up to 40 µL of nuclease-free water and incubated for 4 h at 37 °C. Then, we added 1 µL of DNase (1 U/µL, Thermo Scientific) and incubated for another 30 min at 37 °C. After incubation, reactions were stopped with 360 µL of Stop Solution (1 M NH4OAc, 10 mM EDTA, 0.2% SDS). The resulting dsRNA was purified using phenol:chloroform. We added 1 µL of Glycogen and 400 µL of acid phenol:chloroform (pH 4.5, Thermo Fisher) per reaction and vortexed thoroughly. We centrifuged for 10 min and collected the aqueous top phase into a new tube. We added 400 µL of chloroform, centrifuged for 10 min, and collected the top phase again. We incubated this phase for 20 min at 68 °C and for 45 min at 37 °C for proper annealing. To precipitate pellets, we added 0.5 µL of Glycogen and 1 mL of cold ethanol, vortexed and centrifuged for 20 min. Pellets were washed in 200 µL of 70% ethanol and centrifuged for 10 min. We discarded supernatants and let the pellets dry for 5 min at 37 °C. Pellets were resuspended in 12 µL of nuclease-free water. All centrifugations were performed at 4 °C and maximum speed. As a quality check, we ran 0.5 µL of purified dsRNA in a 1% agarose gel. Finally, we measured the concentration in a Nanodrop and diluted each dsRNA at a final concentration of 2000 ng/µL. To generate double dsRNA we mixed 1:1 proportions of each dsRNA, to achieve a final concentration of 1000 ng/µL. For single dsRNA we diluted at a final concentration of 1000 ng/µL.

**Injections and harvest.** For injections, we used worms that were 6–8 mm in length. Each animal was injected with 0.1 µg of dsRNA for 3 consecutive days (0.3 µg in total). For *hnf4* RNAi characterisation and single-cell experiments, we generated two replicates per condition (*gfp* and *hnf4*) that were both biological and technical, as they were processed by different researchers. We injected 25 animals per replicate with *hnf4* dsRNA and 25 animals per replicate with *gfp* dsRNA, using a Nanoject II Auto-Nanoliter Injector (Drummond Scientific Company). At 9 days post injection, counted from the last day of injections, we harvested 20 animals per replicate and condition and dissociated them using ACME (as described below). The remaining 5 animals were kept uncut and monitored from day 9 to 15 post injection. For double knockdown experiments, we injected single *gfp* (control) and *hnf4* dsRNA and double *hnf4+gfp, hnf4+nkx2-2, hnf4+foxF-1* dsRNA, with 11 animals per replicate and condition. At 6 days post injection, counted from the last day of injections, we generated three replicates per condition, with 9 animals per replicate and condition and dissociated them using TRIzol™ (as described below). The remaining animals (4–7 per condition) were kept uncut and monitored from day 6 to 20 post injection. For WISH experiments we injected single *gfp* (control) and *alx3*-1 dsRNA using a Nanoject II Auto-Nanoliter Injector (Drummond Scientific Company), in two rounds of injection, with 12 animals per condition. One day after the second round of injection, planarians underwent pre-pharyngeal and post-pharyngeal amputation to induce anterior and posterior regeneration. At 12 days of regeneration, we fixed 8 animals per condition to proceed with WISH experiments (see below), and the remaining 4 were left alive for observation.

**RNA sample preparation of double knockdown animals**
Total RNA was isolated using 500 µL of TRIzol™ Reagent. Tissue samples were homogenized using a tissulator and incubated at room temperature for at least 5 min to permit the complete dissociation of nucleoprotein complexes. The resulting dissociation was centrifuged for 10 min and the resulting supernatant was transferred to a new tube. Then, we added 100 µL of chloroform, vigorously shaked tubes, incubated for 3 min at RT, and centrifuged for 15 min. The colourless upper aqueous phase was transferred to a fresh tube. To precipitate the RNA, we added 250 µL of 2-propanol, incubated for 10 min at RT, centrifuged for 10 min and kept the pellet. Pellets were washed in 500 µL of 75% ethanol and centrifuged at 7500 rcf for 5 min. We discarded supernatants and let the pellets dry for 5 min. Pellets were

resuspended in 20 µL of nuclease-free water. All centrifugations except the last one were performed at 4 °C and 12,000 rcf. As a quality check, we ran 1 µL of purified RNA in a 1% agarose gel. Finally, we measured the concentration in a Nanodrop.

## Whole mount in situ hybridization
Colorimetric WISH was performed as previously described[107]. Animals were euthanized by immersion in 5% N-acetyl-L-cysteine (5 min), fixed with 4% formaldehyde (15 min) and permeabilized with Reduction Solution (10 min). Riboprobe *h1SMcG0000140* was synthesized using a DIG RNA labelling kit (Sp6/T7, Roche). Animals were mounted in 80% glycerol before imaging.

## Microscopy, image processing and quantification
Live animals were photographed with an sCM EX-3 high end digital microscope camera (DC.3000s, Visual Inspection Technology). Fixed and stained animals were observed with a Leica MZ16F stereomicroscope and imaged with a ProgRes C3 camera (Jenoptik, Jena, TH, Germany). Image processing was performed with Adobe Photoshop 2024. Quantification of the *h1SMcG0000140*-positive cells (figure X) was carried out manually in the regenerated region (calculated as one sixth of the animal length) and normalised by the area of this region using ImageJ-win64. *T*-test, as well as a Shapiro–Wilk test for normality, were performed using R v4.0.3.

## RNA-Seq analysis of double knockdown samples
Reads were mapped to the latest version of the *S. mediterranea* genome using kallisto. Counts were imported to R using the tximport package and analysed using DESeq2. We ran a PCA and discarded replicates 2 for the *foxF-1(RNAi)* and the *gfp(RNAi)+hnf4(RNAi)* as they did not group with any of the other samples in the first three components of the PCA. After filtering for lowly expressed genes, each knockdown condition was compared against the condition *gfp(RNAi)*, and the lists of DEGs were pooled and compared using the upsetR package[108]. Gene scores for each set of DEGs were calculated and visualised as described above. ANANSE interaction scores were visualised as described before.

## ACME dissociation
Tissue dissociation and fixation were performed using ACME as described in ref.[55] with the following modifications: Incubation time was 35 min for batch 11, and 45 min for batches 14 and 23. After incubation, all samples were pipetted up and down as in the original protocol. Then, batches 11, 14 and 23 were kept on ACME solution (on ice) for 3 consecutive rounds of filtration to help remove cell aggregates and undissociated tissue fragments. Samples were first filtered through 50 µL and 30–40 µL strainers (Celltrics). Then, they were centrifuged at 1000 × *g* for 5 min (4 °C) to reduce the volume of the solution to 1–2 mL; by discarding part of the supernatant and resuspending the pellet in the remaining volume. Samples were then filtered into 15 mL tubes using 1 mL filter tips (Flowmi). To remove ACME solution and wash cells, we added 7–8 mL of buffer (1x phospate-buffered saline (PBS) 1% bovine serum albumin (BSA)) to the filtered samples and centrifuged at 1000 × *g* for 5 min (4 °C). The supernatant was discarded, samples were resuspended in 900 µL of buffer and filtered one last time, using a 40 µm strainer, into 1.5 mL tubes. We added 100 µL of DMSO per sample and froze them at −80 °C.

## SPLiT-Seq
Batch 1 was entirely processed using the SPLiT-seq protocol described in García-Castro et al.[55]. Batches 8, 11, 14 and 23 were processed using the modifications introduced in Leite et al.[109] with the following variations:

## Sample preparation.
Frozen dissociated cells (unsorted) were thawed, centrifuged twice at 1000 × *g* for 5 min (4 °C), resuspending in ~450 µL of buffer (1× PBS 1% BSA), and filtered through a 50 µL strainer (CellTrics). For each sample, we stained a 1:3 aliquot (50 µL sample +100 µL buffer) for 20 min at RT (in the dark) using 1.5 µL of DRAQ5 (0.5 mM stock) and 0.6 µL of Concanavalin-A conjugated with Alexa-Fluor 488 (1 mg/mL stock, Invitrogen). The remaining cells were kept at 4 °C. To estimate cell concentration (singlets/µL), three measurements of 10 µL were taken for each aliquot by flow cytometry. To count singlet events, we used an FSC-H vs FSC-A gate to select singlets, a Concanavalin-A positive gate to select cells with cytoplasm and a DRAQ5 positive gate to select cells vs cellular debris. The remaining unstained cells were diluted according to this singlet cell count, in 0.5x PBS, to a final working concentration of ~625k cells/mL (5000 singlets per well).

## Plate loading.
Batch 11 comprises four different samples: *gfp* RNAi replicate 1, *hnf4* RNAi replicate 1, *gfp* RNAi replicate 2 and *hnf4* RNAi replicate 2. Each of these samples was loaded separately into specific wells (12 wells per sample) during round 1 of barcoding, so they could be deconvoluted during the bioinformatic analysis.

## FACS.
For each SPLiT-seq experiment, we sorted two separated libraries obtaining the following numbers: 19k cells (library 8.3), 24k cells (library 8.4), 17k cells (library 11.3), 18.5k cells (library 11.4), 6.1k cells (library 14.3), 6.6k cells (library 14.4), 19k cells (library 23.1) and 19k cells (library 23.2).

## PCR amplification.
Samples were amplified for 10-11 qPCR cycles.

## scATAC-seq library preparation
Nuclei suspensions for scATAC-seq were obtained from trypsin dissociated cells. Essentially, we chopped planarians into small pieces on ice and incubated the pieces in 2–4 ml of PBS containing 1% BSA and 1% Trypsin for 25–30 min at room temperature, gently pipetting up and down until fragments were completely dissociated. Cells were then pelleted at 1000 × *g* for 5 min at 4 °C and resuspended in n 4–5 ml of PBS containing 1% BSA. We filtered the cells through a 40 µm cell strainer (Becton-Dickinson) and through a 20 µm nylon net filter (Millipore). We pelleted cells at 1000 × *g* for 5 min at 4 °C and added 100 µl of chilled lysis buffer (10 mM Tris-HCl (pH 7.4), 10 mM NaCl, 3 mM MgCl$_2$, 0.1% Tween-20, 0.1% Nonidet P40 Substitute, 0.01% Digitonin, 1% BSA in Nuclease-free Water) to the cell pellet and mixed it by pipetting up and down 10 times. We incubated the mixture on ice for 3–5 min and added 1 ml of chilled Wash Buffer (10 mM Tris-HCl pH 7.4, 10 mM NaCl, 3 mM MgCl$_2$, 1% BSA, 0.1% Tween-20 in Nuclease-free Water) to the lysed cells and mixed it by pipetting up and down 5 times. We centrifuged the mixture at 500 rcf for 5 min at 4 °C and removed the supernatant carefully without disturbing the nuclei pellet. Nuclei were then resuspended in Diluted Nuclei Buffer (10X Genomics) and counted before injection in the 10X Genomics Chromium, following the manufacturer's protocol. The libraries were then amplified using Nextera library prep and sequenced in a NextSeq Illumina sequencer to obtain 75PE reads.

## Bulk ATAC-Seq library preparation
To generate nuclei suspension for bulk ATAC-seq, we flash frozen planarians for 1 min in liquid nitrogen and resuspended in cold lysis buffer (10 mM Tris-HCl (pH 7.4), 10 mM NaCl, 3 mM MgCl$_2$, 0.1% Nonidet P40 Substitute, in Nuclease-free Water). While in lysis buffer on ice, planarians were dissociated by smashing them against a 100 µm cell strainer with the aid of a syringe plunger. The resulting nuclei suspension was centrifuged (500 rcf for 5 min at 4 °C) and resuspended in PBS 0.04% BSA. A fraction (50 µl) of the nuclei suspension was labelled with DRAQ5 (1.5 µl of 0.5 mM stock), counted with a cytometer (as in the Sample

preparation section of the SPLiT-Seq above), and the volume to have 20k nuclei was calculated. The resulting volume was centrifuged and resuspended in 50 µl of tagmentation buffer (5 mM MgCl2, 10 mM Tris HCl, 9.4% Dimethylformamide in Nuclease-free water), 2.5 µL of custom loaded Tn5 (5′ TCGTCGGCAGCGTCAGATGTGTATAAGAGACAG and 5′ GTCTCGTGGGCTCGGAGATGTGTATAAGAGACAG) was added to the nuclei suspension and incubated at 37 °C for 30 min. After tagmentation, we resuspended nuclei in stop reaction mix (20 mM EDTA, 0.5 mM Spermidine, in nuclease-free water) and incubated at 37 °C for 15 min. We proceeded with the library preparation by purifying the tagmentation product using the Monarch PCR DNA Cleanup Kit (New England Biolabs) and following manufacturer's instructions. The tagmentation product was eluted in 10 µl of Nuclease-free Water, and the DNA concentration was measured using the NanoDrop. To evaluate the optimal number of amplification cycles, we first ran a qPCR using 1 µl of tagmented DNA, mixed with 5 µl of 2× Kapa HiFi HotStart ReadyMix (Roche), 0.5 µl of PCR_PF (25 µM, 5′-AATGATACGGCGACCACCGA-GATCTACACAATCCGCGTCGTCGGCAGCGTCAGATGTGTAT), 0.5 µl of PCR_PR (25 µM, 5′-CAAGCAGAAGACGGCATACGAGATTCATTAGGGTC TCGTGGGCTCGGAGATGTG), 3 µL of nuclease-free water and 0.5 µL of EvaGreen® Dye (20X in Water, Biotium). We then amplified 1ul of tagmented DNA using the same mix of the qPCR without EvaGreen, with the following conditions: 30 s at 98 °C; 10 s at 98 °C, 30 s at 65 °C, and 60 s at 72 °C repeated for 11 cycles; final elongation for 5 min at 72 °C. Finally, we purified the PCR products, quantified it with NanoDrop, and check the tagmentation profile using an Agilent 2100 bioanalyzer. The library was sequenced in a NovaSeq X Plus PE150 Illumina sequencer.

### Gene functional annotation

**Prediction of protein sequences.** We first extracted the coding sequences from the latest version of the *S. mediterranea*[60] using AGAT 'agat_convert_sp_gff2gtf.pl' (https://doi.org/10.5281/zenodo.3552717) and standard parameters. The resulting set of coding sequences was then transformed to protein sequence using TransDecoder v5.5.0 (https://github.com/TransDecoder/TransDecoder). first, we ran 'TransDecoder.LongOrfs' with standard parameters; second, we ran hmmscan vs Pfam database and BLAST vs Swissprot database, with parameters:'-max_target_seqs 1 -evalue 1e-5' and default parameters respectively, to gather supporting evidence for coding transcripts; third, we ran 'TransDecoder.Predict' with parameters '--retain_pfam_-hits pfam.domtblout --retain_blastp_hits blastp.outfmt6 --single_bes-t_only'. We manually curated and removed gene and transcript features with few or no hits against any known protein in our annotation databases (see below) that were overlapping with other features that did have hits against such databases.

**Querying against previous annotations.** The resulting set of predicted protein sequences (hereafter referred to as proteome) was queried against three previous genome annotations of *S. mediterranea* –Dresden v4, Dresden v6[110,111], and the extended annotation used in Garcia Castro et al.[55], to retrieve reciprocal best hits. Briefly, we ran BLASTp against each set of predicted protein sequences of each genome annotation version, using standard parameters. Secondly, genes without a clear one-to-one reciprocal match were queried more leniently against the previous versions; we retrieved all hits with *e*-value <0.001.

**eggNOG functional annotation.** The resulting proteome of *S. mediterranea* was queried using EggNOG mapper41 with the parameters: '-m diamond --sensmode sensitive --target_orthologs all --go_evidence non-electronic' against the EggNOG metazoa database. From the EggNOG output, GO term, functional category COG, and gene name association files were generated using custom bash code.

**Transcription factor annotation.** The resulting proteome of *S. mediterranea* was queried for evidence of Transcription Factor (TF) homology using (i) InterProScan[112] against the Pfam[113], PANTHER[114], and (ii) SUPERFAMILY[115,116] domain databases with standard parameters, (iii) using BLAST reciprocal best hits[117] against Swiss-Prot transcription factors[118], and (iv) using OrthoFinder[119] with standard parameters against a set of model organisms (Human, Zebrafish, Mouse, *Drosophila*) with well annotated transcription factor databases (following AnimalTFDB v4.0[120]). For the latter, a given *S. mediterranea* gene was counted as TF if at least another TF gene from any of the species belonged to the same orthogroup as the *S. mediterranea* gene. The different sources of evidence were pooled together, and we kept those *S. mediterranea* genes with at least two independent sources of TF evidence. In addition to this, we also added to our list all the *S. mediterranea* genes with a match of high query coverage and *e*-value < 0.001 against any of the TFs reported in Neiro et al.[39]. The resulting list of 665 genes was manually curated to assign a TF class to each gene based on their sources of evidence.

**Transcription Factor motif annotation.** The resulting proteome of *S. mediterranea* was queried for TF motif annotation using gim-memotifs motif2factors, using the proteomes and the JASPAR 2020 TF annotation of *Homo sapiens* and *Mus musculus* as reference, as well as the proteomes of several protostome metazoan species (Supplementary Data 15, Supplementary Fig. 12A) to provide additional phylogenetic signal for the automated transfer. Secondly, we ran the JASPAR similarity prediction tool[121] on those TFs retrieved with our sequence homology annotation that did not get any transferred motif, using the JASPAR 2024 motif database which overlaps with the JASPAR 2020 database. We manually curated this motif annotation by adding any TF with an associated motif by Neiro et al.[39]. The resulting list of 401 (out of 665) TFs was subsequently used for running ANANSE.

**Definition of promoters.** We defined promoters as gene regions ranging 200 bp upstream the TSS of genes and 200 bp of the TSS using 'bedtools flank'[122] with standard parameters.

### Single cell transcriptomic analysis

**Gene annotation pre-processing.** We parsed the genome annotation GFF3 file of *S. mediterranea* and converted it to Gene Transfer Format (GTF) using AGAT 'agat_convert_sp_gff2gtf.pl' (https://doi.org/10.5281/zenodo.3552717), after which we used a custom python script to add the gene_id, transcript_id, gene_name and transcript_name fields. This was done in order to comply with the requirements of dropseqtools and the SPLiT-Seq pipeline workflow (https://github.com/RebekkaWegmann/splitseq_toolbox) which envelops algorithms from Drop-seq_tools-2.3.0 (https://github.com/broadinstitute/Drop-seq, hereafter dropseqtools; see below).

**SPLiT-seq read processing.** Single cell RNA-seq libraries were pre-processed as previously described[55,57,63]. A total of 170,173,041 reads were sequenced. These were assayed for QC purposes using FastQC (https://www.bioinformatics.babraham.ac.uk/projects/fastqc). We concatenated shallow and deep sequencing using cat. We used CutAdapt v2.8[123] to trim read 1 (transcripts) and read 2 (UMI and barcodes) sequences, using standard parameters. Reads were checked to be in phase using grep, and the resulting phased reads were paired using pairfq makepairs (https://github.com/sestaton/Pairfq). The resulting reads were transformed into sam format using picard FastqToSam and mapped against the *S. mediterranea* genome (with a tailored annotation for dropseqtools; see above) using the SPLiT-Seq pipeline workflow wrapper that uses Picard (http://broadinstitute.github.io/picard/), STAR[124], and dropseqtools.

**Generation of scRNA matrices.** For each library, the resulting outputs of running the SPLiT-Seq pipeline (tag_bam_with_gene_function) were used to generate a gene x cell sparse matrix keeping cells with a minimum of 100 genes detected. When required, such as libraries 1–3 (from an experimental design containing *Dugesia japonica* cells) and 11.3 and 11.4 (with *hnf4*i cells), any cells not coming from control *S. mediterranea* organisms were discarded by re-running the command 'DigitalExpression' from dropseqtools using a set of white-listed cell barcodes corresponding to *S. mediterranea* cells (libraries 1-3) or *S. mediterranea gfp*i control cells (libraries 11.3 and 11.4).

**Matrix concatenations.** The resulting matrices were concatenated using Seurat v4.3.0[62] in R v4.0.3[125]. Briefly, we created independent Seurat Objects for each separate library and performed Proportional Fitting-log1p-Proportional Fitting normalisation[126] on each separate matrix. We labelled cells from each library with their respective library ID (Supplementary Data 1) and merged all the Seurat objects into a single concatenated sparse matrix.

**Seurat analysis.** The concatenated sparse matrix was queried for highly variable genes using 'SelectIntegrationFeatures()' from the Seurat package. We scaled the data and performed PCA using the function 'runPCA()' with the following parameters: 'npcs = 120'. We integrated the data using the function 'runHarmony()' from Harmony[127] and the following parameters: 'dims.use = 1:120, theta = 3, lambda = 3, nclust = 40, max.iter.harmony = 20'. We identified neighbours for the k-nn graph using the function 'FindNeighbors()' and the following parameters = 'dims = 1:120, k.param = 35'. We identified clusters using the function 'FindClusters()' with the Louvain algorithm[128] and the following parameters: 'resolution = 2, algorithm = 1, random.seed = 75'. We computed an UMAP projection using the function 'RunUMAP()' and the following parameters: 'dims = 1:120, reduction = "harmony", n.neighbours = 35, min.dist = 0.5, spread = 1, metric = "euclidean", seed.use = 1'. To find cell markers, we used the function 'FindAllMarkers()' with the following parameters: 'only.pos = TRUE, return.thresh = 1, logfc.threshold = 0', and subsequently sorted these markers based on average logFC to keep the top 30 markers per gene cluster.

**Alignment to reference dataset.** We first ported the AnnData object from our previous study on allometry of cell types Emili et al.[57] (hereafter the Sizes object) to Seurat using custom Python and R scripts. After transferring and transforming into a Seurat object, we ran the function 'FindTransferAnchors()' from Seurat using our Seurat analysis object (see section above) as query and the Sizes object as the reference, and the following parameters: 'dims = 120'. The predicted labels of each cell in the query dataset were added as a metadata column. We assigned the most frequent predicted label to each Seurat cluster we obtained and manually curated ambiguous assignments checking diagnostic markers from the reference dataset. Label transferring was visualised using igraph[129].

**Pseudobulk computational dissection and normalisation (without conditions or replicates).** We aggregated the counts of every gene in each cluster using a custom R function, which yielded a gene × cluster matrix of expression. In parallel to this, we created a gene × cluster matrix that quantifies how many cells from each cluster are expressing a gene, using a custom R function with the parameter 'min_counts = 1'. To adjust for expression dynamics across clusters from very different sizes (in terms of number of cells), we calculated a "cell weight" matrix that leverages the expression matrix and the cell number matrix to calculate, for every gene in every cluster, a "score" of expression constraint using the formula: $w_{ij} = 1 - exp(-C_{ij}) = 1 - exp(-a_{ij}/b_{ij})$; where $w_{ij}$ is the cell weight of gene i in cluster j, $a_{ij}$ is the fraction of cells from cluster j expressing gene i, and $b_{ij}$ is the fraction of cells NOT from cluster j expressing gene i. This weighs down the expression of genes with counts scattered among a low fraction of cells in large clusters as opposed to smaller clusters that comparatively capture less reads but more frequently inside that cluster. We ran this code inside a custom R function with the parameters 'min_counts = 30' for genes with at least 30 counts through the dataset, and the parameter 'min_cells = 3' to retrieve expression information for genes expressed in at least three cells of a given cluster of interest'.

We normalised the matrix of expression by "library size" using the 'DESeqDataSetFromMatrix()' function with parameter 'design = ~ condition' and the 'counts()' function with parameter 'normalised = TRUE' from the R package DESeq2[100]. This normalised expression matrix was weighed using the cell weights described above. The resulting cell-weighted normalised expression matrix was used for downstream analyses.

**Co-occurrency analysis.** Co-occurrency analysis was performed as previously described[63,130]. Briefly, the pseudobulk normalised matrix was subjected to Pearson correlation at the cell type level using bootstrapping and subsampling, using the function 'treeFromEnsembleClustering()' with the following parameters: 'h = c(0.75,0.9), p = 0.05, n = 1000, bootstrap = FALSE, clustering_algorithm = "hclust", clustering_method = "average", cor_method = "pearson"', and using all the genes in the pseudobulk matrix as 'vargenes'.

**WGCNA analysis.** We filtered our normalised expression matrix to keep genes with CV > 1.25 and scaled it across rows, and subjected this to the WGCNA algorithm[70]. We picked soft power beta = 8 as it resulted in the highest fit to a scale-free topology model[69]. The resulting adjacency matrix was weighted using topology overlap and the resulting Topological Overlap Matrix (TOM) was clustered using the function 'hclust()' with the following parameters: 'method = "average"'. The resulting gene tree was cut into different modules of co-expression using the 'cutreeDynamic()' function from the WGCNA package[70] with the following parameters: 'deepSplit = 3, pamRespectsDendro = FALSE, minClusterSize = 50'. We reclassified the resulting modules in specific or mixed modules based, for each module, on the number of outliers (value > (1.5 × standard deviation) + mean) on the distribution of upper quartiles of expression in cell clusters of the genes in that module ("s" if only one cluster clearly showcased a higher expression of the genes of that module, "m" if two or more clusters showcased higher expression of the genes of that module). In addition to this, we sorted these modules based on the identity of their clusters of peak expression. To create profiles of the relative amount of gene expression of each module, we calculated the average expression profile per module, and then we did an internal normalisation by dividing every count on each cluster by the sum of counts on all clusters. The resulting frequencies of expression in each cluster represent the average expression signal in each cluster, and these were visualised as stacked bar plots using the ComplexHeatmap R package[131]. To calculate cell type specificity, we calculated the Tau metric as previously described[67]. We implemented Yanai et al.'s formula in a custom R function and applied this to all genes using our pseudobulk, normalised gene expression matrix. Values were visualised using the ComplexHeatmap R package. We subsampled each module to randomly retrieve thirty genes and visualised their expression profiles using the ComplexHeatmap R package[131].

**Transcription factor expression analysis.** We filtered the pseudobulk normalised expression matrix to keep only the 665 genes classified as TFs by our previous analysis (see above). We correlated the expression of these TFs against the average expression profile of every WGCNA module (so-called connectivity score in WGCNA). The resulting expression and connectivity matrices were visualised using the ComplexHeatmap R package.

**WGCNA motif enrichment analysis.** We retrieved the promoters of the gene from each WGCNA module and performed motif enrichment analysis using the findMotifsGenome.pl wrapper from the HOMER suite[132] with the following parameters: '-p 12 -mis 3 -mset vertebrates', and using as background a subsample of the promoters of every *S. mediterranea* gene that did not belong to the queried module. We concatenated the motif enrichment results into a single matrix which we filtered to keep only motifs with $q$-value < 0.1 ($q$-value is defined as the minimum false discovery rate at which an observed enrichment is significant; thus, it is an adjusted $p$-value corrected for false discovery). Next, we manually curated the association of significant motifs and high-connectivity TFs by inspecting, for a given module X, if the sequence logos of the motifs enriched in promoters of module X resembled those of any group of TFs highly connected to module X. For this, we contrasted the HOMER sequence logos to the JASPAR 2024 database[133]. These results were visualised using the ggplot2[134] and ComplexHeatmap R packages[131]. To correlate TF connectivity and motif enrichment, we correlated the motif enrichment profiles of each significant motif with the TF connectivity of each TF to the same modules. This correlation was used as a proxy to further explore agreement between TF connectivity and motif enrichment. We visualised agreement between TF connectivity and motif enrichment via scatter plots using the ggplot2 R package.

**WGCNA graph analysis.** We generated a large graph taking the TOM matrix as an input adjacency matrix using the R package igraph[129]. We pruned lowly-scored interactions and used a threshold that maximised the number of evenly-sized connected components. We calculated cross-connections between modules by constructing a gene x module matrix used to count how many genes from each module are direct neighbours to a given gene x. We normalised this matrix by dividing the number of connections of gene x to each module by the size of the module that gene x is part of. These numbers were later aggregated at the module level to retrieve the number of normalised cross-connections between modules. The resulting matrix was transformed into a graph using 'graph_from_adjacency_matrix()' from igraph with parameters 'mode = "upper", weighted = TRUE, diag = FALSE', and the number of cross-connections was used for edge size to highlight the largest amounts of cross-connections.

In addition to this, we created three more module-wise graphs. We first correlated the motif enrichment profile of each module (specifically, the percentage of regions -in this case, gene promoters- with enrichment of each significant detected motif) and used this as an adjacency matrix for a module-wise motif enrichment graph. We then performed functional category enrichment of the genes of every module to obtain a module x functional category enrichment matrix using a custom wrapper of Fisher's test. We correlated the functional category enrichment profile of each module (specifically, the percentages of enrichment) and used this as an adjacency matrix for a module-wise functional category graph. Lastly, we correlated the TF connectivity profile of each module and used this as an adjacency matrix for a module-wise TF connectivity similarity graph.

We merged these four graphs using igraph and retained pairwise module connections detected in at least two of our four analyses to generate a module-wise similarity graph. We detected communities of modules using the function 'cluster_label_prop()' and the list of instances of co-occurring module-module edges as weights for the parameter 'weights'.

**Gene Ontology Enrichment analysis.** Gene Ontology Enrichment analysis was run using a custom wrapper of the R package 'topGO'[135], using all the genes with detected expression in our pseudobulk expression matrix (see above) and the classicFisher test with elim algorithm. Gene Ontology terms with less than five annotated genes in the whole genome of *S. mediterranea* (see eggNOG functional

annotation above) were discarded. These results were visualised using the ggplot2[134].

**Pseudobulk computational dissection of scRNA (with conditions and/or replicates).** We created a pseudo-bulk supermatrix leveraging the cell annotation data derived from our scRNA-seq analysis (e.g. clustering and broad cell types) along with additional cell information like sample characteristics (e.g. experiment, condition, replicates). This was achieved by aggregating, for a given gene X, cell type Y, experiment I, and replicate J, all the counts of gene X from cells belonging to the same cluster Y and under identical conditions (experiment I and replicate J). This process effectively generated a supermatrix with genes in rows and 'pseudo-samples' (combinations of cell type, experiment, and replicate) in columns. These pseudo-samples encompassed various combinations of cell types and conditions, such as RNA interference treatment and replicates (e.g. biological, technical, library etc.).

**Differential gene expression analysis (one-versus-neoblasts).** We performed DGE analysis to compare differentiated cell types against neoblasts as follows: For a given contrast (i.e. comparison of cell type X vs neoblast), we first extracted the relevant pseudo-samples from the pseudo-bulk supermatrix. Secondly, because we were comparing cell types, we used "cell type" as conditions and we used the different batches of experiments (libraries 1-3, libraries 8.3 and 8.4, libraries 11.3 and 11.4, libraries 14.3 and 14.4, and libraries 23.1 and 23.2) as replicates. Third, we ran DESeq2[100] inside a custom wrapper with a contrast of "condition 1" (cell type X) relative to "condition 2" (neoblasts). Genes were identified as differentially expressed if having a p-adjusted below 0.05 (negative binomial test).

## Single cell ATAC-seq analysis

**Generation of scATAC matrices.** The sc-ATACseq library was mapped using CellRanger[61]. We created a reference index for CellRanger using cellranger mkref and providing the genome FASTA and the GTF annotation (see above). We then mapped the scATAC-seq reads using 'cellranger-atac' with the reference index and standard parameters to create a region x cell sparse matrix for downstream analysis.

**Seurat/Signac analysis.** The resulting region x cell sparse matrix was loaded onto Seurat and Signac alongside the genome annotation used for mapping, to create a chromatin assay using the function 'CreateChromatinAssay()' and parameters 'min.features = 45', after which we turned into a Seurat object. We ran the function 'NucleosomeSignal()' with standard parameters to map signal from nucleosomes, and subset the Seurat object with the following parameters: 'peak_region_fragments <1000 & pct_reads_in_peaks > 15 & nucleosome_signal <1'. We identified neighbours for the k-nn graph using the function 'FindNeighbors()' and the following parameters: 'reduction = "lsi", dims = 2:30, k.param = 10'. We identified clusters using the function 'FindClusters()' with the Louvain algorithm[128] and the following parameters: 'resolution = 1, algorithm = 3'. We computed an UMAP projection using the function 'RunUMAP()' and the following parameters: 'dims = 2:30, reduction = "lsi", seed.use = 1'. To find cell markers in the scATAC data (either in the "peaks" or "RNA" assays), we used the function 'FindAllMarkers()' with the following parameters: 'only.pos = TRUE, min.pct = 0.1, logfc.threshold = 0.1'.

**Alignment between datasets.** To identify the cell type of the scATAC cells, we aligned the two datasets to transfer the scATAC Seurat cluster identity to cells in the scRNA object. We did it this way because the scATAC-seq data had lower resolution. This in turn, allows us to know what broad cell type of the scRNA data corresponds to scATAC clusters. We first calculated the Gene Activity function using the

'GeneActivity()' function of signac with standard parameters. We normalised the resulting Gene Activity data using the function 'NormalizeData()' and the parameters: 'normalisation.method = "LogNormalize", scale.factor = median(nCount_RNA)', where 'nCount_RNA' was the number of counts detected per cell in the scATAC Seurat object. We intersected the scRNA object variable features and the genes detected with gene activity in the scATAC object, and used these as common features for aligning the datasets using the function 'FindTransferAnchors()' with the following parameters: 'reduction = "cca", k.anchor = 5, k.filter = NA, k.score = 10, max.features = 1000', and using the scRNA object as query and the scATAC as reference. We transferred the labels using the 'TransferData()' function with the following parameters: 'weight.reduction = scrna_pca, dims = 2:30', where 'scrna_pca' was the PCA calculated for the scRNA object. The predicted labels of each cell in the query dataset were added as a metadata column. We assigned the most frequent predicted label (the scATAC cluster) to each cell type cluster we annotated in the scRNA dataset. We manually annotated the scATAC clusters after inspecting the results of the label transferring. Label transferring was visualised using igraph[129].

### Co-accessibility modules via OCR WGCNA analysis

We first ran a pseudobulk aggregation without replicates or conditions, only the cell types, as described above. The resulting matrix of OCR counts was normalised by "library size" using the 'DESeqDataSetFromMatrix()' function with parameter 'design = ~ condition' and the 'counts()' function with parameter 'normalised = TRUE' from the R package DESeq2[100]. We performed cell type co-occurrency as described above using this normalised matrix to retrieve a tree of cell type similarity based on their profile of chromatin accessibility in these differentially accessible OCRs. We scaled this matrix across rows and ran WGCNA using soft power 7 after visual inspection of the scale-free topology fit and median connectivity dynamics. The resulting TOM matrix was turned into a dissimilarity matrix and clustered using the R function 'hclust' with parameters 'method = "ward.D2"'. We cut the clustering in different modules of co-accessibility using the function 'cutreeDynamic()' from the WGCNA package[70] and the following parameters: 'deepSplit = 3, pamRespectsDendro = FALSE, minClusterSize = 100'. We reclassified the modules as described above for the WGCNA co-expression modules. To assess the likeness in expression of the genes associated to these OCRs, we ran the pseudobulk aggregation with broad cell type labels on the scRNA-seq dataset and normalised the counts as described above. We then subsetted and reordered this matrix to keep the genes associated to the OCRs used in the ATAC WGCNA analysis. We correlated the expression profile of every gene with the average accessibility profile of the ATAC module of their associated OCRs and retrieved the top 20 highly correlating genes with each module for visualisation using the ComplexHeatmap package.

To evaluate the agreement between co-expression and co-accessibility modules, we quantified and tested the enrichment of OCR/gene pairs between pairs of co-expression and co-accessibility modules. Briefly, we quantified how many OCR/gene pairs belonged to the same pair of modules of co-expression and co-accessibility and compared this to the null expectancy (upper-tail geometric test), using a custom R wrapper as previously described[136].

### WGCNA co-accessible OCR motif enrichment analysis

We performed motif enrichment analysis using the 'findMotifsGenome.pl' wrapper from the HOMER suite[132] with standard parameters, using the OCRs of interest as foreground and the rest of OCRs detected by cellranger as background. We concatenated the motif enrichment results into a single matrix which we filtered to keep only motifs with $q$-value < 0.1. These were visualised using the ggplot2[134] and ComplexHeatmap R packages[131].

### Pseudobulk computational dissection of scATAC-seq (for differential chromatin accessibility analysis)

We created a pseudo-bulk supermatrix with OCRs in rows and "pseudo-samples" in columns in the same fashion as described above. For replicates, we randomly split the cells into two pseudoreplicates. For one-versus-all analyses, we ran independent pseudo-bulk count aggregations, labelling every non-cell of interest as "else", aggregating counts from cells different from the cell type of interest (e.g. muscle cells vs non-muscle cells).

### Differential chromatin accessibility analysis (one-versus-all)

We performed differential chromatin accessibility analysis (DCA) to compare each differentiated cell type against the rest as follows: For a given contrast (e.g. muscle vs everything else), we first ran the pseudobulk count aggregation as described above. Secondly, we ran DESeq2[100] inside a custom wrapper with a contrast of "condition 1" (cell type X) relative to "condition 2" (everything else). Genes were identified as differentially expressed if having a $p$-adjusted below 0.05 (negative binomial test).

### Differential chromatin accessibility analysis (one-versus-neoblasts)

We performed differential chromatin accessibility analysis to compare differentiated cell types against neoblasts as follows: For a given contrast (i.e. comparison of cell type X vs neoblast), we first extracted the relevant pseudo-samples from the pseudo-bulk supermatrix of scATAC-seq. Secondly, because we were comparing cell types, we used "cell type" as conditions and the pseudoreplicates as replicates. Third, we ran DESeq2[100] inside a custom wrapper with a contrast of "condition 1" (cell type X) relative to "condition 2" (neoblasts). Genes were identified as differentially expressed if having a p-adjusted below 0.05 (negative binomial test).

### Association of OCRs to genes

We associated every OCR to a gene using 'bedtools closestbed'[122] with the following parameters:'-k 1 -D ref -a all_peaks_sorted.bed -b TSS.bed', where 'all_peaks_sorted.bed' is the BED file of all OCR coordinates and 'TSS.bed' is the BED file of the TSS coordinates of all genes.

### Co-accessibility modules via OCR WGCNA analysis, using Differentially accessible OCRs only

We first ran a pseudobulk aggregation without replicates or conditions, only the cell types, as described above. The resulting matrix of OCR counts was normalised by "library size" using the 'DESeqDataSetFromMatrix()' function with parameter 'design = ~ condition' and the 'counts()' function with parameter 'normalised = TRUE' from the R package DESeq2[100]. We subsetted this matrix to keep all the OCRs detected as significantly open (log2FC > 0) in any of the differentiated cell types of our DCA analysis (see Methods above). We performed cell type co-occurrency as described above using this normalised matrix to retrieve a tree of cell type similarity based on their profile of chromatin accessibility in these differentially accessible OCRs. We scaled this matrix across rows and ran WGCNA using soft power 8 after visual inspection of the scale-free topology fit and median connectivity dynamics. The resulting TOM matrix was turned into a dissimilarity matrix and clustered using the R function 'hclust' with parameters 'method = "ward.D2"'. We cut the clustering in different modules of co-accessibility using the function 'cutreeDynamic()' from the WGCNA package[70] and the following parameters: 'deepSplit = 4, pamRespectsDendro = FALSE, minClusterSize = 30'. We reclassified the modules as described above for the WGCNA co-expression modules. To assess the likeness in expression of the genes associated to these OCRs, we ran the pseudobulk aggregation with broad cell type labels on the scRNA-seq dataset and normalised the counts as described above. We then subsetted and reordered this matrix to keep the genes

associated to the differentially accessible OCRs used in the ATAC WGCNA analysis. We correlated the expression profile of every gene with the average accessibility profile of the ATAC module of their associated OCRs and retrieved the top 20 highly correlating genes with each module for visualisation using the ComplexHeatmap package.

### Bulk ATAC-seq analysis
We mapped the bulk ATAC-seq reads using a custom perl wrapper of bowtie2 as described in Pérez-Posada et al.[136]. We then called for peaks using a custom bash wrapper of MACS2 as described in Pérez-Posada et al.[136], with the following parameters: ' -f BED --nomodel --extsize 100 --shift 45 --buffer-size 50000 -g 840213658 -p 0.001'.

### Gene track and chromatin visualisation
**Generation of gene annotation tracks.** We parsed the GTF gene annotation track using GenomicRanges and rtracklayer[137] in R, and transformed it to a suitable format using the R package GenomicRanges[138].

**Generation of scRNA-seq alignment tracks.** For every gene_function_tagged BAM file of our scRNA-seq libraries, we ran sinto (https://github.com/timoast/sinto) to split them into different bam files, one per broad cell type, by filtering reads labelled with barcodes from cells assigned to that cell type, using the following parameters: '-barcodetag XC'. After splitting, we merged all the bam files from different libraries but from the same cell type to create one unique alignment file per cell type. We used bamCoverage to convert them to BigWig files with standard parameters. BigWig files were visualised using the 'GenomicRanges' and 'gviz' R packages[138,139].

**Generation of scATAC-seq-seq alignment tracks.** We split the scATAC-seq BAM file generated by cellranger-atac using sinto and the list of cell barcodes labelled for each cell type, to generate independent BAM files for each cell type. We used bamCoverage to convert them to BigWig files with the following parameters: '--normalizeUsing BPM'. BigWig files were visualised using the 'GenomicRanges' and 'gviz' R packages[138,139].

**Genomic Coordinates visualisation.** We retrieved the genomic coordinates of scATAC-seq markers and plotted them using a custom gviz[139] wrapper in R. These were manually inspected.

**Chromatin enrichment profiles.** Chromatin plots were generated and visualised using the GenomicRanges[138] and EnrichedHeatmap package[140].

### ANANSE analysis
**Chromatin pre-processing.** We used the same BAM files generated using sinto (https://github.com/timoast/sinto) as input for ANANSE binding. For a peak catalogue file, we re-centred the coordinates of the peaks called by cellranger-atac around the summit of the signal using a custom python wrapper of 'bedtools slop'[122] and 'samtools mpileup'[141]. We kept peaks with a minimum of 5 counts detected.

**Gene expression pre-processing.** We generated independent tables of normalised counts for each broad cell type using the pseudobulk approach without replicates or conditions, only broad cell type labels, as described above.

**ANANSE binding.** For each broad cell type, we ran ANANSE binding[59] with parameters '--jaccard-cutoff 0.2', providing the genome FASTA, the respective broad cell type BAM file, the summit-centred peak file, and the motif2factors lookup database we generated using gimme motif2factors (see above).

**ANANSE network.** For each broad cell type, we ran ANANSE network[59] with parameters: '--full-output', providing the genome FASTA, the respective broad cell type counts table, the GTF annotation of S. mediterranea, and the resulting.h5 ANANSE binding file for the respective cell type.

**ANANSE influence.** We ran ANANSE influence[59] for every pair of contrasts (from neoblast network to a differentiated cell type X), using the following parameters: '-n 12 -i 250000'. The resulting output (influence data frame and differential network) were analysed in R using 'igraph'[129].

**ANANSE network pre-processing, graph analysis and visualisation.** The resulting ANANSE networks were imported into R and analysed as follows. For every network, we subsetted the network to keep interactions above score value 0.8. We created graph objects using igraph and removed genes without neighbours.

We calculated centrality, out-centrality, in-degree, and out-degree for every gene using igraph. Relative out-degree was calculated as in-degree divided by the sum of in-degree and out-degree. We calculated the number of active TFs as the number of genes with outdegree above 0. For every network, we extracted the centrality values for all the TF genes in the network. We collapsed these values together in a TF x cell type network matrix of centrality values. We correlated the columns of this matrix using the following parameters: 'method = "ward.D2"'. For graph visualisation, we subsetted the graphs to keep the influential TFs from the ANANSE influence analysis, and we kept the top 2 interactions (based on ANANSE's prob score) of each TF. For graph visualisation of candidate TFs with data from the literature, we subsetted the graphs to keep the TF of interest and the top interactions between the TF of interest and its direct neighbours, together with any interaction between the direct neighbours. The resulting graphs were plotted using the graphopt algorithm from igraph[129].

For target visualisation of each network, we first retrieved the target genes of each of the top five TFs of each fate. For each TF in each fate, we kept interactions with target genes above the 0.95 quantile (top 5% interactions). We visualised them using the 'geom_jitter()' function from the R package ggplot2[134]. Names of target genes detected in the literature from PlanMine[110,111] were manually curated and shown next to the strip chart.

To detect groups of co-influential TFs, we collapsed the influence score values of each TF in the network of each fate into a single TF x cell fate influence score matrix. We clustered these TFs using Pearson correlation of their influence profile and the method 'ward.D2'. The resulting tree was cut above 0.7 to retrieve small clusters of TFs with similar influence profiles across fates. These clusters were automatedly sorted as described above for WGCNA modules. Within each cluster, we correlated every TF with the average co-influence profile of their parent cluster and sorted them in descending order. We chose the top five TFs for each cluster for visualisation using the R package 'ComplexHeatmap'[131].

### Re-analysis of publicly available data
RNA-Seq reads from[75,82,94,95] were downloaded from GEO (accession numbers SAMN06142985, SRP002478, GSE241516, GSE80562, GSE72389, PRJNA235907) and processed and analysed following approximate instructions from their original publications, where available. Briefly, we mapped and quantified the reads against the latest version of the S. mediterranea genome using a combination of bowtie2[142] and HTSeq[143], or salmon[144]. For DGE, the resulting counts were analysed using DESeq2[100], and DEGs were identified based on adjusted p-value (Wald test) and log2Foldchange. For the antero-posterior transcriptomic dataset and the prep knockdown dataset, reads were normalised in FPKM and processed as originally

described[93,94]. Detailed methods are available in Supplementary Note 4.

For gene score analyses, for each cell, we first calculated the average expression of the genes of interest; we then calculated the average expression of a random subsample of all genes; then we subtract the second value from the first value. These values, one per cell, were then normalised, pooled based on cell cluster, and plotted and visualised using ggplot2[134]. To visualise the interactions predicted by ANANSE between TFs and DEGs, we imported the ANANSE networks of the cell type of interest into R, and subsetted them to keep only interactions stemming from the TF of interest. We labelled these genes as downregulated, or nondownregulated, in each of the network data, and visualised their predicted scores and weighted binding.

### *hnf4* Knock-down scRNA-seq analysis

**scRNA-seq pre-processing and generation of scRNA matrices.** We pre-processed the reads of libraries 11.3 and 11.4, from the *hnf4*i experiment, as discussed above. To generate scRNA matrices for each library, the resulting outputs of running the SPLiT-Seq pipeline (tag_bam_with_gene_function) were used to generate a gene x cell sparse matrix keeping cells with a minimum of 100 genes detected using the command 'DigitalExpression' from dropseqtools.

**Matrix concatenations.** The resulting matrices were concatenated using Seurat v4.3.0[62] in R v4.0.3. Briefly, we created independent Seurat Objects for each separate library and performed Proportional Fitting-log1p-Proportional Fitting normalisation[126] on each separate matrix. We ran 'FindVariableFeatures' using the parameters: 'selection.method = "vst", features = 20,000'. We labelled cells from each library with their respective library ID (Supplementary Data 27) and merged all the Seurat objects into a single object with a concatenated sparse matrix.

**Seurat analysis.** The concatenated sparse matrix was queried for highly variable genes using 'SelectIntegrationFeatures()' from the Seurat package. We scaled the data and performed PCA using the function 'runPCA()' with the following parameters: 'npcs = 50'. We integrated the data using the function 'runHarmony()' from Harmony[127] and the following parameters: 'dims.use = 1:50, theta = 3, lambda = 3, nclust = 40, max.iter.harmony = 20'. We identified neighbours for the k-nn graph using the function 'FindNeighbors()' and the following parameters = 'dims = 1:50, k.param = 65'. We identified clusters using the function 'FindClusters()' with the Louvain algorithm[128] and the following parameters: 'resolution = 2.5, algorithm = 1, random.seed = 75'. We computed an UMAP projection using the function 'RunUMAP()' and the following parameters: 'dims = 1:50, reduction = "harmony", n.neighbours = 65, min.dist = 0.5, spread = 1, metric = "euclidean", seed.use = 1'.

**Alignment to reference dataset.** To align the *hnf4*i dataset to our whole atlas analysed in this study, we ran the function 'FindTransferAnchors()' from Seurat using the *hnf4*i Seurat object as query and the Seurat object of our scRNA atlas of thirteen libraries as the reference, and the following parameters: 'dims = 70'. The predicted labels of each cell in the query dataset were added as a metadata column. We assigned the most frequent predicted label to each Seurat cluster we obtained. Upon inspection and downstream analyses, we manually curated the assignment of phagocyte progenitors from the *hnf4*i libraries (as opposed to the control libraries) as aberrant phagocyte progenitors. Label transferring was visualised using igraph[129].

**Cell abundance analysis.** To calculate the over- or under-representation of cells from experimental condition in our dataset, we performed pre- and post-hoc Chi-Squared test as described

previously[57] using a custom R wrapper. These resulting residuals were visualised using ComplexHeatmap[131].

**Gene score analysis.** We first retrieved markers for each cell type of interest (neoblasts, phagocyte progenitors, phagocytes, epidermis) using the function 'FindMarkers' from Seurat and the following parameters: 'only.pos = TRUE, return.thresh = 1, logfc.threshold = 0', and from this retrieved the top 50 markers based on log fold change. For each set of markers of interest, we calculated a gene score as follows: for each cell, we first calculated the average expression of the genes of interest; we then calculated the average expression of a random subsample of five percent of all genes; then we subtract the second value from the first value. These values, one per cell, were pooled based on cell cluster and experimental condition, and plotted and visualised using ggplot2[134].

**Differential gene expression analysis.** To perform DGE analysis on each cell type separately, we first computationally dissected the *hnf4*i dataset to retrieve a super matrix of genes in rows and "pseudo-samples" in columns, each "pseudo-sample" being a combination of cell type, experimental condition, and replicate, as described above. For each type, we first subsetted this supermatrix to extract the columns relevant to the cell type of interest. We then filtered out genes with less than one count in at least two replicates. We then ran DGE analysis using a custom R wrapper of DESeq2[100], with the following contrast: 'c("condition","HNF4i","control")'. Genes were identified as DEGs if having a *p*-value below 0.05 (negative binomial test). For each cell type, the results of DESeq2 (log2FC and -log *p* value) were plotted using ggplot2[134]. This analysis was performed first using broad cell types, but also using the individual, narrow cell types (Supplementary Fig. 21A,B).

**Downstream analysis of differentially expressed genes.** We retrieved the lists of DEGs from the cell types with the highest fraction of DEGs in their analysis. We calculated the overlap between these using the eulerr R package. For each cell type, the results of DESeq2 (log2FC and -log *p* value) were plotted using ggplot2[134]. For each set (exclusive to phagocytes, exclusive to parenchyma, common to both, or all of them together) performed Gene ontology enrichment and motif enrichment analyses as described above.

To visualise the *hnf4* interaction predicted by ANANSE on the DEGs, we imported the ANANSE networks of phagocytes and parenchyma into R, and subsetted them to keep only interactions stemming from *hnf4*. We labelled these genes as DEG in each of the sets (exclusive to phagocytes, exclusive to parenchyma, or common to both) and visualised their predicted score. Logistic regression was performed using the 'glm' function from R with parameters: ' family = binomial(link = "logit")'.

### Reporting summary

Further information on research design is available in the Nature Portfolio Reporting Summary linked to this article.

## Data availability

The sequencing and processed data generated in this study (including scRNA-seq reads, scATAC-seq reads, bulk ATAC-seq reads, bulk RNA-seq reads) have been deposited in the GEO database under accession code GSE274286. The processed data (processed files such as.RDS single cell Seurat objects) have also been deposited in the GEO database under accession code GSE274286. The processed data generated in this study are provided as Supplementary Data, and additional processed data can be found at our Code Repository (see Code Availability). The planarian genome assembly used in this study is available in the NCBI database under accession PRJNA1052007. Sequencing data of planarian anteroposterior transcriptomics is available in the NCBI SRA under accession PRJNA357536 [https://www.ncbi.nlm.nih.gov/bioproject/357536/]. Sequencing data of planarian *pax2/5/8-1* and *soxP-3*

knockdown experiments used in this study are available in the NCBI Database under accession PRJNA293934. Sequencing data of planarian *p53* knockdown experiments used in this study are available in the NCBI Database under accession PRJNA293934 [https://www.ncbi.nlm.nih.gov/bioproject/ PRJNA293934/]. Sequencing data of planarian *coe* knockdown experiments used in this study are available in the NCBI Database under accession PRJNA235907. Sequencing data of planarian *prep* knockdown experiments used in this study are available in the NCBI Database under accession PRJEB4686. Sequencing data of planarian *alx3-1* knockdown experiments used in this study are available in the NCBI Database under accession PRJNA1008560.

## Code availability
The code used for all the analyses in this study is available in GitHub (https://github.com/scbe-lab/regulatory_logic) as well as Zenodo (https://doi.org/10.5281/zenodo.17253117)[145].

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

## Acknowledgements

Research at the SCBE lab at Oxford Brookes University and at the Living Systems Institute is supported by MRC grants (MR/S007849/1 and MR/W017539/1), a BBSRC Grant (BB/V014447/1) and a Leverhulme Trust grant (RPG-2019-332 and RPG-2023-330) to J.S. H.G.-C. and E.E. were supported by Nigel Groome studentships from Oxford Brookes University. S.F. and S.J.v.H. were supported by the Netherlands Organization for Scientific Research (NWO grant 016.Vidi.189.081) to S.J.v.H. We thank Isabel Liao for advice and discussions with the weight normalisation, María Rosselló for advice and discussions with the differential gene expression analysis, Luis Ferrández-Peral for advice with the knockdown transcriptomic analyses, and all members of the Solana lab for useful discussion, input and assistance. We thank the Wellcome Centre for Human Genetics for their expertise in generating the scATAC-seq dataset, especially Rory Bowden and Hubert Slawinski. Flow cytometry was performed at the Sir William Dunn School of Pathology Flow Cytometry Facility, University of Oxford with the assistance of Dr Robert Hedley.

## Author contributions

J.S. conceived the study, designed the experiments, and provided supervision. H.G.C. and E.E. generated cell dissociations and performed single-cell transcriptomic experiments using *Schmidtea mediterranea*, both in unperturbed and knock-down animals. J.S. generated the library of scATAC-seq. V.V. generated the bulk ATAC-Seq library. C.A.B. and N.J.K. performed preliminary bioinformatic analyses. S.F. performed preliminary ANANSE bioinformatic analyses. J.S. and A.P.P. designed the final bioinformatic analyses. A.P.P. performed all the final bioinformatic analyses. S.F. and S.J.V.H. contributed to the interpretation of the ANANSE network data. A.P.P. generated the figures. A.P.P. and D.S.D. retrieved and re-analysed the transcriptomics data from the public literature. A.G.F., A.P.P., J.S., and F.C. designed the double knockdown experiments. A.G.F. performed the double knockdown experiments. A.P.P. performed the computational analyses of the double knockdown experiments. A.P.P. and J.S. wrote the manuscript, with contributions from all other authors. All authors read and approved the final version of the manuscript.

## Competing interests

The authors declare no competing interests.
