## [Peer Review file · Nature Communications]

Multimodal single cell analyses reveal gene networks of planarian stem cell differentiation

Corresponding Author: Dr Jordi Solana

Version 0:

Reviewer comments:

Reviewer #1

(Remarks to the Author)

Summary

This manuscript describes the characterization and integrated analysis of snATACseq and scRNAseq from *Schmidtea mediterranea*. Some of the data are new, and some are re-analysis and integration of previously published data. The analysis generated predicted regulons of TFs, and the authors selected *hnf4* for validation via RNAi followed by scRNAseq.

Major points:

1. The primary objective of the manuscript is not clear. The abstract and introduction imply that the study aims to develop generalizable methods to address challenges such as integrating perturbational data into single-cell data and/or constructing accurate gene regulatory networks for multicellular organisms. However, in the Results, the paper immediately shifts diving into designing experiments for sequencing large amounts of high-throughput data in multicellular organisms and characterizing TF influences across cell types. There is no integration of perturbation data other than differential gene analysis performed at the end. Even though *hnf4*'s effect on neoblast differentiation is prospectively tested, the absence of other TFs conflicts with the claim in the abstract that the goal is to validate regulatory logic for the organism as a whole.

2. Figure labeling has multiple mistakes, and it is confusing for understanding, here are some examples:

- Figure 1

- o Page 3, line 120: the cited figure by its description does not match 1F.
- o Page 3, line 121: Figure 1G is mentioned but it is missing in figure page.

- Figure 2

- o Page 3, line 176: the cited figure by its description does not match 2C.
- o Page 2, line 181: the cited figure by its description does not match 2D.

- Figure 5

- o Page 8, line 350- 352: the cited figure by its description does not match 5E.
- o Page 8, line 356: the cited figure by its description does not match 5F.
- o Page 8, line 356- 358: the cited figure by its description does not match 5G.

3. Integration of ATAC and RNA data from different cells makes the assumption that transcriptomic state is well-predicted by chromatin accessibility. But this assumption is not necessarily true, especially when the system is differentiating pluripotent stem cells. The extent to which this undermines the network/GRN reconstruction needs to be addressed.

Minor points:

Data is not available.

Animal sizes were randomly selected. Are these animals in similar ages? Size difference could indicate age difference, adding another independent variable to the experiment.

Page 3, line 108-110: For identifying scATAC-seq cell types, based on Supplementary Figure 3 and the Methods, it appears the author utilized all scRNA-seq data, including those treated with RNAi for *hnf4*. Could this influence the cell type

identification for the ATAC-seq data, given that the scATAC-seq data were derived from cells without RNAi treatment for hnf4?

Figure 1E contains blocks that are too small to clearly distinguish their colors, and the row labels are confusing. Additionally, the figure caption lacks a detailed explanation that would help to understand Figure 1E.

Consider putting Figure 2A and 2B in supplementary figures. These figures provided too much information and not all information is discussed in the result. You could replace Figure 2A by selecting only the relevant modules or parts of the modules. Similarly, replace Figure 2B by focusing on a select group of TFs, as was done in Figure 2C. Later, in page 5, line 197-199, the claim of differentially accessible OCRs and genes have similar motif enrichment in will be easier to observe. Page 4, line 169-171: The author claims that TFs with high TF/module connectivity also show high motif/module enrichment. However, the provided Figure 2C and 2D does not strongly support this claim, though the author highlights some examples that do demonstrate the association. A counterexample is the GATGTG box and its corresponding TF mitf1-1, which does not exhibit high motif connectivity in modules m37, m45, m23, m46, m32, m41, and m42. GATGTG box has high TF/module connectivity but low motif/module enrichment in these modules. A statistical analysis, such as matrix correlation or element-wise comparison, would help determine whether an association between TF/module connectivity and motif/module enrichment truly exists.

Page 5, line 194-196: The author mentions splitting the cells into two pseudoreplicates, but Figure 3A does not clearly illustrate the splitting process or the subsequent analysis involving these pseudoreplicates. Additionally, no rationale is provided for why two pseudoreplicates were used instead of simply performing two differential chromatin accessibility analyses on the original dataset.

Page 5, line 197-199: the author claim that differentially accessible OCRs is similar to WGCNA however Figure 3B is not supporting this claim. Figure 3B show cell motif enrichment analysis for a list of TFs in different cell types. This list of TFs is different in Figure 2C. Also, the cell types are different then in transcriptome analysis.

Page 5, line 205-206: Supplementary 8E does not have good figure caption and axis label. It is confusing to read. Similarly, Supplementary Figure 9A as well.

Page 5, line 216-220: The author used differentially accessible OCRs as input for the WGCNA analysis, while using all gene expressions for the transcriptomic analysis in WGCNA. What is the rationale for using only selected OCRs for WGCNA analysis rather than all OCRs? Additionally, how can the similar cell type combinations between the gene expression and OCRs WGCNA analyses be justified, given the differences in input?

The modules of expression and OCR have the same naming system (m module or s module). It is confusing sometimes when referred in the article. The author should give different names to module of expression and OCR, such as me/se for expression, mo/so for OCR. This unclear naming systems add extra difficulty to understand the analysis done for making connections between gene expression WGCNA analysis and OCRs WGCNA analysis and corresponding Figure 3G and Supplementary Figure 9G.

Page 7, line 285-298: The author chose hnf4 as the primary TF for further analysis, but there is no discussion of the exclusion criteria for other interesting multi-cell-type influential TFs. Upon closer examination of Figure 4M, several other TFs—such as mxipl, gtf2a1, and mef2-2—also appear to influence multiple cell fates and align more closely with the idea of regulating differentiation from neoblasts to other cell types. Hox4b and est-3 also have high influence scores across multiple cell types, similar to hnf4. In a hypothesis free style such as this manuscript, the author should discuss why was hnf4 chosen over these other candidates with the analysis result.

Page 8, line 226-227: the previous study (Emili et al., 2023) did not include hnf4-silenced (aberrant) phagocyte progenitor cell groups. How did the author characterize this group? What additional considerations were made when identifying gene markers for this group during cell type transfer?

Page 8, line 345-353: To determine whether silencing hnf4 halts all differentiation or halts specifically targets phagocyte differentiation, the author should conduct the same gene score analysis on epidermis and parenchymal cells across the control/B1, control/B2, hnf4i/B1, and hnf4i/B2 cell groups. If the gene scores for all four cell groups in both the epidermis and parenchymal cells indicate differentiation, it can be concluded that differentiation is not universally silenced across all cell types. Additionally, the neoblast score graph shows minimal variation between neoblasts and other cells. A statistical test is recommended to confirm whether the gene scores effectively differentiate cell types.

Page 9, line 395- 396: interaction score calculation is not discussed in the manuscript. Is the interaction score calculating how in silico hnf4 is connected to in vivo DEGs? Without detailed description, Figure 6F hardly supporting the claim that in silico hnf4 regulating genes are overlapping with in vivo DEGs in phagocytes. Showing a simple overlap between the two group of genes in a Venn diagram format could straightforwardly prove the claim.

"Thus, cell differentiation comprises the combined expression of TFs and the combined accessibility of open chromatin regions (OCRs) acting as cis-regulatory element (CREs). This combination creates a 'regulatory logic' forming gene regulatory networks (GRNs) 5-7." Other sources of epigenetic information contribute to 'regulatory logic' including DNA methylation.

"Single-cell methods have democratized the study of differentiation trajectories...". It is unclear what is meant by "democratized" here.

"Gene expression is highly pleiotropic." Taken literally, this means "there are varied causes of gene expression". Yes, this is true, but I don't think this is what the authors meant to convey.

"It is the combination of genes expressed in each cell type that defines their identity." This statement is not universally accepted, many believe that function defines cell identity.

Is *hnf4* expression altered in the *hnf4*-RNAi worms/cells? I did not see this data in the supplemental figures.

(Remarks on code availability)

Reviewer #2

(Remarks to the Author)

This manuscript by Perez-Posada et al. aims to clarify the gene regulatory networks in planarian cells. The study contains a huge amount of data and analysis, and is one of the largest single cell studies in the planarian literature. The scRNAseq data in part was previously published in the context of other studies, but part was newly generated, and the scATACseq data is all new to this study. From the analyses it appears that the data is of adequate quality, however, single cell libraries often suffer from low read depth, and this is also mentioned as a limiting factor in this study. The analyses performed are state-of-the-art and the figures are impressive. It is clear from the manuscript that the authors have put a great deal of effort into the acquisition of the data and in the computational analysis. The amount of resources and skills that this requires are not to be underestimated, and I congratulate the authors on this tour de force.

However, the thing that is sorely missing from this study is a message. With all the data and all the analysis, there doesn't appear to be a novel finding or a new insight that is generated. Due to the lack of message, the manuscript lacks focus, and is difficult to digest. I appreciate that it may be hard to dig through this enormous amount of data and pick something specific to zoom in on, but without that the study remains just a heap of data. The data is complex and multidimensional, and that is to be expected of a single cell dataset, but that just means that the authors need to pre-digest the information better to make this understandable and interesting to the reader. Because there is no focus, the figures keep showing everything, making them too large and too comprehensive. In their current shape it is not possible to read the amount of detail that is in the figures and it is impossible to know what the authors want the reader to see in them. While impressive at first glance, to show everything is almost the same as to show nothing at all.

Figures 1-4 lay out all the data. Figures 1-3 are framed as verification of the data. These figures set the stage for the main message, so it is understandable that not much interpretation is provided. Figure 4 then contains an advanced computational analysis using a published pipeline (ANANSE) integrating all the data. This results in an impressive set of plots, and the reveal of what appear to be the GRNs that the abstract of the study refers to. However even here the interpretation is lacking. The authors find that there are multiple transcription factors involved in gene expression, and that some are cell-type specific, and some are not. The graphs suggest that some transcription factors may form networks regulating each other, but it is not clear how strong the evidence for these connections is, and none of them are experimentally verified. The meaning of these networks therefore remains unclear. Even the names of the transcription factors are difficult to read in the dense figures. After this section there is a bit of follow-up in Figures 5-6 describing the RNAi phenotype of intestinal transcription factor *hnf4*. The authors use scRNAseq to find that knockdown of *hnf4* results in loss of phagocytic cells without a major effect on the neoblasts. This phenotype has however already been reported based on stainings (PMID: 38165802). While the method applied in the current manuscript is considerably more fancy than in the previous study, it again doesn't lead to a significant new insight.

The aim of the authors appears to have been to create an atlas of GRNs, somewhat similar to the single cell RNAseq atlas of planarian cell types that the senior author previously generated. Since then another study has already presented an atlas of the planarian transcription factors based on scRNAseq data (PMID: 38401119). An atlas however can probably not explain the "logic of planarian stem cell differentiation" as the title of this manuscript claims - and while a lofty goal, this manuscript does little to advance this. In fact, the stem cells are hardly discussed in the entire manuscript. The scATACseq data seems challenging (based on the sparse information in the heatmaps shown in the supplementary figures) but even without that data it would have been possible to generate an interesting story. There are many potentially interesting observations that could be the starting point for a new insight if they would have been followed up on. The authors for example point out that previous studies have only identified gene expression clusters that correspond with cell clusters, and that they for the first time identify expression modules that are shared by multiple cell types. Further analysis of these gene sets and their relevance could generate new insights in the functioning of planarian cell types and in the use of conserved gene modules. This would however require significant further work.

An additional problem is that the figure legends frequently are too brief to understand what exactly is shown. This makes it difficult to know how to interpret the data, and makes it hard to follow the authors when they make statements about the data. For example:

Fig 1F. Are the rows single cells showing accessibility at the same locus used in Fig 1E, or are the rows different loci in the cluster and is the scATAC summarized for each row? In either case, why is there no TSS signature?

Fig 2D. What is an enrichment log q value? Is this a $10\log$?

Fig 3B. Is this focused on accessibility of promoters, or is this any ATAC peak? Also, there appears to be very little overlap in TFs between the ones in this panel and those in Fig 2C. This makes it very hard to correlate the data, even though the text states that they are similar. Better explanation of the selection of the TFs and how to interpret the (absence of) overlap would be helpful.

Fig 3 CD. Is each dot in the scatterplot a gene, or a promoter, or an ATAC peak independent of annotation? It is odd that no neoblast-specific accessibility is found. In contrast to what the authors state, a previous study using bulk neoblast ATACseq had reported clear neoblast-specific regions of ATAC signal (PMID: 37801496 - although they found no TSS peaks and no specific TFs that would regulate these regions).

Fig 3FG. It is not clear what these panels represent, and how to interpret the data. Are the same genes (rows) shown in both panels? Isn't it somewhat expected that genes that are more accessible in a tissue produce higher RNA levels in that specific tissue? Is this only intended as verification of the data?

Fig 4. ANANSE is not a commonly known analysis tool (yet) and while the supplement gives general information about the workings of this tool, the application in this study still requires further explanation. The original tool uses enhancer regions, and TF binding sites (and motifs) that are experimentally determined in mammals. How is this translated to planarians that appear to have little enhancer activity, much higher gene density, and where no TF binding sites are known and even the planarian homologs of mammalian factors are often not unequivocal?

Fig 4B. unclear what this is, other than a correlation matrix - but correlation of what exactly?

Fig 4 C-J. What are these strip plots? How should we interpret the values at the strip plot y axis? Why are some TFs shown here - how were they selected, and what should be deduced from the plots? Are these the GRNs that are referred to in the title? Why is there not more emphasis and explanation of these networks?

Fig 4L. Not sure what this means.

Fig 4M. Why are the TFs at the center of the networks in 4C-J often missing from this panel?

Fig 5. Much of this looks like it should have been in the supplement.

Fig 6. This looks like a Main Figure again, but there is no new insight and no take-away message. Why was *hnf4* selected for follow-up? What was the question that was addressed by this experiment?

I won't comment on the Supplementary Figures to avoid repetition but similar points apply there.

In summary, the authors did a fantastic job on the generation of this dataset and on the global analysis. Unfortunately a global analysis does not make for a very compelling story, and in its current shape it is not clear what readers would learn from this study - if they would even manage to read through the entire document. While it is of course entirely up to the authors to present their data in the way that they like, my recommendation would be to abandon the idea of an omnibus, and rather find a focal point that can make the data shine.

(Remarks on code availability)

NA - I think there are bigger issues than whether the code is readable or not.

Reviewer #3

(Remarks to the Author)

(Remarks on code availability)

Reviewer #4

(Remarks to the Author)

Summary:

In this manuscript, the authors aim to uncover how transcription factors influence the differentiation of planarian stem cells into various lineages. They use scRNA-seq and scATAC-seq to characterize the chromatin and transcriptional profiles of these cells. As in other recent papers from this group, the authors use a method of scRNA-seq they developed that optimally preserves and profiles the transcriptome of planarian cells. They do a thorough job of analyzing many thousand cells and perform experiments with biological replicates, including one in which they knock down a conserved transcription factor, *hnf4*. They also perform single cell ATAC-seq (scATAC-seq) using planarian cells, a first in the field at this time. The data in this manuscript will be a valuable resource to the field, and the analyses the authors perform have the potential to inform testable hypotheses about the roles of specific transcription factors interactions during planarian stem cell differentiation. However, the authors do not address those questions here. They do confirm an outstanding hypothesis in the planarian literature, namely that *hnf4* regulates the differentiation of cells in the intestinal and parenchymal lineages (Forsthoefel et al 2012, Van Wolfswinkel et al 2014, Fincher et al 2018). We think this data is worthy of publication, but suggest the authors make a better argument for why these data and analyses will advance the field. Below we list some specific questions that the data raised for us and give some suggestions about how to address them.

Major points:

1. In Figure 3, the authors show that muscle and neuronal cells have more cell type-specific open chromatin regions (OCRs). Moreover, this distinction exists without normalizing to neoblast OCRs. Can you address why this might be? Are these lineages specified from neoblasts differently? Do these lineages have more sub-classes, but the resolution of the scATAC-seq wasn't sufficient to distinguish them? Are the TFs that drive this process of certain classes? Something else?

2. As the authors describe in their introduction, multiple TFs often work together to bind targets and regulate transcription synergistically. Have the authors tried any experiments in which they perform combinatorial RNAi of multiple TFs? Perhaps for TFs with weak or no obvious phenotype when knocked down singly? The analyses presented here provide a great roadmap for such experiments.

3. Related, in Figure 4M the authors show that two "s" modules (s01 and s04) are "influenced" by TFs that also influence several other cell fates. However, an "s" module is defined as a set of marker genes that are highly enriched in only one cell type. Can you discuss what is different about the s01 and s04 modules/cell types and/or the TFs that regulate them (versus the other s modules that generally seem to be "influenced" by fate-specific TFs)?

Minor points:

Figure 1D: Where available, please add gene names along with gene IDs to help the reader connect them to each lineage.

Figure 2A: It would be helpful to label the black bars as "log number of genes" and shift the "expression" legend so it's above the data to which it refers. This would make it more immediately clear to the reader what each element represents. Or rather than showing a bar plot for the "log number of genes" you can simply label the number of genes in each module. To this end, aren't "s1-s24" and "m1-m53" gene modules not specific genes? They should then be labeled on the left as modules. This would help connect Figure 2A and 2B.

Figure 2E: Please add the cell type name in addition to the cell cartoon for better clarity and readability.

Figure 3: It's nice that the authors have color-coded the font according to cell type throughout the paper, but would still be helpful to include a specific legend detailing which cell type is represented by each color in the stacks, for example that one in 3F.

Line 319: The authors show that *hnf4*-RNAi worms have a lethal phenotype. However, Lobo et al., 2016 reported no major change in blastema size during regeneration. Please address this finding in the text: is it an RNAi delivery issue? Or is *hnf4* dispensable for intestinal differentiation during regeneration?

In Figure 4M: 1) why are some TFs are not labeled? Are these unknown? Then their gene ID should be listed.

Figure 5C: Can authors color Cluster 17 differently here to highlight it? Additionally, it would be helpful to show a UMAP with *hnf4* transcript expression in control vs *hnf4* RNAi.

Line 352: Figure 5E is referenced when it should be Figure 5D.

Figure 5D: In the phagocyte score plot, please add asterisks or other symbols to indicate statistically significant changes.

Figure 6l: The shaded colors representing cell types/lineages are difficult to see. Also here and possibly in Figure 2 where you show this network, it could be helpful to make a gradient shadow or somehow indicate that the larger network encompassed by the green shadow includes both parenchyma and gut network nodes. Here, without the pink and green node colors the impression is that all these nodes of connectivity are in the green gut network.

Supplemental Figure 12: although the authors describe the percentages of each defect, and the timing of phenotype progression in the text, it would improve understanding for the reader to indicate these details on the figure.

(Remarks on code availability)

There is a significant amount of code in this paper, so we did not run all of it to test reproducibility of every analysis. However, their GitHub page is well organized into logical directories and their code is appropriately annotated so that others should be able to follow the steps of their pipelines. All analyses also appear to have been performed using available packages that can be downloaded from Bioconductor or other publicly available sources.

Reviewer #5

(Remarks to the Author)

(Remarks on code availability)

Version 1:

Reviewer comments:

Reviewer #1

(Remarks to the Author)

Thank you for addressing all of our questions and suggestions thoroughly.

(Remarks on code availability)

Reviewer #2

(Remarks to the Author)

This revised manuscript is considerably improved over the original. The manuscript now has a direction, and the figures are legible. The authors also cleaned up the content of the figures by moving some of the busy panels to the supplement and adding new simplified versions of several of the plots as well as explanatory schematics. In addition, they added some RNAi experiments to help interpret their finding of co-regulated transcription factors. Together, these changes makes that the manuscript is easier to read. However the conceptual novelty that is presented by the manuscript remains meager. I still struggle to define what it is that we have learned from this manuscript that we did not know beforehand. Further, while I appreciate the author's aim to fully interpret the data by adding in inferences from many different sources, in several instances the accumulation of inference upon inference pushes the conclusions beyond the data.

The authors find two "supergroups" of cell types, which were not reported before, and they provide more detail on the hnf-4+ supergroup to show that this includes intestinal cells as well as a subset of parenchymal cells. They further show that hnf-4 works alongside other transcription factors in these two different cell types, and that this in part explains that different genes are affected by loss of hnf-4 in the two different cell types. In one case they identify a transcription factor that appears to function downstream of hnf-4 (nkx-2.2) as its elimination intensifies the effect of hnf-4 knockdown without many additional targeted genes. In another case they find a transcription factor that targets partly overlapping genes in the same cells, but clearly has other functions as well. This is interesting, and indeed had not been reported before, but it is not necessarily insightful.

To give more meaning to the supergroups, the authors appear to interpret them as developmental decision points. They state that they help explain the "regulatory logic of planarian stem cell differentiation" suggesting that the stem cells first choose a supergroup and then continue their differentiation trajectory into the different cell types. However, there is no data to support such a logic. The supergroups definitely have certain shared sets of expressed genes and chromatin accessible regions, but that could just as well be the result of using similar functional modules in completely independently developed cell lineages. For example the genes involved in the construction of cilia (as mentioned by the authors themselves), or the machinery for phagocytosis (which could well be the case for hnf-4), or for protein secretion, can be highly active in multiple independent cell types, without a shared lineage relationship. This doesn't mean that such modules are not interesting, but there is no evidence that they have anything to do with the logic of stem cell differentiation. The authors acknowledge that there could be other reasons for the gene similarities than developmental lineage, but seem to discard such explanations based on the fact that the cells they name "goblet cells" are also affected by hnf-4, and that goblet cells in vertebrates are thought to be exocytotic rather than phagocytotic, but 1. it is not known whether these cells really are homologous to goblet cells, and 2. it is not known whether planarian goblet cells would not have a phagocytic activity. Interpretation of the supergroups as developmental transition states would require the identification of these intermediate stages. Further, the interpretation of these supergroups as developmental stages conflicts with the conclusions in the recent paper by King et al. describing the planarian transcription factors in the neoblasts. It would be helpful to comment on why these studies reach different conclusions concerning neoblast differentiation.

The identification of "influential factors" is interesting, but the interpretation needs to be more cautious. The analysis assumes that transcription factors that undergo large changes in expression level on the RNA level and whose predicted binding site (based on best matches to mammalian transcription factors) is common among altered genes, are the master regulators. This however doesn't have to be the case. Some binding sites are just far more common than others (and in an AT-rich genome such as Schmidtea AT-rich motifs are very common). Further, some transcription factors may achieve effects by small fold changes, and they may regulate only a few other transcription factors who then do the majority of the legwork. Additionally, changes on RNA level are still a step away from protein levels. Combined, this makes that there is a lot of uncertainty in these analyses. This again doesn't mean that the identified transcription factors are not interesting, but their interpretation should be clarified and softened, and showing them as influential factors requires experimental verification.

To experimentally address the second supergroup, the authors include a knockdown of alx3-1 and show that this results in the reduction of staining by a probe matching a predicted secretory cell transcript. They interpret this as evidence that alx3-1 has a role "in the maintenance of secretory cells" (line 344). This is not shown though. It is unknown whether the referenced secretory cells are gone, or whether they just express less of this transcript. It is a common mistake in the field to take the loss of a marker for the loss of a cell type, but this is of course not at all guaranteed. Additionally, there is no evidence that the maintenance of these cells rather than the generation or the gene expression would be affected.

Overall, I think the authors are on the right track, and the manuscript is definitely improved, but the remaining issues should still be addressed.

Minor:

line 33-34

“showcasing that the combination of single cell methods and perturbational studies will be key for characterising GRNs widely”

I am not sure that this can count as a new insight. It would be helpful if the authors can better define what the readers should take away from this manuscript.

line 65-66

It is unclear to me why these predictive approaches would not scale to whole organisms.

line 85-86

“We identified key transcription factors involved in the differentiation of all major cell lineages derived from planarian stem cells.”

This is not shown - only predicted.

lines 301-307

This section mentions Fox-F1 several times, but I don't find it in the referenced figures (Figure 4B and Supplementary Figure 13).

The legends for many of the figure panels are still too brief to fully understand what is shown in the plots.

(Remarks on code availability)

Reviewer #4

(Remarks to the Author)

In the revised manuscript by Pérez-Posada et al, the authors have improved the presentation of their data and the rigor of their analyses. They have addressed most of our minor points, with one significant exception that greatly impacts their ability to address our major points. Specifically, we requested see a UMAP showing the expression of *hnf4* in control versus *hnf4*(RNAi) animals, as we were curious to know if some clusters of cells were affected by the RNAi depletion more than others. Yet in the revision, the authors show that *hnf4* is not reduced; in fact, it increases in expression. The authors provide some theoretical explanations for why this may be and point to other published datasets that also show lack of target transcript knockdown after RNAi (indirect effects that lead to upregulation of *hnf4*, possible detection of dsRNA in the scRNA-seq, varied knockdown efficiency across cell types, etc.). However, these explanations are not particularly satisfactory. First, scRNA-seq should allow for the detection of differential knockdown between cell types. It is true that TFs are often lowly expressed, but *hnf4* expression does not appear to be at the lower end of detection (in this data or other datasets). Second, the tracks showing RNA-seq read mapping to the *hnf4* locus does not convincingly suggest that the increase in *hnf4* transcript in *hnf4*(RNAi) worms is due to an increase in unprocessed nuclear mRNAs. The pattern of RNA-seq reads is very similar in the *gfp*(RNAi) control track, including within introns; the signal is just lower overall in the control *gfp*(RNAi) RNA-seq alignment track. Moreover, this data does not provide any real evidence that HNF4 protein levels are reduced as a result.

Further, knockdown of *hnf4* is the foundation of many experiments of this paper, including those the authors argue make it an important contribution to the field. Stating that RNAi in planarians often does not lead to the depletion of specific target transcripts in other publications is concerning, not reassuring of the conclusions of this paper. Although it may be true that the planarian field should study the dynamics and mechanism of RNAi in this system, the experiments in this paper still assume that injecting dsRNA of *hnf4* sequence will lead to specific *hnf4* transcript and protein depletion. If the authors believe there is an initial knockdown of *hnf4* that subsequently leads to its upregulation through an unknown feedback loop, this can be experimentally tested by performing qPCR for *hnf4* at multiple time points after RNAi (using bulk RNA isolated from control and *hnf4*(RNAi) worms). The authors can also take care to design their qPCR primers so that they do not overlap with their *hnf4* dsRNA construct, a precaution to be sure their qPCR assay is not detecting remnants of the dsRNA treatment.

Minor point:

In figure 1D, the authors label a gene marker for each annotated cell type using the gene ID. It would be helpful to also add the description/name of this gene if it is highly conserved.

(Remarks on code availability)

The code appears to be well-organized and clear to understand.

Version 2:

Reviewer comments:

Reviewer #2

(Remarks to the Author)

I appreciate the clarifications by the authors.

Given the clarifications, the title should be changed to "Regulatory networks of planarian stem cell differentiation". The concept of logic infers something more comprehensive and mechanistic where the models would explain the decisions made by the stem cells. The authors have clarified that they do not intent to claim the existence of progressive lineage decisions, and that the identification of "influential factors" does not imply that these factors are drivers in the lineages. The order, interaction, and regulatory impact of the various transcription factors thus remains unknown. What the authors do show however is the expression of these transcription factors in the different cell populations, and potential connections to target genes. What they describe therefore are networks.

With this clarification I think the manuscript will be a valuable resource for the community.

(Remarks on code availability)

Code is legible and well organized.

Reviewer #4

(Remarks to the Author)

The authors of this manuscript have generated many analyses in an attempt to answer a concern I raised in my last review about the effectiveness of their *hnf4* knockdown by RNAi. However, although these data suggest reasonable hypotheses for why they do not detect loss of *hnf4* expression, they do not sufficiently resolve this concern.

I agree that one should not assume a direct relationship between the levels of a specific mRNA/transcript and the levels of its translated protein. I also agree that an assay measuring TF binding, such as ChIP with a TF-specific antibody, would be the most direct way to assess whether *hnf4*(RNAi) is having the desired effect. However, as the authors point out, this is a challenging experiment to do in this organisms given the lack of epitope conservation (making it unlikely that commercial antibodies for TFs will work) and transgenics (to add an epitope tag to the TF of interest), which is why I did not suggest that assay in my previous review despite my concerns. Yet the authors are not addressing the key point: they used RNAi with the goal of depleting HNF4 so that they could assess the effects of its loss on differentiation. The burden thus lies with them to show that the first step of this experimental paradigm was achieved.

Yes, the downstream analyses showing that most DEGs in *hnf4*(RNAi) worms are detected in those cells enriched for *hnf4* expression and often have HNF4 binding motifs supports the hypothesis that loss of HNF4 led to their reduced expression. Yet an alternative hypothesis is that these results were caused by depletion of an off-target mRNA. That scenario may seem highly unlikely, yet there is precedence for such "phenotypically significant off-target effects" in the literature, including when using long dsRNA to trigger RNAi (Ma et al Nature 2006). As suggested in this paper, and considered standard practice in many models that use RNAi, a better way to validate target specificity is by using multiple non-overlapping dsRNA constructs to induce RNAi. Confirmation that both constructs induce the same phenotypic effects, including changes in target gene expression, would support the assertion that the effects observed are due to the common target i.e, *hnf4*. In addition, as suggested in my prior review, another assay that would test several of the hypotheses suggested by the authors (detection of dsRNA itself, up-regulation of pre mRNAs) would be to use qPCR with carefully designed primers that do not match the dsRNA construct and span exon boundaries.

The above discussion raises issues of technical and experimental rigor that are important and should be addressed, but it does not resolve the larger issue with this manuscript that we raised in our first review. The authors generate several new genomic datasets, yet it remains unclear how their newly generated data (e.g., scATAC-seq) advances the field. There are opportunities for uncovering something new here, yet they are not well explored. For example, in Figure 5 the authors nicely summarize how each dataset (scRNA-seq, scATAC-seq) classifies cells and then how the more complex analysis of TF "influence" classifies them. Yet the experiments that follow do not test which of these schemas is most supported. Such experiments could add significant support for their use of scATAC and analysis methods, advancing the field.

(Remarks on code availability)

General comment to all reviewers

We would like to thank the reviewers for their thoughtful and constructive feedback. In particular, we appreciate the emphasis on the need for a clearer central message, broader validation beyond a single transcription factor, and a more coherent logical flow. In response, we have revised our analyses, added new experiments, streamlined the paper, and expanded the discussion.

The main message of the revised manuscript is the identification of several supergroups of differentiated planarian cells which follow a common regulatory logic.

These include *alx3-1*⁺ cells, encompassing muscle, neuronal, and secretory cell types, and *hnf4*⁺ cells, which include gut phagocytes, goblet cells, and parenchymal cells. This conclusion is supported by an integrated analysis of scRNA-seq, scATAC-seq, and regulatory inference using ANANSE. The super grouping is evident in Figures 2, 3, and 4 and is further substantiated in the newly added Figure 5.

We have completely reanalysed our scATAC-seq data, particularly the WGCNA analysis,

following the reviewers' suggestions. In the original submission, we based WGCNA on regions identified through differential accessibility. Based on the feedback, we now apply WGCNA on the full dataset, retrieving co-accessibility modules that further align with those from the scRNA-seq data. This revised, lenient analysis improves both the structure and interpretability of the manuscript, with parallel designs in Figures 2 and 3.

We have also strengthened the validation of the ANANSE-inferred networks by reanalysing multiple published RNAi datasets. These comparisons show agreement between our predicted transcriptional targets and experimentally observed differentially expressed genes across a variety of transcription factors.

To further explore gene regulatory relationships, **we extended our single cell multiplex analysis of *hnf4*(RNAi) animals by conducting double RNAi knockdowns.** This allowed us to investigate interactions between transcription factors affecting gut phagocytes and parenchymal cells.

The revised manuscript now follows a clearer and more logical structure. We begin with scRNA-seq analysis, followed by a parallel analysis of scATAC-seq data. We then integrate both datasets using ANANSE to infer transcription factor influence, which we validate by reanalysing published RNAi datasets. Based on this integrated approach, we define the *alx3-1*⁺ and *hnf4*⁺ super-groups. We revisit previously published *alx3-1* RNAi data and identify new roles for *alx3-1* in secretory cells, in addition to its known roles in muscle and neuronal populations. We confirm this with new knockdown experiments followed by *in situ* hybridisation. We then test the role of *hnf4* in parenchymal cells using single cell multiplexing with biological replicates, showing that *hnf4* affects both cell abundance and gene expression in these cells. Finally, we explore the broader transcriptional network with double knockdown experiments targeting additional transcription factors.

In the revised discussion, we introduce a new figure summarising our findings. This schematic integrates cell types, transcript groups, chromatin accessibility, and transcription factor

associations. Based on these data, we hypothesise that sigma and gamma neoblasts are likely progenitors of the *alx3-1*⁺ and *hnf4*⁺ super-groups, respectively.

To guide the reviewers through our revised manuscript, here is a **Figure-by-figure summary of changes**:

Figure 1: Most panels remain as in the original version; we moved one panel to Supplementary Figure 5. We added some text and cartoons to improve legibility.

Figure 2: We replaced most panels for simplified versions to improve legibility. Figure 2A is a heatmap showing a subset of modules of co-expression, with the complete set of modules in Supplementary Figure 8. Figure 2B is a heatmap showing expression of relevant TFs. Figure 2C shows the agreement between TF connectivity with modules and motif enrichment in the promoters of the genes from those modules. Figure 2D is the same network of modules as before, with small changes to improve readability.

Figure 3: New Figure with similar design to Figure 2 for legibility. Figure 3A is a heatmap showing a subset of modules of co-accessibility from a new, more lenient analysis. Figure 3B shows motif enrichment of some motifs in these modules of co-accessibility. Figure 3C shows the agreement between the expression of some TFs, their connectivity to modules of co-accessibility, and their motif enrichment in the OCRs from those modules. Original Figure 3 can be found at Supplementary File 11.

Figure 4: We replaced most panels. The cartoon from Figure 4A has been expanded. Figure 4B is now the original Figure 4M. Network panels from the previous version of the manuscript can be found at Supplementary Figure 14. Panels C-I are new. Panels C-F show the subcircuits of relevant TFs from the literature, and the prediction score of their downregulated genes from knockdown works in the literature, which serve as orthogonal validation. Panels C-I show *hnf4* and *alx3-1* in the influence networks of several cell fates.

Figure 5: New Figure. Panels A-C show cell type similarity from our co-expression (A), co-accessibility (B), and TF influence (C) data. Panels D-F show the re-analysis of the *alx3-1* knockdown by Akheralie *et al.*, 2023¹, and our leveraging with our network data as orthogonal validation. We validate its influence in another broad cell type, secretory cells, via knockdown and whole mount *in situ* hybridisation (WISH) in panel G.

Figure 6 (previously Figure 5): Most panels remain unchanged. Plots for the epidermal and parenchymal scores have been added to panel D.

Figure 7 (previously Figure 6): Most panels remain unchanged. Panels of expression and connectivity of candidate *hnf4* partners were moved to Supplementary Figure 13.

Figure 8: New Figure, showing our analyses of double knock-down experiments of *hnf4* with candidate partner TFs for phagocytes (*nkx2-2*) and parenchymal cells (*foxF-1*).

Figure 9: New Figure. Schematic diagram of the main findings and associated hypotheses.

REVIEWER COMMENTS

Reviewer #1 (Remarks to the Author):

Summary

This manuscript describes the characterization and integrated analysis of snATACseq and scRNAseq from *Schmidtea mediterranea*. Some of the data are new, and some are re-analysis and integration of previously published data. The analysis generated predicted regulons of TFs, and the authors selected *hnf4* for validation via RNAi followed by scRNAseq.

Major points:

1. The primary objective of the manuscript is not clear. The abstract and introduction imply that the study aims to develop generalizable methods to address challenges such as integrating perturbational data into single-cell data and/or constructing accurate gene regulatory networks for multicellular organisms. However, in the Results, the paper immediately shifts diving into designing experiments for sequencing large amounts of high-throughput data in multicellular organisms and characterizing TF influences across cell types. There is no integration of perturbation data other than differential gene analysis performed at the end. Even though *hnf4*'s effect on neoblast differentiation is prospectively tested, the absence of other TFs conflicts with the claim in the abstract that the goal is to validate regulatory logic for the organism as a whole.

We thank the reviewer for this suggestion, as it has greatly enhanced the scope of the manuscript. In the revised version we further integrate our data with knockdown data from different factors, both by generating new knockdown experiments and by reanalysing previously published datasets. We have performed a combinatorial knockdown experiment with *hnf4* and two of our suspected TF co-factors, or partners, driving stem cell differentiation into phagocytes (*nkx2-2*) and/or parenchyma (*foxF-1*). Our analyses hint at a potential combinatorial effect of *hnf4* and *nkx2-2*, but not *hnf4* and *foxF-1*. We believe that our work investigating the combinatorial effects of multiple TFs provides a roadmap for future studies. In addition to this, we have analysed public data from the planarian literature involving knock-down of TFs and have retrieved similar results. As a product of this, our revised version contains more validation by perturbational experiments. Importantly, in the revised version, we outline two major super-groups of differentiated cell types (the *alx3-1+* and the *hnf4+* lineages). We generate new data on the *hnf4+* cells (containing both parenchymal and gut phagocytes) that adds up to the originally presented data and reanalyse knockdown data of other transcription factors. Notably, this includes (among others) the TF *alx3-1*, which regulates muscle, neurons and secretory cells. Thus, our revised version contains KD analyses and reanalyses of TFs regulating muscle, neurons, secretory cells, gut phagocytes and parenchymal cells, and we think that our manuscript now covers the regulatory logic of the organism as a whole.

2. Figure labeling has multiple mistakes, and it is confusing for understanding, here are some examples:

- Figure 1

- o Page 3, line 120: the cited figure by its description does not match 1F.

We have updated this as we moved those plots to **Supplementary Figure 5**

- o Page 3, line 121: Figure 1G is mentioned but is it missing in figure page.

We have corrected this to Figure 1E.

- Figure 2

- o Page 3, line 176: the cited figure by its description does not match 2C.

- o Page 2, line 181: the cited figure by its description does not match 2D.

We have re-written this section and simplified Figure 2, following general suggestions.

- Figure 5o Page 8, line 350- 352: the cited figure by its description does not match 5E.

Corrected to Fig. 5D.

- o Page 8, line 356: the cited figure by its description does not match 5F.

Corrected to Fig. 5E.

- o Page 8, line 356- 358: the cited figure by its description does not match 5G.

Corrected to Fig. 5F.

3. Integration of ATAC and RNA data from different cells makes the assumption that transcriptomic state is well-predicted by chromatin accessibility. But this assumption is not necessarily true, especially when the system is differentiating pluripotent stem cells. The extent to which this undermines the network/GRN reconstruction needs to be addressed.

We thank the reviewer for their comments. It is true that the method we chose for integration assumes that transcriptomic state can be predicted from the chromatin accessibility, but this is an assumption already made in the original article that published this method ² and in most other studies that used it afterwards.

Of note, all of our scRNA-seq clusters belonging to previously known broad lineages receive most integration anchors from a single scATAC-seq cluster; in other words, all thirteen transcriptomic cell clusters of neurons are integrated with chromatin cell cluster '0', all seven transcriptomic cell clusters of secretory cells are integrated with chromatin cell cluster '7', and so on. This notion is consistent with the assumption that transcriptomic and chromatin accessibility are highly correlated. Otherwise, in the absence of such correlation, one could expect that the thirteen transcriptomic neuron clusters would have integration anchors with all chromatin clusters, instead of only cluster '0', and the seven transcriptomic secretory clusters would have integration anchors with all chromatin clusters, instead of only cluster '7'. This point applies also to all other chromatin clusters, including epidermal clusters, gut

phagocyte clusters, muscle clusters, etcetera, strongly suggesting that our integration makes sense and indeed transcriptomic states and chromatin accessibility states are positively correlated. Moreover, our findings pertaining neoblasts and their chromatin state align well with independent findings from tissue fractionation experiments (see Response to Reviewer #2). Ultimately, multimodal scRNA/ATAC-seq datasets with both data modalities on the same cells will be key to determining if this notion is true, but our data strongly supports it. We have expanded our Discussion to mention this issue as a potential drawback in network reconstruction.

Minor points:

Data is not available.

At the time of writing this response, raw data is available at GEO accession number GSE274286.

Animal sizes were randomly selected. Are these animals in similar ages? Size difference could indicate age difference, adding another independent variable to the experiment.

For this study, all libraries were generated from asexual *Schmidtea mediterranea* worms. These animals only reproduce by fission, resulting in two or more smaller-size clonal individuals. Planarians can also grow larger or shrink depending on feeding conditions. Therefore, different sizes here do not correspond to differences in age. The cell type differences in animals of different sizes are well understood, as we published recently (Emili *et al* Science Advances 2025³).

Page 3, line 108-110: For identifying scATAC-seq cell types, based on Supplementary Figure 3 and the Methods, it appears the author utilized all scRNA-seq data, including those treated with RNAi for *hnf4*. Could this influence the cell type identification for the ATAC-seq data, given that the scATAC-seq data were derived from cells without RNAi treatment for *hnf4*?

We thank the reviewer for raising this concern. We wanted to clarify that we only used the cells injected with *gfp* dsRNA, which are used as a control. Thus, cells from the *hnf4* knockdown treatment were never incorporated into any of the analyses shown in Figure 1 to 5. We included control cells under the reasoning that these cells do not have an effect in gene expression due to the perturbation of any gene in their genome, as *gfp* is not a gene present in planarians. Indeed, we did not retrieve any cluster that had been previously unknown or clearly uncharacterised in this dataset. Likewise, we argue that our transfer of identities is reverse –we transfer the scATAC cluster identity to the scRNA dataset due to the former having lower resolution than the latter, which together with our observation of no uncharacterised clusters, renders the possibility raised by the reviewer to a minimum. Nevertheless, to fully address the reviewer’s concern, we sought to investigate the question of whether using these libraries might have impacted the detection of scATAC-seq clusters. We have repeated the single cell analyses of the scRNA-seq without using the *gfp(RNAi)* cells from libraries 11.3 and 11.4 and retrieved clusters in a similar manner as before. We proceeded to transfer the identity of scATAC clusters to this non-*gfp(RNAi)* scRNA dataset

as explained in Methods and retrieved identical associations between scATAC clusters (detected by Seurat) and the cell type identities of Seurat clusters detected in the non-*gfp(RNAi)* dataset, in turn retrieved from the allometry dataset publicly available. With this analysis, we are confident that *gfp(RNAi)* cells did not impact the assignment of cell identities to the scATAC data.

Figure 1E contains blocks that are too small to clearly distinguish their colors, and the row labels are confusing. Additionally, the figure caption lacks a detailed explanation that would help to understand Figure 1E.

We have changed Figure 1 and moved the panels showing the genome tracks to a new supplementary Figure. Thus, Figure 1 retains the main message, i.e., that our scRNA-seq and scATAC-seq dataset reveals many genes that are specifically expressed in one broad cell type category (scRNA) with the promoter or other associated nearby chromatin regions specifically open as well (scATAC). The detail of this is now shown in Supplementary Figure 5. We have included further explanations in both the main Figure legend and the Supplementary Figure legend.

Consider putting Figure 2A and 2B in supplementary figures. These figures provided too much information and not all information is discussed in the result. You could replace Figure 2A by selecting only the relevant modules or parts of the modules. Similarly, replace Figure 2B by focusing on a select group of TFs, as was done in Figure 2C. Later, in page 5, line 197-199, the claim of differentially accessible OCRs and genes have similar motif enrichment in will be easier to observe.

We thank the reviewer for this comment. We have revised Figures 2 and 3 accordingly. As part of this, we have moved Fig.2A,2B to Supplementary Figures, and focused only on the relevant modules as suggested. Those panels can be now found at Supplementary Figure 8.

Page 4, line 169-171: The author claims that TFs with high TF/module connectivity also show high motif/module enrichment. However, the provided Figure 2C and 2D does not strongly support this claim, though the author highlights some examples that do demonstrate the association. A counterexample is the GATGTG box and its corresponding TF mitfl-1, which does not exhibit high motif connectivity in modules m37, m45, m23, m46, m32, m41, and m42. GATGTG box has high TF/module connectivity but low motif/module enrichment in these modules. A statistical analysis, such as matrix correlation or element-wise comparison, would help determine whether an association between TF/module connectivity and motif/module enrichment truly exists.

We have correlated the matrices of TFs connectivity and motif enrichment throughout modules, which showcase how motif occurrence tends to agree with connectivity of TFs associated to those same motifs. This figure replaces former Figure 2C and can be found in Supplementary Figure 8C. In Figure 2C, we now focus on the agreement between TF connectivity and motif enrichment for selected relevant TFs and more can be found in Supplementary 8D. We have also reworded this section to better convey our message.

Page 5, line 194-196: The author mentions splitting the cells into two pseudoreplicates, but Figure 3A does not clearly illustrate the splitting process or the subsequent analysis involving these pseudoreplicates. Additionally, no rationale is provided for why two pseudoreplicates were used instead of simply performing two differential chromatin accessibility analyses on the original dataset.

We used two pseudo replicates because the dataset did not have any replicates. In the context of current single cell experiment technologies and costs, replicates are still challenging and costly to generate. Our group has strived to generalise the use of replicates for scRNA-seq, and we envision that scATAC-seq will follow suit. To justify the use of pseudo-replicates, we follow the rationale presented by Hafemeister & Halbritter, 2023⁴ where they argue that differential expression analyses on single cell data can be performed using pseudo-bulk pseudo replicates. We have put this analysis as supplementary and decided to proceed with an all-peaks, correlational procedure similar to what we did in Figure 2 for the scRNA data.

Page 5, line 197-199: the author claim that differentially accessible OCRs is similar to WGCNA however Figure 3B is not supporting this claim. Figure 3B show cell motif enrichment analysis for a list of TFs in different cell types. This list of TFs is different in Figure 2C. Also, the cell types are different then in transcriptome analysis.

We have revised Figure 3 entirely, by performing WGCNA on a less stringent set of OCRs, i.e. not only those that we deemed differentially accessible. In a similar manner to new Figure 2C, we now have evaluated the agreement between TF connectivity to OCR modules, and motif enrichment in OCR modules, some of which are the same TFs as in the co-expression analyses in Figure 2. Likewise, motif enrichment of OCR modules has larger agreement now. And in addition to that, we have improved the estimation of the agreement between co-expression and co-accessibility modules by quantifying and testing the enrichment of co-occurrences of a given co-expression and co-accessibility module based on module membership of OCRs and the module membership of their nearby genes. This

can be found in Supplementary Figure 10F,G. Even if the cell types are “different”, it is a matter of resolution –all neuronal populations can still be pooled as neuronal cell types. Despite having the same exact cell type resolution across datasets is desirable, there is no orphan cell type in the scRNA-Seq data that cannot be tied to a broader cell type from the scATAC-Seq. We have deepened our discussion of the shortcomings of the scATAC-Seq resolution in the Discussion.

Page 5, line 205-206: Supplementary 8E does not have good figure caption and axis label. It is confusing to read. Similarly, Supplementary Figure 9A as well.

We have re-arranged the Supplementary Figures, and former Supplementary Figure 8 panels are now found in Supplementary Figure 4 and Supplementary File 11. We have increased readability by rewording the figure legend and adding labels.

Page 5, line 216-220: The author used differentially accessible OCRs as input for the WGCNA analysis, while using all gene expressions for the transcriptomic analysis in WGCNA. What is the rationale for using only selected OCRs for WGCNA analysis rather than all OCRs? Additionally, how can the similar cell type combinations between the gene expression and OCRs WGCNA analyses be justified, given the differences in input?

We thank the reviewer for this question, as it has greatly enhanced the manuscript. We originally reasoned that performing WGCNA on a set of curated OCRs could compensate the lower resolution of this dataset, and for that reason we sought to perform differential accessibility analysis to improve our detection of OCRs open at specific cell types -or open in at least one differentiated cell type compared to neoblasts. To address the reviewer’s question, we decided to run WGCNA using all the OCRs we retrieved from the pseudo-bulk matrix. Using 14397 OCRs, we detected 67 modules of co-accessibility across multiple cell types and subsequently identified motif enrichment signatures across these modules which are in further agreement with our TF/co-expression observations from Figure 2. These results do not disagree with our initial, more stringent approach using only differentially accessible OCRs. As a result, we have replaced our original analysis with this less stringent analysis, while keeping the original analysis as a Supplementary Note 3 and Supplementary File 11. These new results are presented in the new Figure 3 and Supplementary Figure 10.

We justify the similarities between gene expression and chromatin accessibility WGCNA analyses because the differences in input are due to resolution (e.g. having neuronal populations versus classifying them all as ‘neurons’), and not due to entirely different cell types. Therefore, cell type similarity, on a broad sense, can still be observed in the co-expression data (see for example Figure 1D, where we see modules from different cell types, share common traits such as similar motif enrichment, or TF connectivity).

The modules of expression and OCR have the same naming system (m module or s module). It is confusing sometimes when referred in the article. The author should give different names to module of expression and OCR, such as me/se for expression, mo/so for OCR. This unclear naming systems add extra difficulty to understand the analysis done for making connections between gene expression WGCNA analysis and OCRs WGCNA analysis and corresponding Figure 3G and Supplementary Figure 9G.

We agree with the reviewer on this issue. We have re-named all co-expression modules as sE/mE, and sO/mO for co-accessibility OCR modules, as suggested. In addition, to better explain and show the agreement between co-expression and co-accessibility, we have now included an enrichment analysis testing the co-occurrence of pairs of co-expression and co-accessibility modules based on the module membership of OCRs and their closest-associated genes (see Supplementary Figure 10F,G). We thank the reviewer for this suggestion.

Page 7, line 285-298: The author chose *hnf4* as the primary TF for further analysis, but there is no discussion of the exclusion criteria for other interesting multi-cell-type influential TFs. Upon closer examination of Figure 4M, several other TFs—such as *mxipl*, *gtf2a1*, and *mef2-2*—also appear to influence multiple cell fates and align more closely with the idea of regulating differentiation from neoblasts to other cell types. *Hox4b* and *est-3* also have high influence scores across multiple cell types, similar to *hnf4*. In a hypothesis free style such as this manuscript, the author should discuss why was *hnf4* chosen over these other candidates with the analysis result.

We thank the reviewer for this suggestion, which goes in line with similar comments from other reviewers, as well as with the initial suggestion of performing experiments on more TFs. We argue that *hnf4* posed an ideal scenario as it has been formerly described in gut phagocytes, but not in parenchymal cells, thus we had a good measure for a control situation of perturbing its role in knockdown animals. Likewise, we found *hnf4* to be a clear marker for the supergroup of phagocytes, parenchyma, and basal/goblet cells.

In addition to this, we have also assessed the predictive power of ANANSE using another TF from the neurons-muscle-secretory supergroup of cells, the TF *alx3-1*. While *alx3-1* has been described as important for neuron maintenance and in muscle cells, we show that *alx3-1* knockdown decreases the population of newly differentiated secretory cells (using in situ hybridisation of a predicted target of *alx3-1* in secretory cells, the gene h1SMcG0000140). We refer the reviewer to new Figure 5.

Page 8, line 226-227: the previous study (Emili et al., 2023) did not include *hnf4*-silenced (aberrant) phagocyte progenitor cell groups. How did the author characterize this group? What additional considerations were made when identifying gene markers for this group during cell type transfer?

We named this cell type based on the transfer label analysis, where this was the only cluster receiving labels from two distinct broad types: neoblasts and phagocyte progenitors (Supplementary Figure 13B).

Page 8, line 345-353: To determine whether silencing *hnf4* halts all differentiation or halts specifically targets phagocyte differentiation, the author should conduct the same gene score analysis on epidermis and parenchymal cells across the control/B1, control/B2, *hnf4*/B1, and *hnf4*/B2 cell groups. If the gene scores for all four cell groups in both the epidermis and parenchymal cells indicate differentiation, it can be concluded that differentiation is not universally silenced across all cell types. Additionally, the neoblast score graph shows minimal variation between neoblasts and other cells. A statistical test is recommended to confirm whether the gene scores effectively differentiate cell types.

Regarding the epidermis and parenchymal score, we have performed this analysis, and it is now part of Figure 6, discarding that silencing *hnf4* halts all differentiation. These plots are also reproduced here:

We have also analysed these statistically with a Wilcoxon test, the p-values are represented here:

The main points from this analysis are:

The epidermis score does not change much in any of the comparisons, supporting the idea that *hnf4* RNAi does not interfere with epidermal differentiation, which was the referee's concern.

The parenchymal score indeed shows significant changes, and this is to be expected as *hnf4* indeed interferes with parenchymal cell differentiation. The dynamic is somewhat complex due to the following two facts: a) the phagocyte progenitors from the *hnf4* RNAi sample cluster out in their own cluster, but the parenchymal cells from the *hnf4* RNAi sample are indeed clustered together and b) phagocytes and parenchymal cells share many

transcripts (which is one of the main points of our manuscript) and therefore these scores also share transcripts. Because of this, the parenchyma score changes a lot in the phagocyte progenitor cluster (as it shares transcripts, and this cluster is almost exclusively made of cells from the *hnf4* RNAi sample) and then it also changes in phagocytes and *pgrn+* cells (to a lower extent, as these clusters have both control and *hnf4* RNAi samples mixed).

The neoblast score changes are possibly due to the fact neoblasts express a number of genes in common with other differentiated cell types. This is the case of, for example, orthologues of helicase DDX4 (h1SMcG16303), diphosphate reductase RRM1 (h1SMcG0001110), and mRNA stabiliser YBX2 (h1SMcG0013548), shown below (scale min=0 counts, max=10+ counts).

As part of our re-analyses, we revisited the gene score from former Figure 5 and, although it overall does not change, we noticed the version included in our first submission was from a previous iteration; we apologise for that oversight. We have replaced the plot in former Figure 5 (now Figure 6) and have added the epidermal and parenchymal score (see also here above).

Page 9, line 395- 396: interaction score calculation is not discussed in the manuscript. Is the interaction score calculating how in silico *hnf4* is connected to in vivo DEGs? Without detailed description, Figure 6F hardly supporting the claim that in silico *hnf4* regulating genes are overlapping with in vivo DEGs in phagocytes. Showing a simple overlap between the two group of genes in a Venn diagram format could straightforwardly prove the claim.

When creating the predicted network of TFs and target genes, the interaction score is calculated by ANANSE by leveraging similarity of gene expression, chromatin accessibility in the vicinity of the locus of the target gene, distance of these regions of open chromatin, and enrichment of the motif of the TF of interest in those regions. This is done for every TF and every other gene, including non-TF genes and other TFs. Therefore, ANANSE does not take into account any information as to whether a gene is differentially expressed or not. We have rephrased it to explain it better. Because ANANSE works by assigning a score to every possible interaction of a given TF with every other gene, we believe a Venn diagram would not properly represent this association. We have included it here nevertheless for the reviewer to see. We have added a logistic regression showing a positive trend of having a

higher prediction score and being differentially regulated. We have updated the discussion to expand on the caveats and limitations, especially those pertaining the low resolution of this data and lack of other chromatin marks that would enhance the GRN inference.

"Thus, cell differentiation comprises the combined expression of TFs and the combined accessibility of open chromatin regions (OCRs) acting as cis-regulatory element (CREs). This combination creates a 'regulatory logic' forming gene regulatory networks (GRNs) 5-7." Other sources of epigenetic information contribute to 'regulatory logic' including DNA methylation.

We agree and we have rephrased this.

"Single-cell methods have democratized the study of differentiation trajectories...". It is unclear what is meant by "democratized" here.

We meant that it has facilitated the study of differentiation trajectories with an unprecedented level of resolution and granularity which, coupled with the drop in the cost of technology, have allowed these studies to arrive to the hands of a broader range of members of the scientific community. These methods are also broadly applicable to all organisms. We have rephrased this to use the word "transform".

"Gene expression is highly pleiotropic." Taken literally, this means "there are varied causes of gene expression". Yes, this is true, but I don't think this is what the authors meant to convey.

We have reworded this sentence.

"it is the combination of genes expressed in each cell type that defines their identity." This statement is not universally accepted, many believe that function defines cell identity.

We agree that this definition is not widely accepted. We have rephrased this to express it as a possibility. We have updated the Discussion accordingly.

Is *hnf4* expression altered in the *hnf4*-RNAi worms/cells? I did not see this data in the supplemental figures.

hnf4 is actually expressed at a higher level in *hnf4*(RNAi) worms. This lack of downregulation or even upregulation at the transcript level is a common effect of RNAi gene silencing that typically goes unreported. From our reanalyses of TF KD generated by other authors, *alx3-1*, *coe*, and *p53*, are in the list of downregulated genes in their own knockdown treatment, but *soxP-3*, *pax2/5/8-1*, and *prep*, are not. Similarly, in our double knockdowns *nkx2-2* is in the DEG list after its double knockdown with *hnf4*, but *foxF-1* is not. We believe there are different possibilities that might explain this. First, TFs regulate many other genes and are typically expressed at low levels. Small, undetectable changes in their expression may trigger larger downstream effects on their targets. Second, the level of persistence of RNAi-targeted transcripts is unclear – cleaved mRNAs may remain detectable by RNA sequencing despite being translationally inactive. Third, knockdown efficiency may vary across cell types and shifts in cell type proportions could further complicate interpretation. Fourth, it is unknown whether dsRNA itself or the siRNAs that may derive from it can be reverse transcribed and detected as cDNA in RNA-seq experiments. Finally, feedback mechanisms may upregulate the targeted gene in response to knockdown. This could increase nuclear pre-mRNA levels, but since RNAi acts in the cytoplasm, these transcripts may still be degraded before translation. Altogether, our findings highlight the need for further investigation into how mRNA levels reflect knockdown efficiency, which will be important for interpreting future experiments.

We have included this information and discussion in Supplementary Note 4, the plots of *hnf4* expression in Supplementary Figure 20, and we have mentioned this in the associated text.

To further speculate about these options, we obtained genome tracks of the read mapping to the *hnf4* locus (please note the gene model is in reverse orientation):

This plot shows that there are differences in the distribution of read mapping between the control and the *hnf4(RNAi)* treatment. Knockdown samples have a) higher intronic mapping (observable in all three larger introns) and a more pronounced 5' bias, with increased mapping in the first exon (to the right). While this information is insufficient to conclude anything mechanistic, it does show that there are transcriptional differences, and it is consistent with an elevated transcription rate (elevating the pre-mRNA pool) and perhaps a differential persistence rate of the 5' piece of cleaved transcripts. As stated above, this is a question that deserves further attention, and we thank the referee for pointing that out.

Reviewer #2 (Remarks to the Author):

This manuscript by Perez-Posada et al. aims to clarify the gene regulatory networks in planarian cells. The study contains a huge amount of data and analysis, and is one of the largest single cell studies in the planarian literature. The scRNAseq data in part was previously published in the context of other studies, but part was newly generated, and the scATACseq data is all new to this study. From the analyses it appears that the data is of adequate quality, however, single cell libraries often suffer from low read depth, and this is also mentioned as a limiting factor in this study. The analyses performed are state-of-the-art and the figures are impressive. It is clear from the manuscript that the authors have put a great deal of effort into the acquisition of the data and in the computational analysis. The amount of resources and skills that this requires are not to be underestimated, and I congratulate the authors on this tour de force.

We thank the reviewer for these words, and we are glad that the work is appreciated.

However, the thing that is sorely missing from this study is a message. With all the data and all the analysis, there doesn't appear to be a novel finding or a new insight that is generated. Due to the lack of message, the manuscript lacks focus, and is difficult to digest. I appreciate that it may be hard to dig through this enormous amount of data and pick something specific to zoom in on, but without that the study remains just a heap of data. The data is complex and multidimensional, and that is to be expected of a single cell dataset, but that just means that the authors need to pre-digest the information better to make this understandable and interesting to the reader. Because there is no focus, the figures keep showing everything, making them too large and too comprehensive. In their current shape it is not possible to read the amount of detail that is in the figures and it is impossible to know what the authors want the reader to see in them. While impressive at first glance, to show everything is almost the same as to show nothing at all.

We agree that our original submission was difficult to digest, and we hope that our effort into simplifying the writing and streamlining the narrative proves successful. We also must thank the reviewer for this constructive criticism, for it helped us in finding a story within the data. Aspects of that main message were present in our original submission, but we agree with the reviewer that these got lost in the amount of information presented.

In the revised version, we have solidified findings that were present in Figures 2, 3 and 4 into a new Figure that shows the main message of the paper: the notion that at least two major super-groups of planarian differentiated cells, the *hnf4+* and the *alx3-1+* cells, exist. This is observable by independent sources of information, including RNA expression, chromatin accessibility and TF influence. This is presented in the new Figure 5. The flow of logic then shifts to validate these findings by knocking down *alx3-1* and *hnf4* and showing that they induce effects in the predicted cell types.

Figures 1-4 lay out all the data. Figures 1-3 are framed as verification of the data. These figures set the stage for the main message, so it is understandable that not much interpretation is provided. Figure 4 then contains an advanced computational analysis using a published pipeline (ANANSE) integrating all the data. This results in an impressive set of plots, and the reveal of what appear to be the GRNs that the abstract of the study refers to. However even here the interpretation is lacking. The authors find that there are multiple transcription factors involved in gene expression, and that some are cell-type specific, and some are not. The graphs suggest that some transcription factors may form networks regulating each other, but it is not clear how strong the evidence for these connections is, and none of them are experimentally verified. The meaning of these networks therefore remains unclear. Even the names of the transcription factors are difficult to read in the dense figures. After this section there is a bit of follow-up in Figures 5-6 describing the RNAi phenotype of intestinal transcription factor *hnf4*.

We agree that the figures and associated text needed improvement, and we have re-written the Results section accordingly to help the reader follow the rationale of ANANSE. We have improved our writing in the results section, which is supported by Supplementary Note 3, which we have also expanded. Connections between these TFs, as explained in Supplementary Note 3 and Results, are assigned a probability based on leveraging gene expression and chromatin data. Our original single-cell transcriptomic analysis of *hnf4* RNAi data was intended as a validation. It is true that it was only one factor, but these experiments are difficult to generate, expensive, and technically challenging, to a point that our *hnf4* single-cell transcriptomic analysis is the first of its kind in the planarian literature (not counting other preprinted work from our research group). However, we agree that a validation of one TF is suboptimal. Therefore, to address the reviewer's concern, we have reanalysed TF knockdown data from the literature (See Figure 4C-F, Figure 5, and Supplementary Figures 15-17, for example). This serves as orthogonal validation of the predictive power of our computational approach for GRN inference.

The authors use scRNAseq to find that knockdown of *hnf4* results in loss of phagocytic cells without a major effect on the neoblasts. This phenotype has however already been reported based on stainings (PMID: 38165802). While the method applied in the current manuscript is

considerably more fancy than in the previous study, it again doesn't lead to a significant new insight.

We disagree with the reviewer in this point. The referred paper (cit PMID: 38165802) does not report any effect of *hnf4* RNAi in the parenchymal cells. The significant new insight of our paper is that *hnf4* RNAi leads to both phagocyte and parenchymal cell defects, contrary to what has been reported previously (i.e. effects only in phagocytes). The effects on parenchymal cells are milder, and we suspect that other authors have failed to detect them because of that. Our detection is in fact thanks to our single cell method, that allows us to computationally dissect cell types and detect changes at the cell abundance and the gene expression level in both gut phagocytes and parenchymal cells. This new insight is key to our paper, and it was not reported in the above-mentioned paper.

The aim of the authors appears to have been to create an atlas of GRNs, somewhat similar to the single cell RNAseq atlas of planarian cell types that the senior author previously generated. Since then another study has already presented an atlas of the planarian transcription factors based on scRNAseq data (PMID: 38401119). An atlas however can probably not explain the “logic of planarian stem cell differentiation” as the title of this manuscript claims - and while a lofty goal, this manuscript does little to advance this. In fact, the stem cells are hardly discussed in the entire manuscript. The scATACseq data seems challenging (based on the sparse information in the heatmaps shown in the supplementary figures) but even without that data it would have been possible to generate an interesting story. There are many potentially interesting observations that could be the starting point for a new insight if they would have been followed up on. The authors for example point out that previous studies have only identified gene expression clusters that correspond with cell clusters, and that they for the first time identify expression modules that are shared by multiple cell types. Further analysis of these gene sets and their relevance could generate new insights in the functioning of planarian cell types and in the use of conserved gene modules. This would however require significant further work.

We thank the reviewer for this comment, as it encouraged us to better define the story and scope of the manuscript. We hope that our new discussion figure (Figure 9) helps to convey our message in a clearer way. We understand the “regulatory logic” as the combination of TFs, OCRs and their respective binding to regulate gene transcription. So far, most planarian publications have envisioned a simple model of planarian stem cell differentiation. As a product of this notion, most schematics of planarian stem cell differentiation depict a simple model where each individual cell type is associated with individual sets of TFs. While several TFs are associated to each individual cell type, the opposite – TFs that influence several cell types – is not commonly depicted. Examples of this are present in several papers, such as Scimone et al ⁵, Molina et al ⁶, and Zhu et al ⁷. Similarly, several publications have used *hnf4* as a “gut marker”, leading to a neglect of its expression in parenchymal cells (known from the single cell atlas papers) and its role in those cell types (first shown by us in this work). As a product of this, our paper seeks to change the narrative to a more complex model where several TFs affect individual cell types, but also individual TFs affect several cell types, leading to a more complex regulatory logic, where it is the combinations of factors that regulate differentiation. Our new Figure 9 aims at capturing this notion. Our

manuscript does not claim to solve this logic completely, but we believe that it is a legitimate step forward towards this mission as it provides a roadmap for future double knockdown experiments.

Regarding King *et al.* ⁸, we commend the authors for their work, for their analysis of TFs in single cell data which clearly advances the field. This paper appeared in the final stages of our analyses and while our original manuscript was being written. We agree with the reviewer that part of the aims is overlapping. However, our study provides novel scATAC-seq data and integrates this with scRNA-seq data to predict TF influence in each broad cell type. Out of these analyses, we observe cell types that share a regulatory logic and then validate these findings by analysing new and previously existing TF knockdown bulk and single cell data. This includes describing novel roles of *alx3-1* in secretory cells and *hnf4* in parenchymal cells, components of two of the supergroups we describe. Therefore, we think that our study goes well beyond what King *et al.* reported last year.

Regarding the focus on neoblast, we want to add three comments: a) First of all, from a very conceptual point of view, differentiation is something that happens to stem cells and it is the very process that transforms them into non-stem cells. To analyse stem cell differentiation (as our title says) one must therefore look at differentiated cells. Indeed, our ANANSE influence comparisons are differentiated cells to neoblasts, as one cannot infer “differentiation” without taking into account both. b) Our original manuscript contained single cell ATAC-seq of neoblasts, and we report our finding, which are very consistent with those previously reported by Poulet and colleagues ⁹. c) In our revised version, we speculate about the connection between the neoblasts classes previously reported and our supergroups of differentiation. scATAC-seq data of enough resolution to cluster neoblast classes individually does not yet exist, but we believe that our manuscript advances the field towards this direction.

Finally, regarding the modules of co-expression, we agree with the reviewer that a deep study of gene modules specific to multiple cell types is desirable, in order to disentangle the role of many genes in planaria. But we think that, provided our aim of finding common dynamics at different levels of gene regulation across different cell types and fates, this falls outside the scope of the manuscript. A deeper dive into these gene modules, with deeper resolution and more robust methods to identify gene expression patterns across cell types, is likewise suitable for a follow-up project. Our supplementary materials provide lists of the components of these modules, and their TFs, and we trust that this will become a roadmap for future studies.

An additional problem is that the figure legends frequently are too brief to understand what exactly is shown. This makes it difficult to know how to interpret the data, and makes it hard to follow the authors when they make statements about the data.

For example:

Fig 1F. Are the rows single cells showing accessibility at the same locus used in Fig 1E, or are the rows different loci in the cluster and is the scATAC summarized for each row? In either case, why is there no TSS signature?

We thank the reviewer for pointing this out and we acknowledge that our original figure was not clear enough. Former Figure 1F, now Figure 1E, is a heatmap of accessibility of marker OCRs (open chromatin regions) across the different cells of the dataset. Rows are OCRs, cells are in columns. In former Figure 1E, now Supplementary Figure 5, we aimed at highlighting some examples of genes and nearby OCRs that are expressed, and accessible, in the very same cell types. TSS signature can now be found in Supplementary Figure 4M-U.

Fig 2D. What is an enrichment log q value? Is this a $10\log$?

q-value is defined as the minimum false discovery rate at which an observed enrichment is significant; thus, it is an adjusted p-value corrected for false discovery. We have updated our methods to better convey this.

Fig 3B. Is this focused on accessibility of promoters, or is this any ATAC peak? Also, there appears to be very little overlap in TFs between the ones in this panel and those in Fig 2C. This makes it very hard to correlate the data, even though the text states that they are similar. Better explanation of the selection of the TFs and how to interpret the (absence of) overlap would be helpful.

As part of our re-analysis of the scATAC-seq data, we re-ran the WGCNA more leniently and performed motif enrichment analysis on the OCRs from these modules of co-accessibility. As a result, there is now more agreement between our observations from Figures 2 and 3. We refer the Reviewer to those sections in the manuscript, as well as the associated Supplementary Figures 10-11, Methods, and the Supplementary Note 3, for more information. Please also see response to Reviewer 1 for a similar question about the scATAC-seq data.

Fig 3 CD. Is each dot in the scatterplot a gene, or a promoter, or an ATAC peak independent of annotation? It is odd that no neoblast-specific accessibility is found. In contrast to what the authors state, a previous study using bulk neoblast ATACseq had reported clear neoblast-specific regions of ATAC signal (PMID: 37801496 - although they found no TSS peaks and no specific TFs that would regulate these regions).

We thank the reviewer for their insight. In former Figure 3C,D, now found at Supplementary File 11 B,D, we show volcano plots resulting from the differential accessibility analysis of two types of analyses: comparing a given cell type versus the rest (one-vs-all, former Fig.3C), or comparing a differentiated cell type against neoblasts (former Fig. 3D). Every dot in these plots is, therefore, an Open Chromatin Region (OCR) with \log_2FC and $p.adjusted$ (negative log-transformed) values. When looking for differentially accessible regions only in neoblasts and not in other cell types, our analysis did not retrieve any OCR with positive \log_2FC . This does not mean that we did not find any kind of open chromatin signal in neoblasts (after all, we find neoblasts in our scATAC-Seq analysis); we rather interpret this as neoblasts not having OCRs that are more accessible than in other cell types.

There are however key differences with the analysis performed by Poulet et al ⁹. One key difference is that our analyses mentioned above compare neoblasts to all cell types, but

Poulet et al samples consisted of neoblasts, intestine, epidermis and brain. Many OCRs characteristic of specific types are also open in neoblasts. One can see this, for instance, in our Figure 3A, where many OCRs also appear open in the final grey neoblast column. In such a setting, the question is whether the neoblast against intestine, epidermis and brain comparison performed by Poulet et al could potentially have returned OCRs of muscle, secretory, parenchymal types (i.e. the cell types that they do not have). This is just to say that, beyond commending Poulet and coworkers for their analysis, there are key differences between their analysis and ours that warrant further exploration. We think that such analyses will be the subject of future studies.

Many aspects of the analysis agree. For instance, Former Supplementary Figure 4J agrees with Figure 2A from Poulet et al, the article cited by the reviewer, as we do not find accessibility enrichment around the TSS in the same way we find it for other cell types we identified in our scATAC-Seq data. The fact that the neoblast chromatin markers detected in our scATAC-Seq dataset appear accessible in other cell types, aligns with the authors' discussion on TF-based gene regulation not being the main mechanism for gene regulation in neoblasts, either because no TFs are bound or because many TFs bind, and the signal is even out. Thus, despite differences in methodology and resolution, we think that our data and results largely align with this and similar works from the literature.

Fig 3FG. It is not clear what these panels represent, and how to interpret the data. Are the same genes (rows) shown in both panels? Isn't it somewhat expected that genes that are more accessible in a tissue produce higher RNA levels in that specific tissue? Is this only intended as verification of the data?

Former Figures 3F,G show the agreement between gene expression and chromatin accessibility of OCRs identified with our differential accessibility analysis and further analysed using WGCNA. Row order is the same in both panels, but former Figure 3F is showing accessibility in OCRs, and former Figure 3G is showing gene expression in the closest gene from the same OCR as in F. It is a verification of both the data and our observations derived from the data, as we find that both the gene expression and the chromatin accessibility group phagocytes, basal/goblet, and parenchymal cells together, as well as muscle, neurons, and secretory cells. Provided this was the first time a single cell ATAC-Seq of planaria was analysed, we sought relevant to show this information. Of note that we have re-analysed the scATAC-Seq data analysis less stringently (see response to Reviewer #1) and we have re-worked Figure 3 to enhance readability. Former Figure 3 can be found as Supplementary File 11.

Fig 4. ANANSE is not a commonly known analysis tool (yet) and while the supplement gives general information about the workings of this tool, the application in this study still requires further explanation. The original tool uses enhancer regions, and TF binding sites (and motifs) that are experimentally determined in mammals. How is this translated to planarians that appear to have little enhancer activity, much higher gene density, and where no TF binding sites are known and even the planarian homologs of mammalian factors are often not unequivocal?

Supplementary Note 3 mentions that we associated TFs and candidate motifs using two methods that are based on sequence homology and automated transfer: a phylogeny-based approach that relies on OrthoFinder¹⁰ and several motif databases¹¹, and the JASPAR tool for motif assignment, which works at the protein domain architecture level and assigns a motif to a given TF via similarity regression¹². Additionally, based on former works in the literature¹³, we transferred motifs assigned to any TF in that original publication. As stated in the original publication of ANANSE, every TF is assigned not a single motif but a list of similar motifs, and the highest-scoring one is considered for every TF/target interaction on a case-by-case basis. We are aware that automated transferring involves the assumption of a certain degree of conservation across species, and this may not necessarily be the case. For this reason, most of our bespoke motif analyses are to highlight agreements between TFs and motifs at the TF class level. It is worth mentioning, though, that since ChIP/SELEX or other biochemical data is not widely available for more than a few model species, many other authors are resorting to similar approaches, with positive results. We now mention this limitation in the Discussion and even in more detail in Supplementary Note 3.

Fig 4B. unclear what this is, other than a correlation matrix - but correlation of what exactly?

Legend of Figure 4B reads: "Pearson correlation of cell types based on profile of TF centrality across networks". For every network, we took the centrality values of every TF in that network. We then pooled them together in a TF x cell type network matrix, where missing values were replaced with zero. We then correlated columns (cell types) based on those TF centrality values. The resulting correlation was used to calculate a clustering of cell types. Those Pearson correlation values are showed as a heatmap, and the hierarchical clustering tree observed in rows and columns is the resulting clustering of cell types based on TF centrality correlation. As part of our revision of Figure 4, we have included this panel, together with a graphical explanation, in Supplementary Figure 12C-E.

Fig 4 C-J. What are these strip plots? How should we interpret the values at the strip plot y axis? Why are some TFs shown here - how were they selected, and what should be deduced from the plots? Are these the GRNs that are referred to in the title? Why is there not more emphasis and explanation of these networks?

To better showcase the data present in the networks, we selected two TFs from those networks to show their interaction scores with target genes. For each strip chart, we kept the top 5% highest TF/target interactions. Every dot in these strip charts is, therefore, a gene, and its position in the Y axis is simply the interaction score value. If highlighted (i.e. slightly larger and outlined), it means that this gene has a gene name assigned in PlanMine. TFs were chosen based on their relevance in the literature because they had been previously described. An alternative way to have a look at this data is by looking at Supplementary Files 17,18, where we show every interaction (with top interaction scores) between a given TF in a network and known genes documented in PlanMine.

Fig 4L. Not sure what this means.

Former Figure 4L, which now appears in Supplementary Figure 14J, is a scatter plot of the influence score of different TFs in the phagocytes (y axis) and the parenchyma (x axis) networks. As explained in Results and Supplementary Note 3, ANANSE influence allows to compare two networks, one from a source cell type (neoblasts, in our case) and another from a target cell type (for example, phagocytes). ANANSE calculates a score of influence for each TF in the network of the target cell type, that ranges from 0 to 1 by leveraging differential gene expression between the source and the target cell type. The higher the score, the more likely the network structure of the target cell type is due to the contribution of that TF to the network. Results from ANANSE influence for all cell type networks can be seen in Supplementary Figure 13B-J.

Of note is that, if ANANSE is capturing the underlying biological relationships well, not every TF will be equally influential for all the cell types. This is why some TFs have influence 0 in this scatter plot. Interestingly, some TFs are influential in both of these cell fate networks, and this is what we aimed at showing with this scatter plot.

Fig 4M. Why are the TFs at the center of the networks in 4C-J often missing from this panel?

Because in Figure 4M we chose to showcase the top TFs from each of the groups of co-influence. This was done by calculating an average influence profile for each cluster of co-influential TFs and then correlating the co-influence profile of every TF from a given cluster against said average co-influence profile. The top five correlating TFs are the ones shown in Figure 4M. This is explained in Supplementary Note 3 and Methods, but we have reworded to explain better.

Fig 5. Much of this looks like it should have been in the supplement.

We appreciate the suggestion, but we disagree with this comment. Former Figure 5, which is now Figure 6, presents a TF knockdown in replicates, analysed by single-cell transcriptomics. This is, as of today, a very uncommon type of analysis, only preceded in the planarian literature by other papers of our group. It is important, therefore, to present it in detail. Otherwise, it would have made it look like this is a conventional analysis in the field, and this would have been confusing. Furthermore, the above-mentioned figure presents a novel finding: *hnf4* regulates parenchymal cells at the cell number level.

Fig 6, This looks like a Main Figure again, but there is no new insight and no take-away message. Why was *hnf4* selected for follow-up? What was the question that was addressed by this experiment?

We also appreciate the reviewer's encouragement to better focus the scope of the figure, but we also disagree with the comment that there is no new insight. As stated in the text, we selected *hnf4* because we retrieved independent observations of parenchyma and phagocyte cells sharing common gene regulatory dynamics. This is now better explained in the new Figure 5. Since *hnf4* is among the top influential TFs for both cell fates, and no role in parenchyma had been described prior to our work, we decided to address the validity of this prediction. The figure clearly shows that, after computational dissection of the major cell types, the effects of *hnf4* in terms of gene regulation, are observable in both gut phagocytes

(as predicted from previous papers) and parenchymal cells (a novel insight, predicted by our analysis).

We have revised this section to better convey the flow of the narrative. Together with the new Figure 5, which orthogonally validates the predicted role of another transcription factor, *alx3-1*, in muscle, neurons, and secretory cells, we hope to have conveyed a better story for this manuscript.

I won't comment on the Supplementary Figures to avoid repetition but similar points apply there.

We have revisited the most content-heavy Supplementary Figures to correct any errors detected by the reviewers or ourselves.

In summary, the authors did a fantastic job on the generation of this dataset and on the global analysis. Unfortunately a global analysis does not make for a very compelling story, and in its current shape it is not clear what readers would learn from this study - if they would even manage to read through the entire document. While it is of course entirely up to the authors to present their data in the way that they like, my recommendation would be to abandon the idea of an omnibus, and rather find a focal point that can make the data shine.

As stated above and in other responses to the other Reviewers, we have streamlined the narrative of the figures and focused on discussing the common gene regulatory dynamics for two major groups of cell types, which we have named *hnf4*⁺ group, and *alx3-1*⁺ group.

Reviewer #3 (Remarks to the Author):

Reviewer #4 (Remarks to the Author):

Summary:

In this manuscript, the authors aim to uncover how transcription factors influence the differentiation of planarian stem cells into various lineages. They use scRNA-seq and scATAC-seq to characterize the chromatin and transcriptional profiles of these cells. As in other recent papers from this group, the authors use a method of scRNA-seq they developed that optimally preserves and profiles the transcriptome of planarian cells. They do a thorough job of analyzing many thousand cells and perform experiments with biological replicates, including one in which they knock down a conserved transcription factor, *hnf4*. They also perform single cell ATAC-seq (scATAC-seq) using planarian cells, a first in the field at this time. The data in this manuscript will be a valuable resource to the field, and the analyses the authors perform have the potential to inform testable hypotheses about the roles of specific transcription factors interactions during planarian stem cell differentiation. However, the authors do not address those questions here.

They do confirm an outstanding hypothesis in the planarian literature, namely that *hnf4* regulates the differentiation of cells in the intestinal and parenchymal lineages (Forsthoefel et al 2012, Van Wolfswinkel et al 2014, Fincher et al 2018).

We thank the reviewer for their positive comments. Regarding the hypothesis of *hnf4* regulating the parenchymal cells, we want to note that Forsthoefel et al. 2012 links *hnf4* to planarian intestine, not to parenchymal cells. Van Wolfswinkel et al. 2014 links *hnf4* to gamma-neoblasts, not to parenchymal cells. In Fincher et al. 2018 (as well as other planarian single-cell atlases), the expression of *hnf4* in parenchymal cells (termed *cathepsin+* cells in Fincher *et al*) in addition to intestine is reported for the first time, but this fact is not followed up in the paper, nor the hypothesis of *hnf4* regulating parenchymal cells is formulated. The reviewer is right in pointing out that parts of this were already observed in the previous papers (as cited in our original submission) but we disagree that this was a hypothesis already formulated. Importantly, we formulate this hypothesis not only based on the expression of *hnf4* in parenchymal cells but also based on the fact that parenchymal cells and phagocytes share many other transcripts (Figure 2), open chromatin regions (Figure 3) and other influential TFs (Figure 3).

We thank the reviewer for the encouraging feedback. In line with this and other reviewers' comments, we have re-visited ours as well as public data from the literature to propose a common regulatory landscape for planarian cell differentiation, which identifies at least two supergroups: *hnf4+* cells, and *alx3-1+* cells. New Figure 5, former Figures 5 and 6 (now Fig.6,7) and new Figure 8, further delve into this question.

We think this data is worthy of publication, but suggest the authors make a better argument for why these data and analyses will advance the field. Below we list some specific questions that the data raised for us and give some suggestions about how to address them.

We thank the reviewers for this comment. Our new manuscript contains the first description of the *hnf4+* and *alx3-1+* cell type supergroups, a top-level finding that arises from our analyses in Figures 1 to 4. We therefore believe that these findings advance the field, and we have strengthened and streamlined the narrative of our paper to better convey this idea.

Major points:

1. In Figure 3, the authors show that muscle and neuronal cells have more cell type-specific open chromatin regions (OCRs). Moreover, this distinction exists without normalizing to neoblast OCRs. Can you address why this might be? Are these lineages specified from neoblasts differently? Do these lineages have more sub-classes, but the resolution of the scATAC-seq wasn't sufficient to distinguish them? Are the TFs that drive this process of certain classes? Something else?

In every planarian single-cell analysis to date the epidermis is resolved at higher resolution, including early and late progenitors, likely because it is the cell type with the fastest turnover. Our scATAC-seq dataset is of relative low resolution compared to modern scRNA-seq datasets, but earlier scRNA-seq datasets (for instance, see Wurtzel et al ¹⁴) had similar

resolution to our scATAC-seq. Furthermore, neurons and muscle cells are among the most abundant cell types in planaria, besides epidermis (which is split in three clusters). Therefore, as they are more abundant, our sampling is more likely to retrieve more reads coming from these cell types, resulting in deeper coverage and peak detection in them. We have reworded our Discussion to better explain this.

2. As the authors describe in their introduction, multiple TFs often work together to bind targets and regulate transcription synergistically. Have the authors tried any experiments in which they perform combinatorial RNAi of multiple TFs? Perhaps for TFs with weak or no obvious phenotype when knocked down singly? The analyses presented here provide a great roadmap for such experiments.

We thank the reviewer for this suggestion, as we have included a combinatorial RNAi experiment of *hnf4* with *nkx2-2*, and *hnf4* with *foxF-1*, to try and disentangle the potential partnership of these TFs with *hnf4* in the cell fate to phagocytes and parenchyma, respectively. We believe that future studies will also incorporate the combinatorial effects of multiple TFs, and agree with the reviewer that our analyses provide a roadmap for these experiments

3. Related, in Figure 4M the authors show that two “s” modules (s01 and s04) are “influenced” by TFs that also influence several other cell fates. However, an “s” module is defined as a set of marker genes that are highly enriched in only one cell type. Can you discuss what is different about the s01 and s04 modules/cell types and/or the TFs that regulate them (versus the other s modules that generally seem to be “influenced” by fate-specific TFs)?

All instances of classification into “s” and “m” modules/clusters were performed in an automated, unsupervised manner. For a given group of features (genes, OCRs, TFS) across cell types that belong to the same module, our classifier retrieves the distribution of signal (expression, accessibility, influence score) across all cell types (for example, nine distributions of co-influence for nine cell type fates), calculates a given metric to summarise each and all distributions (which can be the upper quartiles, the medians, or the means), and then counts the number of outliers of the resulting distribution made of those summarising metrics (e.g. the distribution of ten upper quartiles). If only one outlier is detected, then this module is classified as ‘s’. If more than one outlier is detected, it is classified as ‘m’. This procedure can be visualised in Supplementary Figure 7E,F and is explained in Methods and Supplementary Note 1. Like many unsupervised methods, it depends on thresholding and on the statistic. After all, the difference between “m” and “s” modules is gradual, rather than a sharp boundary, and any classification method will find grey areas hard to classify as black or white. When one looks at Supplementary Figure 11L, module ‘s01’ has a maximum, outlier upper quartile for early epidermal progenitors, but there are several other cell types with a high median or upper quartile (epidermis, muscle, neurons). This is in contrast to module ‘s02’ where there is one very clear cell type with high influence, epidermis. We aimed at using this classification as a rough estimate of which groups of TFs are co-influential, and this method successfully allowed us to identify groups of TFs co-influential in the same groups of cell types we observed in our co-expression and chromatin accessibility analyses. Future similar analyses should see a more rigorous

estimate by leveraging this and other methods (such as specificity as calculated by Yanai et al., 2005¹⁵, or other methods of unsupervised classification).

Minor points:

Figure 1D: Where available, please add gene names along with gene IDs to help the reader connect them to each lineage.

We have added gene IDs to most instances of the name of a gene. For instance, in Figures 2-3 and some parts of Figure 4. Panels from Figure 4 and related Supplementary Figures have more in-depth information available in Supplementary Files 16,19 including gene IDs. We have re-worded figure legends and Methods to better point this out.

Figure 2A: It would be helpful to label the black bars as “log number of genes” and shift the “expression” legend so it's above the data to which it refers. This would make it more immediately clear to the reader what each element represents. Or rather than showing a bar plot for the “log number of genes” you can simply label the number of genes in each module. To this end, aren't “s1-s24” and “m1-m53” gene modules not specific genes? They should then be labeled on the left as modules. This would help connect Figure 2A and 2B.

We have revised Figures 2 and 3 as a suggestion from other reviewers and hope to have helped clarify some of these points.

Figure 2E: Please add the cell type name in addition to the cell cartoon for better clarity and readability.

Done.

Figure 3: It's nice that the authors have color-coded the font according to cell type throughout the paper, but would still be helpful to include a specific legend detailing which cell type is represented by each color in the stacks, for example that one in 3F.

Figure 3 has now a colour legend for cell types which is common to all panels within the Figure.

Line 319: The authors show that *hnf4*-RNAi worms have a lethal phenotype. However, Lobo et al.,2016 reported no major change in blastema size during regeneration. Please address this finding in the text: is it an RNAi delivery issue? Or is *hnf4* dispensable for intestinal differentiation during regeneration?

Lobo *et al*¹⁶ inject dsRNA for three consecutive days like us, but cut 5h after the third injection, unlike our phenotyping experiments (and sample collection) where we cut (or collected ACME samples) 9 days after the third injection. We argue that these temporal differences may underlie the phenotypic differences, with 5h not being enough time for the lesions and phagocyte decrease to develop.

In Figure 4M: 1) why are some TFs are not labeled? Are these unknown? Then their gene ID should be listed.

We have added their gene IDs.

Figure 5C: Can authors color Cluster 17 differently here to highlight it? Additionally, it would be helpful to show a UMAP with *hnf4* transcript expression in control vs *hnf4* RNAi.

We have added a legend to better show the position of clusters 17, 14, and 11, in Figure 6C (former Figure 5C). Regarding the UMAP with *hnf4* transcript expression in control vs *hnf4* RNAi, please see response to Reviewer #1.

Line 352: Figure 5E is referenced when it should be Figure 5D.

Done, changed.

Figure 5D: In the phagocyte score plot, please add asterisks or other symbols to indicate statistically significant changes.

We have performed a Wilcoxon test to test for differences between control and knockdown cells for each score in each cluster, found at Supplementary Figure 20F. Please also see a related response to Reviewer 1 for more information.

Figure 6I: The shaded colors representing cell types/lineages are difficult to see. Also here and possibly in Figure 2 where you show this network, it could be helpful to make a gradient shadow or somehow indicate that the larger network encompassed by the green shadow includes both parenchyma and gut network nodes. Here, without the pink and green node colors the impression is that all these nodes of connectivity are in the green gut network.

We have increased the opacity of these communities (here and in Figure 2D) and have added a gradient (albeit this is detected as a single community following a graph algorithm, and not a manual classification).

Supplemental Figure 12: although the authors describe the percentages of each defect, and the timing of phenotype progression in the text, it would improve understanding for the reader to indicate these details on the figure.

We have updated this figure accordingly.

Reviewer #4 (Remarks on code availability):

There is a significant amount of code in this paper, so we did not run all of it to test reproducibility of every analysis. However, their GitHub page is well organized into logical directories and their code is appropriately annotated so that others should be able to follow the steps of their pipelines. All analyses also appear to have been performed using available packages that can be downloaded from Bioconductor or other publicly available sources.

We thank the reviewer for this feedback, as we actively strive to maintain good standards for code availability and reproducibility.

Reviewer #5 (Remarks to the Author):

References

1. Akheralie, Z., Scidmore, T.J. & Pearson, B.J. aristaless-like homeobox-3 is wound induced and promotes a low-Wnt environment required for planarian head regeneration. *Development* **150** (2023).
2. Stuart, T. *et al.* Comprehensive Integration of Single-Cell Data. *Cell* **177**, 1888-1902 e1821 (2019).
3. Emili, E. *et al.* Allometry of cell types in planarians by single-cell transcriptomics. *Sci Adv* **11**, eadm7042 (2025).
4. Hafemeister, C. & Halbritter, F. Single-cell RNA-seq differential expression tests within a sample should use pseudo-bulk data of pseudo-replicates. *bioRxiv*, 2023.2003.2028.534443 (2023).
5. Scimone, M.L., Kravarik, K.M., Lapan, S.W. & Reddien, P.W. Neoblast specialization in regeneration of the planarian *Schmidtea mediterranea*. *Stem Cell Reports* **3**, 339-352 (2014).
6. Molina, M.D. & Cebria, F. Decoding Stem Cells: An Overview on Planarian Stem Cell Heterogeneity and Lineage Progression. *Biomolecules* **11** (2021).
7. Zhu, S.J. & Pearson, B.J. (Neo)blast from the past: new insights into planarian stem cell lineages. *Curr Opin Genet Dev* **40**, 74-80 (2016).
8. King, H.O., Owusu-Boaitey, K.E., Fincher, C.T. & Reddien, P.W. A transcription factor atlas of stem cell fate in planarians. *Cell Rep* **43**, 113843 (2024).
9. Poulet, A., Kratkiewicz, A.J., Li, D. & van Wolfswinkel, J.C. Chromatin analysis of adult pluripotent stem cells reveals a unique stemness maintenance strategy. *Sci Adv* **9**, eadh4887 (2023).
10. Emms, D.M. & Kelly, S. OrthoFinder: phylogenetic orthology inference for comparative genomics. *Genome Biol* **20**, 238 (2019).
11. van Heeringen, S.J. & Veenstra, G.J. GimmeMotifs: a de novo motif prediction pipeline for ChIP-sequencing experiments. *Bioinformatics* **27**, 270-271 (2011).
12. Lambert, S.A. *et al.* Similarity regression predicts evolution of transcription factor sequence specificity. *Nat Genet* **51**, 981-989 (2019).
13. Neuro, J., Sridhar, D., Dattani, A. & Aboobaker, A. Identification of putative enhancer-like elements predicts regulatory networks active in planarian adult stem cells. *Elife* **11** (2022).
14. Wurtzel, O. *et al.* A Generic and Cell-Type-Specific Wound Response Precedes Regeneration in Planarians. *Dev Cell* **35**, 632-645 (2015).
15. Yanai, I. *et al.* Genome-wide midrange transcription profiles reveal expression level relationships in human tissue specification. *Bioinformatics* **21**, 650-659 (2005).
16. Lobo, D., Morokuma, J. & Levin, M. Computational discovery and in vivo validation of *hnf4* as a regulatory gene in planarian regeneration. *Bioinformatics* **32**, 2681-2685 (2016).

Reviewer #2 (Remarks to the Author):

This revised manuscript is considerably improved over the original. The manuscript now has a direction, and the figures are legible. The authors also cleaned up the content of the figures by moving some of the busy panels to the supplement and adding new simplified versions of several of the plots as well as explanatory schematics. In addition, they added some RNAi experiments to help interpret their finding of co-regulated transcription factors. Together, these changes makes that the manuscript is easier to read. However the conceptual novelty that is presented by the manuscript remains meager. I still struggle to define what it is that we have learned from this manuscript that we did not know beforehand. Further, while I appreciate the author's aim to fully interpret the data by adding in inferences from many different sources, in several instances the accumulation of inference upon inference pushes the conclusions beyond the data.

We thank the reviewer for their comments, and hope that our explanations clarify our inferences and how they relate to the data. In response to this we have rephrased several instances, and made changes to our discussion figure, as outlined below.

The authors find two “supergroups” of cell types, which were not reported before, and they provide more detail on the *hnf-4*+ supergroup to show that this includes intestinal cells as well as a subset of parenchymal cells. They further show that *hnf-4* works alongside other transcription factors in these two different cell types, and that this in part explains that different genes are affected by loss of *hnf-4* in the two different cell types. In one case they identify a transcription factor that appears to function downstream of *hnf-4* (*nkx-2.2*) as its elimination intensifies the effect of *hnf-4* knockdown without many additional targeted genes. In another case they find a transcription factor that targets partly overlapping genes in the same cells, but clearly has other functions as well. This is interesting, and indeed had not been reported before, but it is not necessarily insightful.

We think that our experiments dissect at the single cell level the different effects of a TF in different cell types. This TF had been studied earlier, often considering only its expression in one of the cell types. Our experiments not only disentangle these but offer methodological avenues for other TFs that are similarly expressed in multiple cell types.

To give more meaning to the supergroups, the authors appear to interpret them as developmental decision points. They state that they help explain the “regulatory logic of planarian stem cell differentiation” suggesting that the stem cells first choose a supergroup and then continue their differentiation trajectory into the different cell types. However, there is no data to support such a logic. The supergroups definitely have certain shared sets of expressed genes and chromatin accessible regions, but that could just as well be the result of using similar functional modules in completely independently developed cell lineages. For example the genes involved in the construction of cilia (as mentioned by the authors themselves), or the machinery for phagocytosis (which could well be the case for *hnf-4*), or for protein secretion, can be highly active in multiple independent cell types, without a shared lineage relationship. This doesn't mean that such modules are not interesting, but there is no evidence that they have anything to do with the logic of stem cell differentiation. The authors acknowledge that there could be other reasons for the gene similarities than developmental lineage, but seem to discard such explanations based on the fact that the cells they name “goblet cells” are also affected by *hnf-4*, and that goblet cells in vertebrates are thought to be exocytotic rather than phagocytotic, but 1. it is not known whether these cells really are homologous to goblet cells, and 2. it is not known whether planarian goblet cells would not have a phagocytic activity.

Interpretation of the supergroups as developmental transition states would require the identification of these intermediate stages.

We do not discard functional relationships underlying the groupings, we only said that the fact that goblet cells are likely secretory argue against it. We have softened this to “challenging the notion” in the text. Regarding the goblet cells, in spite of their homology with vertebrate goblet cells, other authors from classical microscopists to molecular biologists have argued that they secrete enzymes into the lumen. We have added recent references that discuss this. Our wording was careful (“are thought to have exocytic activity”), and we have maintained that.

Further, the interpretation of these supergroups as developmental stages conflicts with the conclusions in the recent paper by King et al. describing the planarian transcription factors in the neoblasts. It would be helpful to comment on why these studies reach different conclusions concerning neoblast differentiation.

We do not think that our results are contradictory with those of King *et al*, even in the hypothetical scenario that our groups were indeed real lineage decisions. The key question of King et al is at which differentiation stage these lineage decisions (to dozens of specific cell types) are made. Based on single cell data of X1 cells and on clustering experiments and UMAP interpretations, they observe that in some cases, many of these lineage decisions are already made in the S/G2/M neoblasts. That is not inconsistent with those decisions going through several steps, including a first step of choosing a supergroup, to then chose a broad group and then chose a specific cell type, provided that these stages are still contained within the “neoblast” stage. We also want to stress that the barrier from “neoblast” to “postmitotic progenitor” is still hard to draw. King *et al* use “X1” cells (i.e. 4C DNA neoblasts), we use a neoblast single cell cluster, and some committed neoblasts might cluster with their respective differentiated fates. In summary, our findings do not contradict the findings by King et al. when decoupling the molecular dynamics (specific coordinated TFs regulating downstream gene modules via specific OCRs) from the timing in the developmental trajectory (cell cycle state, pre-mitotic/progenitor state, direct specification, progenitor diversification, or diversity at maturation). The question deserves further attention, and our data will help in these future discussions.

However, we agree with the reviewer that our discussion figure conveyed the lineage idea too strongly. The discussion of how our supergroups fits with the analysis by King *et al*. is a nuanced and complex one, and we now think that this is beyond the scope of our discussion. Therefore, in response to this comment, we have removed this section of the discussion figure, keeping only the lower part that conveys more clearly our results. We have edited the discussion to include the possibility that lineage decisions happen within the neoblast compartment, and cited King et al in this passage.

The identification of “influential factors” is interesting, but the interpretation needs to be more cautious. The analysis assumes that transcription factors that undergo large changes in expression level on the RNA level and whose predicted binding site (based on best matches to mammalian transcription factors) is common among altered genes, are the master regulators. This however doesn't have to be the case. Some binding sites are just far more common than others (and in an AT-rich genome such as Schmidtea AT-rich motifs are very common). Further, some transcription factors may achieve effects by small fold changes, and they may regulate only a few other transcription factors who then do the majority of the legwork. Additionally, changes on RNA level are

still a step away from protein levels. Combined, this makes that there is a lot of uncertainty in these analyses. This again doesn't mean that the identified transcription factors are not interesting, but their interpretation should be clarified and softened, and showing them as influential factors requires experimental verification.

We used the word "influential" as it is the one used in the original ANANSE publication. We define what influence means when it is first used in the results section: "TFs whose expression changes the most and show highest binding to differentially expressed genes between two networks". We have lightly edited this part to make it clearer. We have also edited one sentence at the end of the discussion to stress that they are predictions, and to stress the necessity of functional validation.

To experimentally address the second supergroup, the authors include a knockdown of *alx3-1* and show that this results in the reduction of staining by a probe matching a predicted secretory cell transcript. They interpret this as evidence that *alx3-1* has a role "in the maintenance of secretory cells" (line 344). This is not shown though. It is unknown whether the referenced secretory cells are gone, or whether they just express less of this transcript. It is a common mistake in the field to take the loss of a marker for the loss of a cell type, but this is of course not at all guaranteed. Additionally, there is no evidence that the maintenance of these cells rather than the generation or the gene expression would be affected.

The reviewer is right. We have rephrased this section to explicitly mention the two possibilities. Our data shows that *alx3-1* is needed for either the expression of the marker or the maintenance of the cell type as a whole.

Overall, I think the authors are on the right track, and the manuscript is definitely improved, but the remaining issues should still be addressed.

Minor:

line 33-34

"showcasing that the combination of single cell methods and perturbational studies will be key for characterising GRNs widely"

I am not sure that this can count as a new insight. It would be helpful if the authors can better define what the readers should take away from this manuscript.

We have changed this part of the sentence for: "showcasing a comprehensive catalogue of GRN computational inferences that will be key to study planarian stem cell differentiation".

line 65-66

It is unclear to me why these predictive approaches would not scale to whole organisms.

Often these approaches have been used in individual tissues containing a rather small number of terminally differentiated cell types. Their use in more complex scenarios, with different tissues in different stages of differentiation is more challenging. This is what we

attempt in this manuscript. We have rephrased this passage to “*and it is unclear if these methods can scale beyond individual tissues to the whole complex organisms*” to clarify.

line 85-86

“We identified key transcription factors involved in the differentiation of all major cell lineages derived from planarian stem cells.”

This is not shown - only predicted.

We have changed the word here for “predicted”.

lines 301-307

This section mentions Fox-F1 several times, but I don't find it in the referenced figures (Figure 4B and Supplementary Figure 13).

We have revised all the figures and have renamed *foxF-1* where applicable. We also improved the labelling in some figures (e.g. labelling of control and *hnf4(RNAi)* for clarity.

The legends for many of the figure panels are still too brief to fully understand what is shown in the plots.

We summarised the Figure Legends to better comply with the journal's guidelines.

Reviewer #4 (Remarks to the Author):

In the revised manuscript by Pérez-Posada et al, the authors have improved the presentation of their data and the rigor of their analyses. They have addressed most of our minor points, with one significant exception that greatly impacts their ability to address our major points. Specifically, we requested see a UMAP showing the expression of *hnf4* in control versus *hnf4*(RNAi) animals, as we were curious to know if some clusters of cells were affected by the RNAi depletion more than others. Yet in the revision, the authors show that *hnf4* is not reduced; in fact, it increases in expression. The authors provide some theoretical explanations for why this may be and point to other published datasets that also show lack of target transcript knockdown after RNAi (indirect effects that lead to upregulation of *hnf4*, possible detection of dsRNA in the scRNA-seq, varied knockdown efficiency across cell types, etc.). However, these explanations are not particularly satisfactory. First, scRNA-seq should allow for the detection of differential knockdown between cell types. It is true that TFs are often lowly expressed, but *hnf4* expression does not appear to be at the lower end of detection (in this data or other datasets). Second, the tracks showing RNA-seq read mapping to the *hnf4* locus does not convincingly suggest that the increase in *hnf4* transcript in *hnf4*(RNAi) worms is due to an increase in unprocessed nuclear mRNAs. The pattern of RNA-seq reads is very similar in the *gfp*(RNAi) control track, including within introns; the signal is just lower overall in the control *gfp*(RNAi) RNA-seq alignment track. Moreover, this data does not provide any real evidence that HNF4 protein levels are reduced as a result.

Further, knockdown of *hnf4* is the foundation of many experiments of this paper, including those the authors argue make it an important contribution to the field. Stating that RNAi in planarians often does not lead to the depletion of specific target transcripts in other publications is concerning, not reassuring of the conclusions of this paper. Although it may be true that the planarian field should study the dynamics and mechanism of RNAi in this system, the experiments in this paper still assume that injecting dsRNA of *hnf4* sequence will lead to specific *hnf4* transcript and protein depletion. If the authors believe there is an initial knockdown of *hnf4* that subsequently leads to its upregulation through an unknown feedback loop, this can be experimentally tested by performing qPCR for *hnf4* at multiple time points after RNAi (using bulk RNA isolated from control and *hnf4*(RNAi) worms). The authors can also take care to design their qPCR primers so that they do not overlap with their *hnf4* dsRNA construct, a precaution to be sure their qPCR assay is not detecting remnants of the dsRNA treatment.

We agree with the reviewer that this is a confusing aspect of our results and analyses. We have been trying to understand this phenomenon from the very beginning. We argue that many processes, both technical and biological, can account for this. Furthermore, our analysis provides many other sources of evidence indicating that the experiment worked, as explained below, and it would be difficult to conceive these being the product of random chance. We also argue that this situation is common in other experiments and deserves further attention. We provide further information and discussion in this response. We want to start by clearly stating these four points:

- 1) **The mRNA measurement after RNAi is an *indirect* measurement of protein activity.** We disagree with the reviewer in that the level of mRNA knockdown provides real evidence of the protein levels, it can only inform of the RNA levels. As stated in our previous response, several mechanisms can be envisioned that reduce protein activity without reducing RNA – the simplest of them being cleaving the mRNA without degrading it. The only real *direct* measurement of protein activity, in this case,

would not even be a direct quantification of the protein, but a quantification of the protein binding to DNA and mediating RNA transcription, something that to our knowledge has not been done in any planarian knockdown study. There is ample literature covering the moderate but not absolute correlation between mRNA levels and protein abundance – as post-transcriptional regulation exists.

- 2) **The measurement of mRNA is complex, and is affected by many technical parameters**, in addition to the biological processes mentioned above. In our experiments, the upregulation is seen in the single-cell experiments, but in the bulk RNA-seq we see negative log fold change, even though it is not significant. Different measurements, different oligo-dTs used, different library prep methods, and sequencing modes can also affect this. The significance is also measured with different tests, depending on the experiments having replicates, or being time series, etc, and depends on the depth of the sequencing. We think two key technical aspects differentiate our single-cell RNA-seq data from our bulk RNA-seq data: a) the use of an oligo-dTV (where V stands for A, C or G) which might have been more prone for internal priming, and b) the fact that single-cell data is inherently enriched for nuclear RNA (we discussed this in García-Castro *et al.*, in brief, sorting out cellular “debris” leaves populations of whole cells but also broken cells that contain the nucleus).
- 3) **Downregulation of the target mRNA is not always observed after RNAi**, as a product of this complexity, with both biological and technical components. We provided evidence that this is not always the case and we now provide evidence of some analyses where minor differences in the analysis (for instance, considering exonic and intronic reads, versus only intronic) induce the target gene transitioning from significant to non-significant. These give clues on what are the processes that might be taking place, and will lead to further
- 4) **Many other indirect sources of evidence in our experiment indicate that out knockdown was effective**. It would be unsatisfactory, in our opinion, to conclude that our experiment was not effective due to the lack of observable downregulation of *hnf4* mRNA in the single cell experiment, without some sort of explanation or mechanism that explains the other observations that are consistent with an effective knockdown. Chief among them are: 1) the fact that most of the differentially expressed genes are expressed (mRNA detected) in the tissues where *hnf4* is expressed, 2) the enrichment of *hnf4* motifs in the promoters of these genes, 3) the pattern dominated by mRNA downregulation (with relatively less upregulation) classic of TFs and also seen in the RNAi experiments of other TFs reanalysed in this study, 4) the reproducibility of these targets when measured by single-cell experiments and bulk RNA-seq experiments, 5) the reproducibility of the phenotypic effects across replicates, and even within experiments generated in different laboratories (Oxford and Barcelona) by different researchers (HGC, EE, AGF), 6) the fact that these phenotypes are consistent with those reported previously for *hnf4*, 7) the good agreement between the experimental differentially regulated genes and the computationally predicted ANANSE targets. In fact, we believe that from all indirect sources of evidence, the motif enrichment of the targets is the closest to the activity of the protein (i.e. transcription of target genes), as well as the higher predicted score of interaction between *hnf4* and differentially expressed genes for each cell type (Figure 7). If we are to contemplate the possibility of the knockdown not being effective, it is very difficult to conceive what processes would have created observations 1 to 6 out of random chance.

In the following section, we elaborate on some of the aspects mentioned above, providing further information and analyses.

- Correlation of mRNA levels and protein levels.

Gene expression is a nuanced, multi-layered process that involves not only transcription, but also processing, modification, and translation, of the mRNA and the resulting proteins¹. The relationship between transcription and protein translation is a long-standing field that has been studied thoroughly since the advent of molecular techniques. While there is much left to investigate, there is consensus that the relationship between how much a gene is transcribed and how much that transcript is translated, is non-linear. For instance, the dynamic range of variation of protein levels is several orders of magnitude higher than that of transcripts, and the rates at which proteins are synthesised and degraded is different to those of mRNA. In other cases, translation does not take place even if some genes are transcribed². These rates also differs across genes and organism compartments (tissues, cell types). Thus, current efforts approximate mRNA and protein covariation using models that integrate transcription and translation rates with mRNA and protein half-life^{1,3}. It has been observed that protein half-life varies depending on the role of the protein in question. For example, proteins required for decisions such as cell fate tend to have low, stable levels of mRNA that get translated in pulses, yielding short-lived proteins that do not linger. Such is the case of developmental transcription factors. A study by Zecha et al., 2018⁴ measured the half-life of multiple protein families, including transcription factor classes, and found that several TF classes have lower half-life, including the Nuclear Receptor (NR) class. This is the TF class where *hnf4* belongs. Despite the half-life of HNF4 protein has not been described in planaria, *hnf4* is fairly well conserved across animal lineages, both its sequence and its role in cell fate commitment (as shown by us and other authors in the planarian literature). Our observations regarding higher amount of mRNA in knockdown cells (see below) are thus compatible with a scenario where mRNA degradation via RNAi might be interfering with the on-demand burst of translation of short-lived HNF4 protein in planarian cells, which try to compensate it by increasing mRNA transcription.

- Downregulation/upregulation of *hnf4* in bulk vs pseudobulk single-cell samples

When performing differential gene expression analyses in bulk samples (double knockdown, AGF experiment), we see that *hnf4* is not upregulated, in turn having small negative log₂FoldChanges (-0.3539) (Supplementary File 32). The difference between bulk and single cell can be explained by different methods of capturing mRNA in bulk vs SPLiT-Seq (see below).

- Oligo-dT used in each assay

Following on the observations reported in the previous revision, we generated genome mapping tracks for all the replicates and conditions separately, including also our double knockdown experiments (which we mapped to the genome as our DGE analysis was done using kallisto against a set of longest transcripts per gene). These tracks can be seen in Figure Revision A, where we see *hnf4* has mapping coverage in all the samples, including mapping at the intron level. Of note are several regions where there seems to be higher coverage than elsewhere in the intron. One of the reasons why this might have occurred is by internal priming, in turn derived from the oligo-dT used in the single cell experiment. While we used oligo-dT for the double knockdown experiments, our custom

SPLiT-Seq protocol uses oligo-dTV, which is comprised by a pool of different molecules (oligo-dT with a last nucleotide of either A, G, or C). Provided the strongest binding occurs via the first four-five nucleotides of the molecule, and that planarian DNA is AT-rich, we argue that these oligo-TV might have bound more easily into internal poly-A runs other than the mRNA polyA, including regions within the longest intron of *hnf4* (Fig. Revision-A). Indeed we observe the presence of A-rich or AT-rich regions flanking the peak of intronic mapping in single cell samples (Figure Revision-A).

- Nuclear enrichment of single cell samples.

While we still see mapping in the double knockdown samples, the intronic mapping relative to exonic mapping is higher in single cell samples. This has been discussed previously⁵. Essentially, all cell dissociation methods break cells, and then, when “whole cells” (particles with DNA and with cytoplasm) are sorted out from “debris” (particles with cytoplasm but no DNA labelling) this produces a selective elimination of cytoplasmic RNA, i.e. a selective enrichment of nuclear RNA. Nuclear RNA is highly enriched in pre-mRNA. This accounts for a higher capture rate of pre-spliced *hnf4* mRNA compared to the double knockdown experiment, with potential implications in the amount of mapping both to exons and introns (see below).

- Differences in intronic and exonic mapping on *hnf4* gene and other knockdowns

As shown in Figure Revision A, there are several annotated isoforms of the *hnf4* gene (per the latest available version of the planarian genome⁶). As customary practice in our analyses, we collapsed all mapping at the gene level when quantifying gene expression in single cell data (regardless of isoforms), and we mapped the bulk double knockdown samples to the set of longest/protein coding transcripts per gene (in the case of *hnf4*: h1SMcT0019688.4). Therefore, another possibility is that the increased levels of mRNA do not correspond to the protein-coding, longest isoform of *hnf4*, but to other putative isoforms such as h1SMcT0019688.2 or h1SMcT0019688.3, which seem to retain the second and third longest introns of the *hnf4* gene. This could account for an increase in *hnf4* mRNA that does not lead to an increased protein level.

We then quantified the exonic and intronic mapping separately and calculated the log₂FoldChange of expression at the exon and the intron level, separately (Figure Revision B-G). We named the *hnf4* exons as E01-E10 (from right to left, as *hnf4* is annotated on the “-“ strand), and introns as I01-I09. Overall, we observed that the *hnf4* introns showcase an average positive log₂FoldChange in both experiments (single cell and bulk double knockdown) (Fig. Revision-C). This is not the case for the *hnf4* exons, which showcase a negative log₂FoldChange in bulk experiments. While they show a higher log₂FoldChange in the single cell experiment, it is important to note that exonic mapping does not necessarily translate to higher amount of gene product (*hnf4* protein; see our point above), as pre-spliced mRNA has both exons and introns. Pre-spliced exonic mapping can also be one reason for these higher log₂FoldChanges. If the cellular pool of pre-spliced mRNA were higher than the pool of spliced mRNA, our SPLiT-seq technique (sparser than bulk RNA) likely sampled more often from pre-spliced mRNA. This goes in line with the nuclear enrichment in single cell samples (see comment above).

For a deeper look into the bulk double knockdowns experiment, we compared the exonic and intronic mapping of *hnf4* across the different conditions of *hnf4* dosage (Figure Revision D). Interestingly, both exonic and intronic mapping are comparable yet higher

in the full-dose knockdown (*hnf4(RNAi)*) relative to the half-dose knockdown (*hnf4(RNAi)+gfp(RNAi)*). Overall, to us this suggests that higher doses of dsRNA might be inducing a higher transcriptional response to overcompensate the gene silencing, which goes in line with the reviewer suggestion that there might be an initial decrease followed by a compensatory effect. These log₂FC values are on the smaller end of variability, however, so interpretation should be careful.

We then proceeded to investigate whether we might be capturing part of the dsRNA itself. As part of this, we first checked the strandedness of our data by using RSeQC^{7,8} and observed that while our bulk RNA-Seq is unstranded, our SPLiT-Seq data is stranded (in the sense that it has directionality). We then quantified exonic and intronic mapping considering the orientation of the reads relative to the orientation of the genes (strandness) and observed that overall reverse mapping is lower for every sample regardless of condition. Interestingly, for *hnf4*, reverse exonic mapping (in the “+” strand, as *hnf4* is annotated in the “-” strand) is only high for exons E06, E07, and E08, which share sequence identity with the dsRNAi used in both experiments (Fig. Revision-F). We could not assess this in our double knockdown as it is not a stranded RNA-Seq. At any rate, these higher mappings coming from pre-spliced mRNA and/or reverse RNA cannot translate into higher levels of protein, which goes in line with our previous point. We assessed the median difference of exonic log₂FC between reverse and forward mappings and used this as a score to sort all transcription factors for which we detected forward but also reverse mapping in at least one exon. We observed that *hnf4* lies on the top half of TFs with higher exonic log₂FC in reverse than in forward. Albeit not definitive, our observations are consistent with a potential capture of the dsRNA or derivatives of its action.

To expand beyond our *hnf4* experiment, we checked for differences in exonic and intronic mapping on other TF knockdowns (Figure Revision H). Another example where we observe differences between intronic and exonic mapping is *pax2/5/8* from Cheng et al., 2018. We previously reported *pax2/5/8* as not-significantly downregulated because we were considering a double criterion of $p_{\text{adj}} < 0.1$ and absolute log₂FoldChange > 1.5 (above 1.5 or below -1.5). *pax2/5/8* however has a small log₂FoldChange of -0.169786 but a $p_{\text{adj}} < 0.001$, and is thus likely downregulated. When counting exonic and intronic mapping separately, we see that indeed *pax5* coverage in exons have lower log₂FoldChanges (-0.305829 and -0.439782) and coverage in introns have positive or near-zero log₂FoldChange (+0.198051 and -0.024663). A less clear example is *coe*, from Cowles et al., 2013, where intronic mapping still has negative log₂FC (-0.675104) but less so than the exonic log₂FC (-1.53039). Neither of the intronic results are statistically significant, but if anything, this says that they are not downregulated at the intronic level either.

These observations are compatible with our interpretation that cells might be up-regulating the transcription of the knockdown gene of interest to compensate for the effect from RNAi.

In summary, we believe that RNAi might upregulate the pre-mRNA of the target gene. In a situation where a) single-cell experiments are enriched of nuclear RNA, b) our oligo-dTV primer can potentially target more AT rich introns, and c) RNA-seq can still capture dsRNA or its derivatives, this may have caused the observed upregulation of *hnf4* in its own knockdown registered by single cell experiment. This is not observed in bulk RNA-seq experiments, where we see downregulation, although not significant. In general,

other RNA-seq experiments show these trends (higher amount of intronic reads, lower level of knockdown when intronic reads are considered). We outlined here the different aspects that contribute to this technical and biological explanations, and stress that the matter deserves further attention from the planarian and the RNAi community. Nevertheless, we argue that mRNA level is just an indirect proxy for protein activity, and we are confident that our experiments worked due to the many other indirect measurements that indicate an effective knockdown. These include the fact that DGE is observed primarily in the tissues where *hnf4* is expressed, the *hnf4* motif enrichment in the vicinity of these DEGs, the classic TF knockdown signature of predominantly downregulation, the reproducibility of these targets across different techniques, experimenters and laboratories of origin, and the agreement between experimental DGE and ANANSE prediction. It is very hard to conceive how these would emerge in a scenario where the HNF4 TF protein was not effectively knocked-down.

Minor point:

In figure 1D, the authors label a gene marker for each annotated cell type using the gene ID. It would be helpful to also add the description/name of this gene if it is highly conserved.

We have labelled these genes where applicable, using either TF annotations or eggNOG annotation (see Methods).

Reviewer #4 (Remarks on code availability):

The code appears to be well-organized and clear to understand.

References

1. Buccitelli, C. & Selbach, M. mRNAs, proteins and the emerging principles of gene expression control. *Nat Rev Genet* **21**, 630-644 (2020).
2. Wang, D. *et al.* A deep proteome and transcriptome abundance atlas of 29 healthy human tissues. *Mol Syst Biol* **15**, e8503 (2019).
3. Schwanhauser, B. *et al.* Global quantification of mammalian gene expression control. *Nature* **473**, 337-342 (2011).
4. Zecha, J. *et al.* Peptide Level Turnover Measurements Enable the Study of Proteoform Dynamics. *Mol Cell Proteomics* **17**, 974-992 (2018).
5. Garcia-Castro, H. *et al.* ACME dissociation: a versatile cell fixation-dissociation method for single-cell transcriptomics. *Genome Biol* **22**, 89 (2021).
6. Ivankovic, M. *et al.* A comparative analysis of planarian genomes reveals regulatory conservation in the face of rapid structural divergence. *Nat Commun* **15**, 8215 (2024).
7. Wang, L., Wang, S. & Li, W. RSeQC: quality control of RNA-seq experiments. *Bioinformatics* **28**, 2184-2185 (2012).
8. Wang, L. *et al.* Measure transcript integrity using RNA-seq data. *BMC Bioinformatics* **17**, 58 (2016).

A

Figure Revision. A: Planarian genomic landscape of the *hnf4* locus. Blue annotation indicates annotated transcripts of *hnf4*. Black track indicates location of dsRNA (exonic sequence only). Track list (top to bottom): overlay of scATAC-Seq (clipped data range from 0 to 100), percentage of enrichment in A/T nucleotides (binned for 30nt windows, data range from 50% to 100%), RNA-Seq coverage of single cell samples (control and *hnf4*(RNAi), replicates B1 and B2, libraries 11.3 and 11.4; data range clipped from 0 to 50), RNA-Seq coverage of double knockdown bulk RNA-Seq samples (control and knockdowns, replicates 1-3 each; data range from 0 to 100). Lastly, alternative isoform transcripts of *hnf4*.

(Continues next page)

B

C

D

E

F

G

H

TF	Reference	contrast	feature	baseMean	log ₂ FoldChange	lfcSE	stat	pvalue	padj
coe	Cowles et al., 2014	control vs RNAi	intronic	60.32	-0.675104	0.293201	-2.30253	-0.0213055	0.450238
coe	Cowles et al., 2014	control vs RNAi	exonic	158.166	-1.53039	0.196687	-7.78082	7.20554E-15	1.19817E-12
pax2/5/8-1	Cheng et al., 2018	0d+control vs 12d+RNAi	intronic	74.5451	0.198051	0.194295	8.48596	0.204617	0.729914
pax2/5/8-1	Cheng et al., 2018	0d+control vs 18d+RNAi	exonic	365.829	-0.305829	0.121631	43.4303	9.58675E-08	1.26787E-06
pax2/5/8-1	Cheng et al., 2018	0d+control vs 12d+RNAi	intronic	74.5451	-0.0246673	0.206699	8.48596	0.204617	0.729914
pax2/5/8-1	Cheng et al., 2018	0d+control vs 18d+RNAi	exonic	365.829	-0.439782	0.12393	43.4303	9.58675E-08	1.26787E-06

Figure Revision (Cont). **B:** detailed schematics of the exons and introns of the *hnf4* gene in *S. mediterranea*. Exons have been named from E01 to E10, and introns have been named from I01 to I09. **C:** boxplots showing changes in the RNA levels for the exons and introns of *hnf4* across experiments (single cell and bulk). **D:** boxplots showing changes in the RNA levels for the exons and introns of *hnf4* across comparisons for the double knockdown bulk experiments. **E:** Top: boxplots showing the number of exonic counts in forward and reverse for all the samples of the single cell experiment. Bottom: schematics showing our naming convention for "forward" and "reverse" mapping. **F:** jitter plot showing changes in the RNA levels of the exons of *hnf4* (control vs RNAi) for each strand. Note that *hnf4* is in the "-" and thus reverse corresponds to the "+" strand. Lines connect the same introns. **G:** Ranking of *hnf4* based on the median difference of exonic foldchange (FC reverse - FC forward) in the single cell experiments for all the TFs with detected expression in forward and reverse. **H:** Table of results of quantifying differential gene expression for *pax2/5/8* and using different mapping strategies (quantifying exonic only or intronic only).

Reviewer #2 (Remarks to the Author):

I appreciate the clarifications by the authors.

Given the clarifications, the title should be changed to "Regulatory networks of planarian stem cell differentiation". The concept of logic infers something more comprehensive and mechanistic where the models would explain the decisions made by the stem cells. The authors have clarified that they do not intent to claim the existence of progressive lineage decisions, and that the identification of "influential factors" does not imply that these factors are drivers in the lineages. The order, interaction, and regulatory impact of the various transcription factors thus remains unknown. What the authors do show however is the expression of these transcription factors in the different cell populations, and potential connections to target genes. What they describe therefore are networks.

With this clarification I think the manuscript will be a valuable resource for the community.

We thank the reviewer for pointing this. We have followed this advice and have changed the title of our manuscript to **"Multimodal single cell analyses reveal gene networks of planarian stem cell differentiation"**. Regarding our use of "gene networks" instead of "regulatory networks", we argue that the classical definition of Gene Regulatory Networks (GRNs) pioneered by Davidson and colleagues generally involves the perturbation of most if not all elements of a GRN to establish their regulatory hierarchy. Provided that our joint approach of single cell, computational inference, and candidate perturbation differs from this, we consider the term "gene networks" reflects well the scope of this manuscript. Regarding our use of the expression "multimodal single cell analyses", we argue that this reflects one of the main aspects of our manuscript, which is a novel atlas of single cell ATAC-seq in planarians (at the time of initial submission) plus the combination of RNA and ATAC analyses.

Reviewer #2 (Remarks on code availability):

Code is legible and well organized.

Reviewer #4 (Remarks to the Author):

The authors of this manuscript have generated many analyses in an attempt to answer a concern I raised in my last review about the effectiveness of their hnf4 knockdown by RNAi. However, although these data suggest reasonable hypotheses for why they do not detect loss of hnf4 expression, they do not sufficiently resolve this concern.

I agree that one should not assume a direct relationship between the levels of a specific mRNA/transcript and the levels of its translated protein. I also agree that an assay measuring TF binding, such as ChIP with a TF-specific antibody, would be the most direct way to assess whether hnf4(RNAi) is having the desired effect. However, as the authors point out, this is a challenging experiment to do in this organisms given the lack of epitope conservation (making it unlikely that commercial antibodies for TFs will work) and transgenics (to add an epitope tag to the TF of interest), which is why I did not suggest that assay in my previous review despite my concerns. Yet the authors are not addressing the key point: they used RNAi with the goal of depleting HNF4 so that they could assess the effects of its loss on differentiation. The burden thus lies with them to show that the first step of this experimental paradigm was achieved.

Yes, the downstream analyses showing that most DEGs in *hnf4*(RNAi) worms are detected in those cells enriched for *hnf4* expression and often have HNF4 binding motifs supports the hypothesis that loss of HNF4 led to their reduced expression. Yet an alternative hypothesis is that these results were caused by depletion of an off-target mRNA. That scenario may seem highly unlikely, yet there is precedence for such “phenotypically significant off-target effects” in the literature, including when using long dsRNA to trigger RNAi (Ma et al Nature 2006). As suggested in this paper, and considered standard practice in many models that use RNAi, a better way to validate target specificity is by using multiple non-overlapping dsRNA constructs to induce RNAi. Confirmation that both constructs induce the same phenotypic effects, including changes in target gene expression, would support the assertion that the effects observed are due to the common target i.e. *hnf4*. In addition, as suggested in my prior review, another assay that would test several of the hypotheses suggested by the authors (detection of dsRNA itself, up-regulation of pre mRNAs) would be to use qPCR with carefully designed primers that do not match the dsRNA construct and span exon boundaries.

We thank the reviewer for sharing their concerns and we would like to add to this topic. We have now added more explanations about this absence of downregulation, the enrichment of nuclear RNA in single cell data, and the observed but not significant downregulation in the bulk RNA-seq data. The qPCR experiment, if consistent with the other lines of evidence about the mRNA level, would likely fail to find significant differences. We insist that this result would not show that the knockdown did not work, it can only fail to show that the knockdown worked. This is an important difference, at the heart of the indirect nature of this source of evidence. We have argued that other sources of indirect evidence indeed support that the knockdown was effective, and we have now explained this in the main text, and in detail in the response, for the readers to evaluate.

Regarding the existence of off-target effects (OTEs) and the possibility that some of these would be “phenotypically significant” as reported by Ma *et al*, we agree with the reviewer that this is a possibility – one that affects indeed every RNAi paper in planarians, and this is regardless of the presence or absence of a significant knockdown measured by any technique at any time point: this possibility always exists, and we never argued against it. However, there is an important aspect that we wish to clarify. Ma *et al* report that these OTEs “exist” and that they have to be further investigated. We argue that this is what we have done. These OTE-mediated effect must occur at a given rate, and this can be studied. In our case, after knocking down a transcription factor, we set out to profile its specific signature: presence of motifs in the differentially regulated genes, signature profile dominated by downregulations, effects predominant in the cell types where the TF is expressed, and good agreement with the theoretical model. It is true that all these effects can indeed be due to OTEs (as the effects reported in every planarian RNAi paper could), but these additional layers of evidence are independent, and have each their associated statistical analyses (it is true that one could argue that the network model is not “independent” and indeed uses information from the differential gene expression analysis, the tissue specificity and the motifs – but the other three are indeed independent). All this mounting evidence supports the effectivity of our knockdown. The argument against it examined here is that there are OTEs (which is indeed a very likely possibility), but that these OTEs (by chance) happen in a “phenotypically significant” way at different independent levels: they happen to be by chance enriched in *hnf4* motifs, they happen to induce by chance mostly downregulation, as a TF would do, they happen to take place by chance mostly in the tissues where *hnf4* is expressed, and they happen to overlap by chance with the results of our network model, all of these with enough strength to pass a statistical test. We believe that the possibility of this occurring is

minuscule. We have now presented these arguments in the main text and the revision materials, so that the readers can evaluate it.

The above discussion raises issues of technical and experimental rigor that are important and should be addressed, but it does not resolve the larger issue with this manuscript that we raised in our first review. The authors generate several new genomic datasets, yet it remains unclear how their newly generated data (e.g., scATAC-seq) advances the field. There are opportunities for uncovering something new here, yet they are not well explored. For example, in Figure 5 the authors nicely summarize how each dataset (scRNA-seq, scATAC-seq) classifies cells and then how the more complex analysis of TF “influence” classifies them. Yet the experiments that follow do not test which of these schemas is most supported. Such experiments could add significant support for their use of scATAC and analysis methods, advancing the field.

As stated previously in other responses to the reviewers, we argue that the main novelty of our manuscript are two-fold: first, the formulation of a hypothesis that planarian differentiated cell types can be grouped into super-groups based on shared gene regulatory features, including co-expression of gene modules, co-accessibility of chromatin regions, and co-regulation of the transition from stem cell to differentiated cell by common sets of TFs. Second, our manuscript provides a useful resource for the community to explore the co-expression and co-accessibility landscapes of all the major planarian cell types at a cellular resolution, and to formulate novel hypotheses about gene regulation and stem cell differentiation.